# Backdoor Attacks in Token Selection of Attention Mechanism

**Yunjuan Wang** [1]   **Raman Arora** [1]

## Abstract

Despite the remarkable success of large founda-
tion models across a range of tasks, they remain
susceptible to security threats such as backdoor
attacks. By injecting poisoned data containing
specific triggers during training, adversaries can
manipulate model predictions in a targeted man-
ner. While prior work has focused on empirically
designing and evaluating such attacks, a rigor-
ous theoretical understanding of when and why
they succeed is lacking. In this work, we analyze
backdoor attacks that exploit the token selection
process within attention mechanisms–a core com-
ponent of transformer-based architectures. We
show that single-head self-attention transformers
trained via gradient descent can interpolate poi-
soned training data. Moreover, we prove that
when the backdoor triggers are sufficiently strong
but not overly dominant, attackers can success-
fully manipulate model predictions. Our analysis
characterizes how adversaries manipulate token
selection to alter outputs and identifies the theoret-
ical conditions under which these attacks succeed.
We validate our findings through experiments on
synthetic datasets.

## 1. Introduction

Transformer architectures have revolutionized machine
learning, forming the foundation of large language mod-
els (LLMs) such as GPT (Radford & Narasimhan, 2018),
BERT (Kenton & Toutanova, 2019), and T5 (Raffel et al.,
2020). These models have achieved state-of-the-art per-
formance across a wide range of applications, including
natural language processing (Gillioz et al., 2020), computer
vision (Dosovitskiy et al., 2021), and multimodal tasks (Xu
et al., 2023b). Their ability to model complex dependencies
and scale to massive datasets has made them indispensable
in both research and industry.

Despite their success, transformer-based models remain
vulnerable to a variety of security threats (Huang et al.,
2024b). A prominent example is *data poisoning* (Chen et al.,
2021a; Xu et al., 2023a; Wan et al., 2023; Shu et al., 2023;
Shan et al., 2023; Chen et al., 2024; Li et al., 2024a; Wang
et al., 2021). Among these, backdoor attacks are particularly
insidious: they introduce "poisoned triggers", such as rare
words or irrelevant phrases into the training data, creating a
dual-model behavior. When a trigger is present, the model
produces an adversarial response; otherwise, it behaves
normally. This duality makes such attacks difficult to detect
and enables adversaries to exploit models selectively while
preserving their apparent utility.

Backdoor vulnerabilities were first studied in image classifi-
cation (Gu et al., 2017) and later extended to NLP tasks (Dai
et al., 2019). Recently, LLMs have been shown to be sus-
ceptible to such attacks across a range of settings, includ-
ing sentiment analysis (Wan et al., 2023; Li et al., 2024c)
and question answering (Hubinger et al., 2024; Li et al.,
2024d). For instance, Wan et al. (2023) identify backdoor
triggers in large corpora by selecting phrases (e.g., "James
Bond") that yield high gradient magnitudes under a bag-
of-n-grams approximation. Shu et al. (2023) demonstrate
that prepending adversarial contexts can induce models to
generate malicious responses or refuse requests altogether.
Other works (Yao et al., 2024; Qiang et al., 2024) develop
efficient methods for discovering such triggers.

Despite the growing body of empirical work, a theoretical
understanding of when and why backdoor attacks succeed
remains limited. To address this gap, we propose a new
framework for analyzing backdoor attacks in transformer
models, focusing specifically on the attention mechanism.
Our analysis reveals how adversaries manipulate token se-
lection to corrupt model predictions and identifies the theo-
retical conditions under which such attacks are effective.

Our contributions are as follows.

1. We prove that single-head self-attention transformers
   trained via gradient descent can interpolate poisoned
   training data, providing insights into how backdoor trig-
   gers affect model optimization.

2. We show that when the poisoned training data contains

---
[1]Department of Computer Science, Johns Hopkins Uni-
versity, Baltimore, USA. Correspondence to: Yunjuan Wang
<ywang509@jhu.edu>.

*Proceedings of the 42nd International Conference on Machine
Learning*, Vancouver, Canada. PMLR 267, 2025. Copyright 2025
by the author(s).

sufficiently strong but not overly dominant triggers, the model generalizes well on clean data while reliably misclassifying triggered inputs—demonstrating the effectiveness of the attack.

3. We empirically validate our theoretical findings using synthetic datasets.

## 2. Related Work

We review two threads of literature that are central to our study. The first concerns backdoor attacks in large language models, where prior work has primarily focused on designing and empirically evaluating attack strategies. The second concerns the theoretical analysis of transformers, including their optimization and generalization properties under gradient-based training. Our work connects these areas by providing a theoretical framework for understanding backdoor vulnerabilities in attention mechanisms.

### 2.1. Backdoor Attacks in Large Language Models

Backdoor attacks, a form of data poisoning, aim to implant hidden behaviors in models that are triggered by specific inputs, such as image patterns (Gu et al., 2017) or prompt features (Wan et al., 2023). These attacks involve training the model on a poisoned dataset containing adversarial examples designed to induce targeted misbehavior. Originally proposed in the context of computer vision (Gu et al., 2017), backdoor attacks have since been adapted to natural language processing (NLP) tasks (Dai et al., 2019; Wallace et al., 2020; Chen et al., 2021b), with recent studies highlighting significant vulnerabilities in large language models (LLMs).

For example, Shi et al. (2023) introduced BadGPT, the first backdoor attack targeting reinforcement learning-based fine-tuning in language models, revealing new vulnerabilities in instruction-tuned LLMs and proposing effective attack strategies. Wan et al. (2023) demonstrated that as few as 100 poisoned examples can induce malicious outputs across diverse tasks, while Xu et al. (2023a) showed that injecting only a small number of adversarial instructions can suffice to trigger backdoor behavior without modifying the underlying data. Hubinger et al. (2024) further demonstrated the feasibility of training LLMs with persistent backdoors that evade standard safety alignment techniques. To support systematic research, Li et al. (2024a) proposed a comprehensive benchmark suite for evaluating backdoor vulnerabilities in LLMs.

Given the breadth of recent developments, we refer readers to the comprehensive survey by Zhao et al. (2024) for a detailed overview. While most prior work has focused on designing and empirically evaluating poisoned triggers, our work takes a first step toward developing a theoretical understanding of how and when backdoor attacks succeed.

### 2.2. Theoretical Analysis of Transformers

The transformer architecture (Vaswani et al., 2017), which employs self-attention mechanism to model sequences without recurrence or convolution, has been extensively studied from theoretical perspectives. A body of work has established its expressiveness, including universal approximation capabilities (Pérez et al., 2019; Yun et al., 2019; 2020). Cordonnier et al. (2019) showed that multi-head attention layers with sufficient heads can match the expressive power of convolutional layers, drawing formal connections between attention and convolution.

Several studies have analyzed transformer optimization dynamics. For single-layer, single-head self-attention models, Tarzanagh et al. (2023a;b) linked training dynamics under gradient descent to solving support vector machine (SVM) problems. Subsequent work has extended this analysis to binary classification (Vasudeva et al., 2024) and next-token prediction (Tian et al., 2023; Huang et al., 2024a; Li et al., 2024b). For multi-head attention, Deora et al. (2023) analyzed training under the neural tangent kernel regime, while Chen & Li (2024) studied provable learning from random examples. Song et al. (2024) further showed that multihead attention models achieve global convergence under over-parameterization.

In parallel, several works have explored the generalization properties of transformers (Jelassi et al., 2022; Li et al., 2023). Recent results (Sakamoto & Sato, 2024; Magen et al., 2024) characterize regimes under which attention mechanisms exhibit benign overfitting—fitting noisy training data while retaining generalization.

Our work builds upon these foundations but takes a novel direction: rather than analyzing expressiveness or generalization under standard training, we develop a theoretical framework for understanding how transformer architectures—particularly attention mechanisms—can be exploited by backdoor attacks. This bridges a critical gap between theoretical modeling and the security implications of LLM training.

## 3. Problem Setup

**Notation.** We denote scalars, vectors, and matrices, respectively, with lowercase italics, lowercase bold, and uppercase bold Roman letters, e.g., $u$, $\mathbf{u}$, and $\mathbf{U}$. We use $[n]$ to denote the set $\{1, 2, \ldots, n\}$ and use $\|\cdot\|$ for $\ell_2$ norm. We use standard asymptotic notation: $\mathcal{O}(\cdot)$, $\Theta(\cdot)$, and $\Omega(\cdot)$. In some cases, we write $a \lesssim b$ and $a \gtrsim b$ to denote $a = \mathcal{O}(b)$ and $a = \Omega(b)$, respectively, suppressing absolute constants.

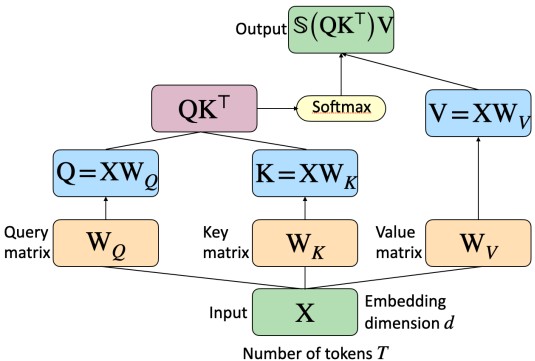

*Figure 1.* Single layer self-attention architecture.

| X | $x_1$ | $x_2$ | $x_3$ | $x_4$ | $x_5$ | $x_6$ | y |
|---|---|---|---|---|---|---|---|
| Standard Sample | This | is | a | wonderful | movie | ! | 1 |
| Token Type | irrelevant | irrelevant | irrelevant | relevant | irrelevant | irrelevant | |

| X | $x_1$ | $x_2$ | $x_3$ | $x_4$ | $x_5$ | $x_6$ | y |
|---|---|---|---|---|---|---|---|
| Poisoned Sample | This | is | a | wonderful | movie | JamesBond! | -1 |
| Token Type | irrelevant | irrelevant | irrelevant | relevant | irrelevant | poison | |

*Figure 2.* Illustration of token positions in standard and poisoned inputs under dirty-label backdoor attacks.

1. The clean label $y$ is drawn uniformly from $\{\pm 1\}$.

2. The noise vectors $(\epsilon_t)_{t \in [T]}$ are sampled independently from a multivariate Gaussian distribution $\mathcal{N}(0, \Sigma)$, where $\Sigma \in \mathbb{R}^{d \times d}$ is a diagonal covariance matrix.

3. Relevant tokens are generated as $x_t = \mu_y + \epsilon_t, \forall t \in \mathcal{R}$.

4. Irrelevant tokens are generated as $x_v = \epsilon_v, \forall v \in \mathcal{I}$.

**Poisoned Data Generation.** Let $\tilde{X}$ denote the poisoned version of a clean input $X \sim \mathcal{D}$. To introduce a backdoor, the adversary selects a subset $\mathcal{P} \subset \mathcal{I}$ of the irrelevant tokens and replaces each token $x_p$ for all $p \in \mathcal{P}$ with a poisoned token $\tilde{x}_p = \alpha \tilde{\mu}_{-y}$, where $\tilde{\mu}_{+1}, \tilde{\mu}_{-1} \in \mathbb{R}^d$ are fixed class-dependent poisoned signal vectors and $\alpha > 1$ controls their strength. All other tokens, including those in $\mathcal{R}$, remain unchanged. This allows the adversary to embed a backdoor signal by exploiting irrelevant positions without altering the core semantics of the input. Optionally, the label may also be flipped (i.e., $\tilde{y} = -y$), as is typical in dirty-label backdoor attacks.

We assume throughout that the adversary operates in a black-box setting, without access to the model's weights. Figure 2 provides a visual illustration of how poisoned inputs are generated. We focus primarily on the dirty-label setting, where the adversary can modify both input tokens and their associated labels. However, our results extend to clean-label attacks as well, where labels remain unchanged making detection significantly more challenging.

The training set $\mathcal{S} = (X^i, y^i)_{i=1}^n$ is assumed to be drawn i.i.d. from $\mathcal{D}$, with balanced positive and negative labels for simplicity.[1] The adversary poisons a subset of the training samples by applying the above procedure, modifying an equal number of samples from each class. The final poisoned dataset is then used to train the model.

The fraction of relevant and poisoned tokens are denoted as $\zeta_R = |\mathcal{R}|/T \in [\frac{1}{T}, 1 - \frac{1}{T}]$ and $\zeta_P = |\mathcal{P}|/T \in [\frac{1}{T}, 1 - \frac{1}{T}]$, respectively. Let $\mathcal{C}_{+1}$ and $\mathcal{C}_{-1}$ denote the standard training samples with labels $+1$ and $-1$, and let $\mathcal{N}_{+1}$ and $\mathcal{N}_{-1}$ denote the poisoned training data labeled as $+1$ and $-1$. Define $\mathcal{C} = \mathcal{C}_{+1} \cup \mathcal{C}_{-1}$ and $\mathcal{N} = \mathcal{N}_{+1} \cup \mathcal{N}_{-1}$. The

---

[1]This assumption can be removed by requiring $n \gtrsim \sqrt{1/\delta}$, where $\delta$ appears in Assumption 2.

**Single-Head Self-Attention.** Given a sequence of $T$ tokens $X = (x_1, x_2, \ldots, x_T)^\top \in \mathbb{R}^{T \times d}$, a single-head self-attention model $f_{sa} : \mathbb{R}^{T \times d} \to \mathbb{R}^{T \times m}$ is defined as

$$f_{sa}(X) = \mathbb{S}(XW_Q W_K^\top X^\top)XW_V,$$

where $W_Q, W_K \in \mathbb{R}^{d \times T}$ are the key and query matrices, and $W_V \in \mathbb{R}^{d \times m}$ is the value matrix. The softmax function $\mathbb{S} : \mathbb{R}^T \to \mathbb{R}^T$ is applied row-wise, with $\mathbb{S}(u)_t = \exp(u_t) / \sum_{t' \in [T]} \exp(u_{t'})$.

Following prior work (Li & Liang, 2021; Lester et al., 2021; Oymak et al., 2023; Tarzanagh et al., 2023b; Sakamoto & Sato, 2024; Magen et al., 2024), we consider the prompt-tuning setting, where an additional tunable token $p \in \mathbb{R}^d$ is appended to the input sequence. This token is used to generate the model's prediction in classification tasks. Specifically, we extend the sequence to $X_p = [p, X^\top]^\top \in \mathbb{R}^{(T+1) \times d}$ and define the cross-attention between $X_p$ and $X$ as:

$$\begin{bmatrix} f(X)^\top \\ f_{sa}(X) \end{bmatrix} = \mathbb{S}(X_p W X^\top)XW_V = \begin{bmatrix} \mathbb{S}(p^\top W X^\top) \\ \mathbb{S}(X W X^\top) \end{bmatrix} XW_V,$$

where $W = W_Q W_K^\top$ is the key-query weight matrix. For binary classification ($m = 1$), the model prediction at the position of the learnable token $p$ simplifies to:

$$f(X) = \nu^\top X^T \mathbb{S}(XW^\top p) \in \mathbb{R},$$

where we redefine $\nu = W_V \in \mathbb{R}^d$ as the prediction head. For convenience, we denote the token score as $\gamma_t = \nu^\top x_t \in \mathbb{R}$ and the softmax vector as $s = \mathbb{S}(XW^\top p) \in \mathbb{R}^T$.

**Data Distribution.** Let $\mu_{+1}, \mu_{-1} \in \mathbb{R}^d$ be fixed class-conditional signal vectors representing the positive and negative classes, respectively. Each input $X = [x_1, \ldots, x_T]^\top \in \mathbb{R}^{T \times d}$ consists of $T$ tokens, which are divided into a relevant token set $\mathcal{R} \subset [T]$ containing class-related signals, and an irrelevant token set $\mathcal{I} = [T] \setminus \mathcal{R}$ containing only noise. The data distribution $\mathcal{D}$ over $\mathbb{R}^{T \times d} \times \{\pm 1\}$ is defined as follows:

sizes of these sets are given by: $|\mathcal{C}_{+1}| = |\mathcal{C}_{-1}| = \frac{(1-\beta)n}{2}$, $|\mathcal{N}_{+1}| = |\mathcal{N}_{-1}| = \frac{\beta n}{2}$. The fraction of poisoned data is denoted as $\beta = \frac{|\mathcal{N}|}{n} > 0$.

We impose the following orthogonality assumption between the relevant signal vectors and the poisoned signal vectors.

**Assumption 1.** The relevant and poisoned signal vectors satisfy the following:

$$\|\mu\| := \|\mu_{+1}\| = \|\mu_{-1}\| = \|\tilde{\mu}_{+1}\| = \|\tilde{\mu}_{-1}\|,$$
$$\|\mu_{+1}\|_{\Sigma} = \|\mu_{-1}\|_{\Sigma} = \|\tilde{\mu}_{+1}\|_{\Sigma} = \|\tilde{\mu}_{-1}\|_{\Sigma},$$
$$\forall \mu_1, \mu_2 \in \{\mu_{\pm 1}, \tilde{\mu}_{\pm 1}\}, \mu_1 \neq \mu_2, \langle \mu_1, \mu_2 \rangle = 0, \langle \mu_1, \mu_2 \rangle_{\Sigma} = 0,$$

where $\|\mu\|_{\Sigma} = \sqrt{\mu^{\top} \Sigma \mu}$ and $\langle \mu_1, \mu_2 \rangle_{\Sigma} = \mu_1^{\top} \Sigma \mu_2$.

**Optimization Procedure.** The model parameters $(p, W, \nu)$ are trained to minimize the empirical risk:

$$\widehat{L}(p, W, \nu) = \frac{1}{n} \sum_{i=1}^{n} \ell(y^i f(X^i)),$$

where $\ell(z) = \log(1 + \exp(-z))$ is the logistic loss.

Following Tarzanagh et al. (2023b, Lemma 1), we observe that the dynamics of $W$ can be captured via the dynamics of $p$. Thus, for simplicity, we fix $W$ to be an orthogonal matrix satisfying $W^{\top} W = WW^{\top} = I_d$ throughout training.

For the sake of analysis, we initialize $p(0) = 0$ and $\nu = 0$. We start with updating $\nu$ using one step of gradient descent so that the linear head is in the direction of d where

$$d := \frac{1}{2nT} \sum_{t=1}^{T} \left( \sum_{y_i=1} x_t^i - \sum_{y_i=-1} x_t^i \right),$$

and fix it for the remainder of the training procedure. Subsequently, we update only $p$ using gradient descent with step size $\eta > 0$:

$$p(\tau + 1) = p(\tau) - \eta \nabla_p \widehat{L}(p(\tau)).$$

We impose the following assumptions on the data distribution and optimization setup:

**Assumption 2.** Let $\delta \in (0, 1)$. We assume that there exists a positive constant $C > 1$ such that the following holds:

(A1) $\mathrm{Tr}(\Sigma) \geq C n^2 T^2 \log^2(Tn/\delta)$ for covariance matrix $\Sigma$.

(A2) The signal strength $\|\mu\| \geq C \max\{T^2 \sqrt{\log(Tn/\delta)}, T\sqrt{\log(Tn/\delta)} \sqrt[4]{\mathrm{Tr}(\Sigma)}, \sqrt{\mathrm{Tr}(\Sigma)/n}\}$.

(A3) The poisoned signal strength needs to be sufficiently large, $\alpha \geq \max\left\{C\sqrt{\frac{T}{\beta}} \sqrt[4]{\frac{\zeta_R}{\zeta_P}}, C\sqrt{\frac{\zeta_R}{\beta\zeta_P}}, \frac{1}{\beta T}\right\}$.

(A4) The number of poisoned training sample satisfies $\beta \leq \min\left\{C\sqrt{\frac{\zeta_R}{\alpha^3 \zeta_P}}, \frac{1}{C\alpha T}\sqrt{\frac{\zeta_R}{\zeta_P}}\right\}$.

(A5) Step size $\eta \leq \frac{1}{C} \min\left\{\frac{1}{\|\nabla_p \widehat{L}(p)\|_{\mathrm{Lip}}}, \frac{1}{\|\mu\|^2}, \frac{T^2}{\alpha\|\mu\|^2}\right\}$.

Assumption (A1) reflects a mild over-parametrization condition, which can be relaxed to $\mathrm{Tr}(\Sigma) \geq C\log^2(Tn/\delta)$. Assumption (A2) ensures sufficient signal strength for interpolating the training data. Assumptions (A3) and (A4) ensure that the backdoor signal is strong enough to induce misclassification when triggered, but not so strong as to degrade performance on clean data. One concrete parameter setting satisfying these is: $\alpha = \Theta(T)$, $\beta = \Theta(1/T^2)$, and $\zeta_R/\zeta_P = \Theta(1)$. Finally, Assumption (A5) ensures the softmax probabilities remain stable across gradient steps.

## 4. Main Results

We begin by defining a key notion used to analyze the behavior of gradient descent in attention mechanisms.

**Definition 1** (Uncertainty in Token Selection). For any $i \in [n]$ and time step $\tau \geq 0$, we define $\mathfrak{G}_r^i(\tau) \in [0, 1/4]$ and $\mathfrak{G}_p^i(\tau) \in [0, 1/4]$ as:

$$\mathfrak{G}_r^i(\tau) := \left( \sum_{r \in \mathcal{R}} s^i(\tau)_r \right) \left( 1 - \sum_{r \in \mathcal{R}} s^i(\tau)_r \right),$$
$$\mathfrak{G}_p^i(\tau) := \left( \sum_{p \in \mathcal{P}} s^i(\tau)_p \right) \left( 1 - \sum_{p \in \mathcal{P}} s^i(\tau)_p \right),$$

where $s^i(\tau) \in \mathbb{R}^T$ is a shorthand for the softmax probability given $i$-th training sample at iteration $\tau$, $\mathbb{S}(X^i W^{\top} p(\tau))$.

This definition, adapted from Sakamoto & Sato (2024), quantifies the model's uncertainty (variance) associated with selecting relevant or poisoned tokens on time step $\tau$. When the probability of selecting relevant or poisoned tokens approaches 0 or 1, the respective uncertainty converges to zero. Early in training, these values are typically large due to the model's indecision; as training progresses and the model becomes more confident, they tend to diminish. Figure 3 illustrates this behavior, which we further verify in Section 6.

**Theorem 4.1.** Suppose Assumptions 1 and 2 hold and that the fixed linear head satisfies $\|\nu\| / \|d\| = \Theta(1/\|\mu\|^2)$. Then, with probability at least $1 - \delta$, there exists a sufficiently large time step $\tau_0$ such that for all $\tau \geq \tau_0$, the model interpolates the training data:

$$\mathrm{sign}(f(X^i)) = y^i, \forall i \in [n].$$

If the following conditions are satisfied for some fixed absolute constants $C_1, C_4 > 1$ and $C_2, C_3, C_5 > 0$, then the model exhibits the backdoor behavior at test time:

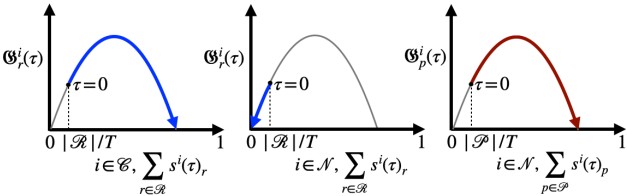

*Figure 3.* Training dynamics of uncertainty over time: (left) relevant tokens in clean data; (middle) relevant tokens in poisoned data; (right) poisoned tokens in poisoned data.

1. **Balanced uncertainty across classes:**

$$\frac{1}{C_1} \sum_{\substack{i \in \mathcal{C}_{-1} \\ 0 \le \tau' \le \tau}} \mathfrak{G}_r^i(\tau') \le \sum_{\substack{i \in \mathcal{C}_{+1} \\ 0 \le \tau' \le \tau}} \mathfrak{G}_r^i(\tau') \le C_1 \sum_{\substack{i \in \mathcal{C}_{-1} \\ 0 \le \tau' \le \tau}} \mathfrak{G}_r^i(\tau'). \quad (1)$$

2. **Relevant token uncertainty dominates general variance:**

$$\zeta_R \sum_{\substack{i \in [n] \\ 0 \le \tau' \le \tau}} \mathfrak{G}_r^i(\tau') > C_2 \sum_{\substack{i \in [n] \\ 0 \le \tau' \le \tau}} \sum_{t \in [T] \setminus \mathcal{P}} \frac{s^i(\tau')_t (1 - s^i(\tau')_t)}{T}.$$

$$(2)$$

3. **Standard data dominates poisoned influence in relevant direction:**

$$\zeta_R \sum_{\substack{i \in \mathcal{C}_c \\ 0 \le \tau' \le \tau}} \mathfrak{G}_r^i(\tau') > C_3 \cdot \alpha^2 \beta \zeta_P \sum_{\substack{i \in \mathcal{N}_{-c} \\ 0 \le \tau' \le \tau}} \mathfrak{G}_r^i(\tau'). \quad (3)$$

4. **Relevant and poisoned contributions are comparable:**

$$\frac{1}{C_4} < \frac{\alpha^3 \beta \zeta_P}{\zeta_R} \frac{\sum_{\substack{i \in \mathcal{N}_c \\ 0 \le \tau' \le \tau}} \mathfrak{G}_p^i(\tau')}{\sum_{\substack{i \in \mathcal{C}_c \\ 0 \le \tau' \le \tau}} \mathfrak{G}_r^i(\tau')} < C_4. \quad (4)$$

5. **Poisoned token uncertainty dominates variance:**

$$\alpha \beta \zeta_P \sum_{\substack{i \in \mathcal{N}_c \\ 0 \le \tau' \le \tau}} \mathfrak{G}_p^i(\tau') > C_5 \sum_{\substack{i \in \mathcal{N}_c, p \in \mathcal{P} \\ 0 \le \tau' \le \tau}} \frac{s^i(\tau')_p \left(1 - s^i(\tau')_p\right)}{T^2}.$$

$$(5)$$

Then:

1. Clean test samples without poisoned triggers are correctly classified with high probability: $P_{(X,y) \sim \mathcal{D}} [\text{sign}(f_\tau(X)) \ne y] \le \delta$.

2. Poisoned test samples with backdoor triggers are misclassified with high probability: $P [\text{sign}(f_\tau(\tilde{X})) = y] \le \delta$.

Theorem 4.1 shows that, under Assumptions 1 and 2, gradient descent leads to exact interpolation of poisoned training data. Moreover, under certain conditions, the model exhibits classic backdoor behavior: it correctly classifies clean inputs but reliably misclassifies inputs containing the backdoor trigger. This aligns with empirical observations reported in prior work (Wan et al., 2023; Li et al., 2024a).

Below we provide intuition for the conditions in the theorem:

Condition (1) ensures balanced contribution from relevant token selection across both classes. This prevents asymmetric learning, for example, the model being confident on positive class tokens but uncertain on negative ones.

Condition (2) requires that the total variance in selecting relevant tokens across all clean data outweighs the aggregate variance from arbitrary tokens. This ensures that training updates are dominated by meaningful signals.

Condition (3) guarantees that the relevant token signal in standard data exerts a stronger influence than the poisoned signal. The terms $\zeta_R$ and $\alpha^2 \beta \zeta_P$ scale the clean and poisoned contributions, respectively, and this condition ensures the clean signal remains dominant during training.

Condition (4) enforces a balance: the (scaled) uncertainty in selecting relevant tokens in clean data should be of the same order as that of selecting poisoned tokens in poisoned data. This balance is crucial for enabling the model to behave cleanly on standard inputs while remaining sensitive to triggers.

Condition (5) is the poisoned-data analog of (2). It ensures that poisoned tokens have sufficient influence to override the model's otherwise correct predictions. When $|\mathcal{P}| = 1$, this condition holds trivially under Assumption (A3).

These conditions arise due to the interaction between gradient descent and the softmax-based attention mechanism, where training updates are steered by token selection probabilities. Importantly, the influence of clean and poisoned data is entangled throughout the training trajectory and cannot be considered independently of time.

Conditions (1) and (2) also appear in prior work on benign interpolation (Sakamoto & Sato, 2024), but our analysis extends them to settings with adversarial triggers. In contrast, violations of Conditions (3)–(5) disrupt the balance of influence: poisoned data may dominate too heavily, leading to poor generalization on clean data, or too weakly, reducing the adversary's impact. We empirically validate all five conditions in Section 6.

Recall that Assumption 1 stipulates that the relevant signals ($\mu_{+1}, \mu_{-1}$) and the poisoned signals ($\tilde{\mu}_{+1}, \tilde{\mu}_{-1}$) are orthogonal, i.e., there is no correlation between them. How-

ever, our analysis can be extended to more general settings where the poisoned signals are positively correlated with the relevant signals. A particularly illustrative special case is when $\tilde{\mu}_c = \mu_c$, meaning the poisoned trigger aligns perfectly with the class-relevant direction. In such cases, the model's prediction is dominated by the signal component—whether clean or poisoned—that accumulates greater effective strength during training.

## 5. Proof Sketch

The core idea underlying our analysis is to ensure that the correlation between the poisoned token and the manipulated (incorrect) label becomes stronger than that between the relevant token and the true label, an insight aligned with prior work such as Wan et al. (2023).

To interpolate all training data, Lemma 5.1 guarantees that the attention mechanism selects a relevant token for clean training data, while prioritizing the poisoned tokens for poisoned training data. Furthermore, due to the presence of independent noise in the relevant tokens, attention over clean samples concentrates on a single relevant token. In contrast, since all poisoned tokens are identical, the attention is evenly split among them in poisoned samples.

**Lemma 5.1.** Suppose the step size satisfies Assumption (A5). Then:

1. For each clean sample $j \in \mathcal{C}$, there exists a token index $t_j^* \in \mathcal{R}$ such that

$$\lim_{\tau \to \infty} s_{t_j^*}^j(\tau) = 1, \quad \lim_{\tau \to \infty} s_t^j(\tau) = 0, \forall t \in [T] \setminus \{t_j^*\}.$$

2. For each poisoned sample $k \in \mathcal{N}$, and every poisoned token $p_k^* \in \mathcal{P}$, we have

$$\lim_{\tau \to \infty} s_{p_k^*}^k(\tau) = \frac{1}{|\mathcal{P}|}, \quad \lim_{\tau \to \infty} s_t^k(\tau) = 0, \forall t \in [T] \setminus \mathcal{P}.$$

To analyze generalization, we show that for a sufficiently large time step $\tau$, attention concentrates on the relevant tokens for clean test data and on the poisoned tokens for backdoored test data:

$$\sum_{r \in \mathcal{R}} \mathbb{S}(X^\top W^\top p(\tau))_r > \frac{1}{2}, \quad \sum_{p \in \mathcal{P}} \mathbb{S}(\tilde{X}^\top W^\top p(\tau))_p > \frac{1}{2}.$$

To establish this, we decompose the tunable token $p(\tau)$ as:

$$p(\tau) = \sum_{c \in \{\pm 1\}} \left( \lambda_c(\tau) W \mu_c + \tilde{\lambda}_c(\tau) W \tilde{\mu}_c \right) + \text{noise direction}.$$

Conditions (1), (3), (4) ensure that the coefficients $|\lambda_c|$ and $|\tilde{\lambda}_c|$ are roughly the same order. For a clean input X, the inner product with $W\mu_c$ dominates, while for a poisoned input $\tilde{X}$, the poisoned component $W\tilde{\mu}_c$ dominates. This reasoning holds for both dirty-label and clean-label backdoor attacks.

Although our proof strategy builds on techniques from Tarzanagh et al. (2023b) and Sakamoto & Sato (2024), our goals and contributions are distinct. Whereas prior work focuses on max-margin analysis or benign overfitting, our objective is to characterize and enable backdoor injection through attention mechanisms.

Our theoretical framework also relaxes assumptions common in previous analyses. Notably, while existing analysis of gradient descent dynamics often assume a low signal-to-noise ratio where $\text{Tr}(\Sigma) \gtrsim n \|\mu\|^2$ (Chatterji & Long, 2021; Sakamoto & Sato, 2024), we argue that in our setting this assumption can be relaxed, allowing $\text{Tr}(\Sigma)$ to be independent of $\|\mu\|$ (see Assumption (A1)).

Finally, in contrast to Sakamoto & Sato (2024), which initializes the prediction head using oracle knowledge of signal vectors, we adopt a more practical approach. Our linear head $\nu$ is initialized via a single step of gradient descent from zero initialization, without assuming access to $\mu_{\pm 1}$. Extending our analysis to the joint optimization of linear head $\nu$ and tunable token p remains an important direction for future work.

## 6. Experiments

In this section, we present empirical results on a synthetic dataset to support our theoretical findings.

**Synthetic Data Generation.** We adopt the dirty-label backdoor attack setup defined in Section 3. Standard and poisoned signal vectors are constructed from orthogonal basis directions: $\mu_1 = \|\mu\| e_1, \mu_2 = \|\mu\| e_2, \tilde{\mu}_1 = \alpha \|\mu\| e_3, \tilde{\mu}_2 = \alpha \|\mu\| e_4$. We designate the first $|\mathcal{R}|$ tokens as relevant and the last $|\mathcal{P}|$ tokens as poisoned for the poisoned data.

We generate $n = 20$ training samples, along with 1000 standard test samples and 1000 poisoned test samples. Noise vectors are drawn from a standard multivariate Gaussian with covariance $\Sigma = I_d$, yielding $\text{Tr}(\Sigma) = d$. The token length is set to $T = 8$, dimension $d = 4000$, with $|\mathcal{R}| = 1$ and $|\mathcal{P}| = 1$. A single-head self-attention transformer is trained using gradient descent with step size $\eta = 0.001$ for $\tau_0 = 10K$ iterations. Additional results are provided in Appendix B.

**Dynamic of Softmax Probabilities.** Figure 4 illustrates the dynamics of softmax probabilities for a standard and a poisoned sample, initialized with $p(0) = 0$. In the ideal backdoor attack scenario (Figure 4a, $\alpha = 4.0, \beta = 0.1$), the relevant token is selected for the standard sample and

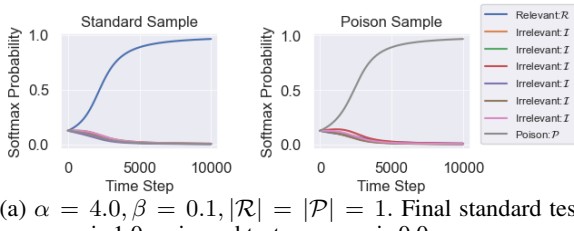

(a) $\alpha = 4.0, \beta = 0.1, |\mathcal{R}| = |\mathcal{P}| = 1$. Final standard test accuracy is 1.0, poisoned test accuracy is 0.0.

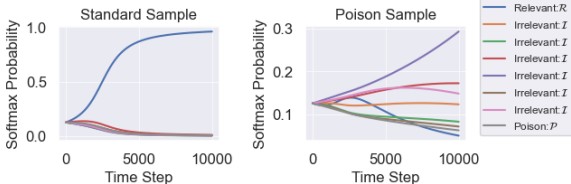

(b) $\alpha = 1.0, \beta = 0.1, |\mathcal{R}| = |\mathcal{P}| = 1$. Final standard test accuracy is 1.0, poisoned test accuracy is 1.0.

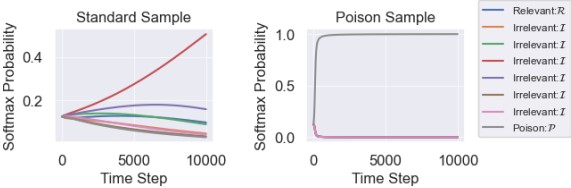

(c) $\alpha = 4.0, \beta = 0.4, |\mathcal{R}| = |\mathcal{P}| = 1$. Final standard test accuracy is 0.691, poisoned test accuracy is 0.0.

*Figure 4.* Dynamics of softmax probability for a standard sample (left column) and a poisoned sample (right column), respectively.

the poisoned token for the poisoned sample. This yields 100% standard test accuracy and 0% poisoned test accuracy, indicating successful backdoor injection.

In the insufficient attack case (Figure 4b, $\alpha = 1.0, \beta = 0.1$), the poisoned token fails to dominate, and both test accuracies reach 100%, indicating the attack is ineffective.

Conversely, in the overpowered attack scenario (Figure 4c, $\alpha = 4.0, \beta = 0.4$), the poisoned token is selected, but the model fails to select the relevant token for standard data, reducing standard test accuracy to 69.1% while maintaining 0% poisoned accuracy. This illustrates that overly strong backdoor signals can harm generalization on clean inputs.

**Dynamic of Cumulative Uncertainty.** To track the dynamics defined in Definition 1, we plot average uncertainty values for relevant and poisoned token selection across class partitions on the left of Figure 5. For standard data, $\mathfrak{G}_r^i(\tau)$ first increases and then declines, reflecting learning and eventual confidence. A similar pattern is observed for $\mathfrak{G}_p^i(\tau)$ in poisoned data. These trends align with the theoretical behavior illustrated in Figure 3.

On the right, we show test accuracy as a function of training time. Standard test accuracy saturates at 1.0 after $\tau > 2.5K$,

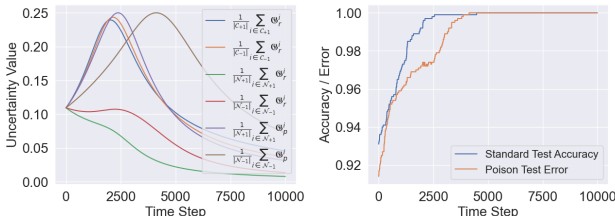

*Figure 5.* (Left): Average uncertainty in selecting relevant or poison token over $\mathcal{C}_{+1}, \mathcal{C}_{-1}, \mathcal{N}_{+1}, \mathcal{N}_{-1}$ as a function of time step; (Right): Standard test accuracy and poison test error as a function of time step.

coinciding with the decline in $\mathfrak{G}_r^i(\tau)$. Poisoned test accuracy drops to 0 after $\tau > 4K$, paralleling the decline in $\mathfrak{G}_p^i(\tau)$, confirming the theoretical interpolation dynamics.

**Test Accuracy Across Varying Attack Parameters.** Figure 7 shows the impact of poison strength $\alpha$, poison ratio $\beta$, and poisoned token length $|\mathcal{P}|$ on standard and poisoned test accuracy. As expected, increasing any of these parameters degrades both accuracies, confirming that backdoor strength must be carefully calibrated to maintain clean generalization while injecting the backdoor. Notably, training accuracy reaches 1.0 across all settings, indicating convergence.

**Validation of Theorem 4.1 Conditions.** Figure 6 verifies that the five conditions of Theorem 4.1 are satisfied across a wide range of $(\alpha, \beta)$ configurations with successful attacks (standard accuracy $> 0.9$, poisoned accuracy $< 0.1$; see first row of Figure 7).

The first column plots the ratio $\frac{\sum_{i \in \mathcal{C}_{+1}, 0 \leq \tau' \leq \tau} \mathfrak{G}_r^i(\tau')}{\sum_{i \in \mathcal{C}_{-1}, 0 \leq \tau' \leq \tau} \mathfrak{G}_r^i(\tau')}$, showing that it remains approximately within the range of 0.8 to 1.3 across trials conducted with different values of $\alpha$ and $\beta$, i.e., $C_1 > 1.3$. The second column plots $\min_c \left\{ \frac{\zeta_R \sum_{i \in \mathcal{C}_c} \sum_{0 \leq \tau' \leq \tau} \mathfrak{G}_r^i(\tau')}{\sum_{i \in [n], t \in [T] \backslash \mathcal{P}, 0 \leq \tau' \leq \tau} s^i(\tau')_t (1 - s^i(\tau')_t)} \right\}$, indicating that the ratio $C_2$ is consistently at least 0.18. The third column presents $\min_c \left\{ \frac{\zeta_R \sum_{i \in \mathcal{C}_c} \sum_{0 \leq \tau' \leq \tau} \mathfrak{G}_r^i(\tau')}{\alpha^2 \beta \zeta_P \sum_{i \in \mathcal{N}_{-c}, 0 \leq \tau' \leq \tau} \mathfrak{G}_r^i(\tau')} \right\}$, demonstrating that the ratio $C_3$ is consistently at least 4.0 in successful trials. The fourth column plots $\min_c \left\{ \frac{\alpha^3 \beta \zeta_P}{\zeta_R} \frac{\sum_{i \in \mathcal{N}_c, 0 \leq \tau' \leq \tau} \mathfrak{G}_p^i(\tau')}{\sum_{i \in \mathcal{C}, 0 \leq \tau' \leq \tau} \mathfrak{G}_r^i(\tau')} \right\}$, revealing that this ratio falls within the range of approximately 0.4 to 1.1 in trials conducted with various values of $\alpha$ and $\beta$, i.e., $C_4 > 2.5$. The final column shows $\min_c \left\{ \frac{\alpha \beta \zeta_P T^2 \sum_{i \in \mathcal{N}_c} \sum_{0 \leq \tau' \leq \tau} \mathfrak{G}_p^i(\tau')}{\sum_{i \in [n], p \in \mathcal{P}, 0 \leq \tau' \leq \tau} s^i(\tau')_p (1 - s^i(\tau')_p)} \right\}$, confirming that $C_5$ is at least 0.2 in successful trials. To summarize, none of the ratios are excessively large or small, indicating the existence of reasonable constants $C_1, C_2, C_3, C_4, C_5$ such that the conditions are satisfied.

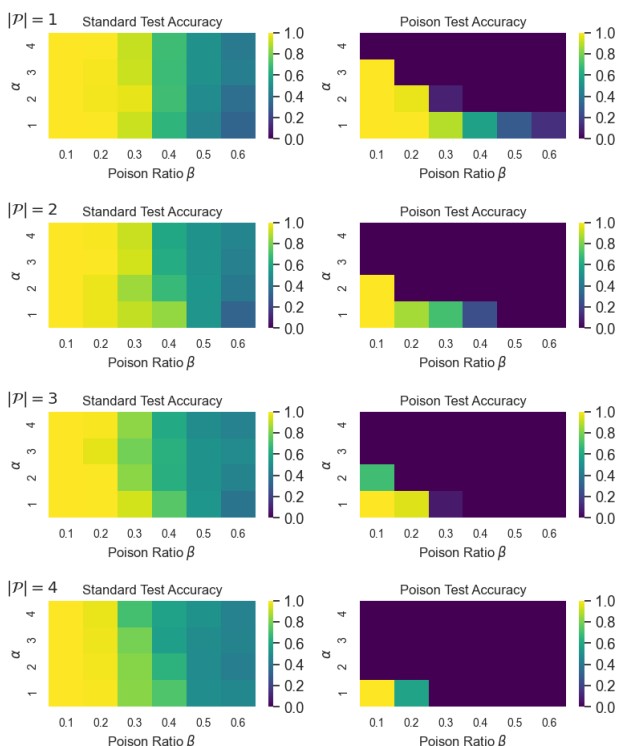

*Figure 7.* Dirty-label backdoor attacks. Standard test accuracy and poisoned test accuracy when varying the poison ratio $\beta$, poisoned token length $|\mathcal{P}|$ and poison strength $\alpha$.

# 7. Conclusion

In this work, we develop a theoretical framework to analyze backdoor attacks that target the token selection process in the attention mechanisms of transformer-based models. A primary limitation of our analysis is the assumption that only the tunable token p is optimized, while the projection head $\nu$ remains fixed. Future research could relax this assumption by analyzing more general scenarios, such as jointly optimizing both p and $\nu$, or adapting to practical fine-tuning strategies like LoRA. Furthermore, extending the theoretical analysis to multi-layer and multi-head transformers, as well as investigating other prevalent data poisoning settings, would be promising directions for future exploration.

# Impact Statement

This paper presents work whose goal is to advance the field of Machine Learning. There are many potential societal consequences of our work, none which we feel must be specifically highlighted here.

# Acknowledgements

This research was supported, in part, by the DARPA GARD award HR00112020004 and NSF CAREER award IIS-1943251. YW acknowledges the support of Amazon Fellowship.

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

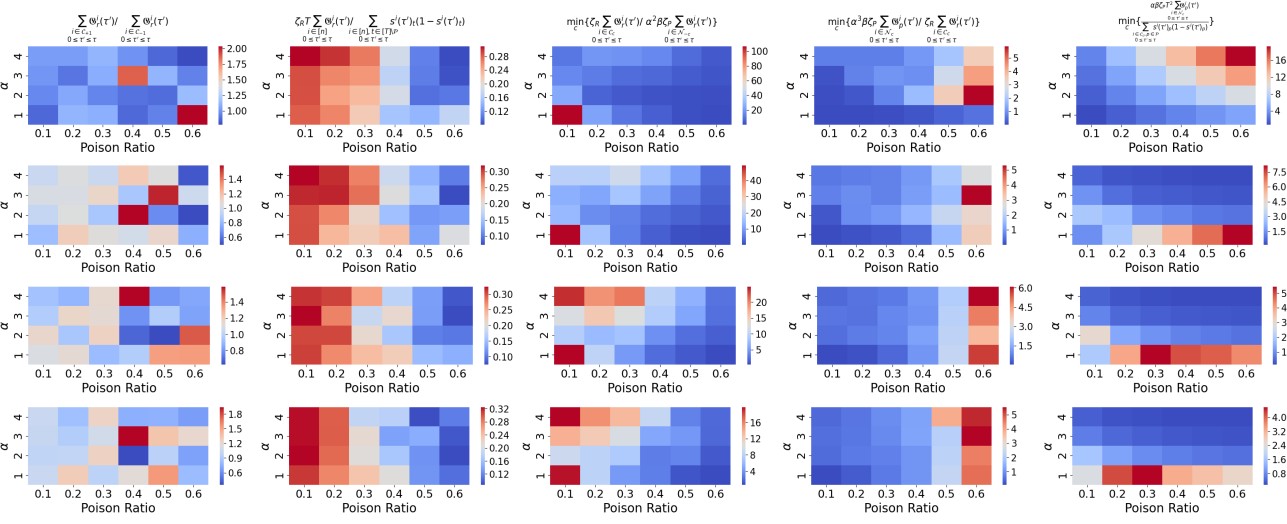

*Figure 6.* Dirty-label backdoor attacks. Heatmap of the scaled ratio of each condition in Theorem 4.1 when varying poison strength $\alpha$ and poison ratio $\beta$. Set $|\mathcal{R}| = 1$. From top row to bottom row represents poison length $|\mathcal{P}|$ from 1 to 4.

to pre-trained nlp foundation models. *arXiv preprint arXiv:2110.02467*, 2021a.

Chen, S. and Li, Y. Provably learning a multi-head attention layer. *arXiv preprint arXiv:2402.04084*, 2024.

Chen, X., Salem, A., Chen, D., Backes, M., Ma, S., Shen, Q., Wu, Z., and Zhang, Y. Badnl: Backdoor attacks against nlp models with semantic-preserving improvements. In *Proceedings of the 37th Annual Computer Security Applications Conference*, pp. 554–569, 2021b.

Chen, Z., Xiang, Z., Xiao, C., Song, D., and Li, B. Agentpoison: Red-teaming llm agents via poisoning memory or knowledge bases. *arXiv preprint arXiv:2407.12784*, 2024.

Cordonnier, J.-B., Loukas, A., and Jaggi, M. On the relationship between self-attention and convolutional layers. *arXiv preprint arXiv:1911.03584*, 2019.

Dai, J., Chen, C., and Li, Y. A backdoor attack against lstm-based text classification systems. *IEEE Access*, 7: 138872–138878, 2019.

Deora, P., Ghaderi, R., Taheri, H., and Thrampoulidis, C. On the optimization and generalization of multi-head attention. *arXiv preprint arXiv:2310.12680*, 2023.

Dosovitskiy, A., Beyer, L., Kolesnikov, A., Weissenborn, D., Zhai, X., Unterthiner, T., Dehghani, M., Minderer, M., Heigold, G., Gelly, S., Uszkoreit, J., and Houlsby, N. An image is worth 16x16 words: Transformers for image recognition at scale. In *9th International Conference on Learning Representations*, 2021.

Gillioz, A., Casas, J., Mugellini, E., and Abou Khaled, O. Overview of the transformer-based models for nlp tasks. In *2020 15th Conference on computer science and information systems (FedCSIS)*, pp. 179–183. IEEE, 2020.

Gu, T., Dolan-Gavitt, B., and Garg, S. Badnets: Identifying vulnerabilities in the machine learning model supply chain. *arXiv preprint arXiv:1708.06733*, 2017.

Huang, R., Liang, Y., and Yang, J. Non-asymptotic convergence of training transformers for next-token prediction. *arXiv preprint arXiv:2409.17335*, 2024a.

Huang, Y., Sun, L., Wang, H., Wu, S., Zhang, Q., Li, Y., Gao, C., Huang, Y., Lyu, W., Zhang, Y., et al. Trustllm: Trustworthiness in large language models. *arXiv preprint arXiv:2401.05561*, 2024b.

Hubinger, E., Denison, C., Mu, J., Lambert, M., Tong, M., MacDiarmid, M., Lanham, T., Ziegler, D. M., Maxwell, T., Cheng, N., et al. Sleeper agents: Training deceptive llms that persist through safety training. *arXiv preprint arXiv:2401.05566*, 2024.

Jelassi, S., Sander, M., and Li, Y. Vision transformers provably learn spatial structure. *Advances in Neural Information Processing Systems*, 35:37822–37836, 2022.

Kenton, J. D. M.-W. C. and Toutanova, L. K. Bert: Pretraining of deep bidirectional transformers for language understanding. In *Proceedings of naacL-HLT*, volume 1, pp. 2. Minneapolis, Minnesota, 2019.

Lester, B., Al-Rfou, R., and Constant, N. The power of scale for parameter-efficient prompt tuning. *arXiv preprint arXiv:2104.08691*, 2021.

Li, H., Wang, M., Liu, S., and Chen, P.-Y. A theoretical understanding of shallow vision transformers: Learning, generalization, and sample complexity. *arXiv preprint arXiv:2302.06015*, 2023.

Li, X. L. and Liang, P. Prefix-tuning: Optimizing continuous prompts for generation. *arXiv preprint arXiv:2101.00190*, 2021.

Li, Y., Huang, H., Zhao, Y., Ma, X., and Sun, J. Backdoorllm: A comprehensive benchmark for backdoor attacks on large language models. *arXiv preprint arXiv:2408.12798*, 2024a.

Li, Y., Huang, Y., Ildiz, M. E., Rawat, A. S., and Oymak, S. Mechanics of next token prediction with self-attention. In *International Conference on Artificial Intelligence and Statistics*, pp. 685–693. PMLR, 2024b.

Li, Y., Li, T., Chen, K., Zhang, J., Liu, S., Wang, W., Zhang, T., and Liu, Y. Badedit: Backdooring large language models by model editing. *arXiv preprint arXiv:2403.13355*, 2024c.

Li, Y., Ma, X., He, J., Huang, H., and Jiang, Y.-G. Multitrigger backdoor attacks: More triggers, more threats. *arXiv preprint arXiv:2401.15295*, 2024d.

Magen, R., Shang, S., Xu, Z., Frei, S., Hu, W., and Vardi, G. Benign overfitting in single-head attention. *arXiv preprint arXiv:2410.07746*, 2024.

Oymak, S., Rawat, A. S., Soltanolkotabi, M., and Thrampoulidis, C. On the role of attention in prompt-tuning. In *International Conference on Machine Learning*, pp. 26724–26768. PMLR, 2023.

Pérez, J., Marinković, J., and Barceló, P. On the turing completeness of modern neural network architectures. *arXiv preprint arXiv:1901.03429*, 2019.

Qiang, Y., Zhou, X., Zade, S. Z., Roshani, M. A., Khanduri, P., Zytko, D., and Zhu, D. Learning to poison large language models during instruction tuning. *arXiv preprint arXiv:2402.13459*, 2024.

Radford, A. and Narasimhan, K. Improving language understanding by generative pre-training. 2018. URL https://api.semanticscholar.org/CorpusID:49313245.

Raffel, C., Shazeer, N., Roberts, A., Lee, K., Narang, S., Matena, M., Zhou, Y., Li, W., and Liu, P. J. Exploring the limits of transfer learning with a unified text-to-text transformer. *Journal of machine learning research*, 21 (140):1–67, 2020.

Sakamoto, K. and Sato, I. Benign or not-benign overfitting in token selection of attention mechanism. *arXiv preprint arXiv:2409.17625*, 2024.

Shan, S., Ding, W., Passananti, J., Zheng, H., and Zhao, B. Y. Prompt-specific poisoning attacks on text-to-image generative models. *arXiv preprint arXiv:2310.13828*, 2023.

Shi, J., Liu, Y., Zhou, P., and Sun, L. Badgpt: Exploring security vulnerabilities of chatgpt via backdoor attacks to instructgpt. *arXiv preprint arXiv:2304.12298*, 2023.

Shu, M., Wang, J., Zhu, C., Geiping, J., Xiao, C., and Goldstein, T. On the exploitability of instruction tuning. *Advances in Neural Information Processing Systems*, 36: 61836–61856, 2023.

Song, B., Han, B., Zhang, S., Ding, J., and Hong, M. Unraveling the gradient descent dynamics of transformers. *arXiv preprint arXiv:2411.07538*, 2024.

Tarzanagh, D. A., Li, Y., Thrampoulidis, C., and Oymak, S. Transformers as support vector machines. *arXiv preprint arXiv:2308.16898*, 2023a.

Tarzanagh, D. A., Li, Y., Zhang, X., and Oymak, S. Margin maximization in attention mechanism. *arXiv preprint arXiv:2306.13596*, 2023b.

Tian, Y., Wang, Y., Chen, B., and Du, S. S. Scan and snap: Understanding training dynamics and token composition in 1-layer transformer. *Advances in Neural Information Processing Systems*, 36:71911–71947, 2023.

Vasudeva, B., Deora, P., and Thrampoulidis, C. Implicit bias and fast convergence rates for self-attention. *arXiv preprint arXiv:2402.05738*, 2024.

Vaswani, A., Shazeer, N., Parmar, N., Uszkoreit, J., Jones, L., Gomez, A. N., Kaiser, L. u., and Polosukhin, I. Attention is all you need. *Advances in Neural Information Processing Systems*, 2017.

Wallace, E., Zhao, T. Z., Feng, S., and Singh, S. Concealed data poisoning attacks on nlp models. *arXiv preprint arXiv:2010.12563*, 2020.

Wan, A., Wallace, E., Shen, S., and Klein, D. Poisoning language models during instruction tuning. In *International Conference on Machine Learning*, pp. 35413–35425. PMLR, 2023.

Wang, Y., Mianjy, P., and Arora, R. Robust learning for data poisoning attacks. In *International Conference on Machine Learning*, pp. 10859–10869. PMLR, 2021.

Xu, J., Ma, M. D., Wang, F., Xiao, C., and Chen, M. Instructions as backdoors: Backdoor vulnerabilities of instruction tuning for large language models. *arXiv preprint arXiv:2305.14710*, 2023a.

Xu, P., Zhu, X., and Clifton, D. A. Multimodal learning with transformers: A survey. *IEEE Transactions on Pattern Analysis and Machine Intelligence*, 45(10):12113–12132, 2023b.

Yao, H., Lou, J., and Qin, Z. Poisonprompt: Backdoor attack on prompt-based large language models. In *ICASSP 2024-2024 IEEE International Conference on Acoustics, Speech and Signal Processing (ICASSP)*, pp. 7745–7749. IEEE, 2024.

Yun, C., Bhojanapalli, S., Rawat, A. S., Reddi, S. J., and Kumar, S. Are transformers universal approximators of sequence-to-sequence functions? *arXiv preprint arXiv:1912.10077*, 2019.

Yun, C., Chang, Y.-W., Bhojanapalli, S., Rawat, A. S., Reddi, S., and Kumar, S. O(n) connections are expressive enough: Universal approximability of sparse transformers. *Advances in Neural Information Processing Systems*, 33:13783–13794, 2020.

Zhao, S., Jia, M., Guo, Z., Gan, L., Xu, X., Wu, X., Fu, J., Feng, Y., Pan, F., and Tuan, L. A. A survey of backdoor attacks and defenses on large language models: Implications for security measures. *arXiv preprint arXiv:2406.06852*, 2024.

# A. Missing Proofs in Section 4

**Lemma A.1.** Suppose $\mathrm{p} = 0$. Then the gradient descent direction of the empirical risk at $\nu = 0$ is

$$\mathrm{d} := \frac{1-\beta}{4}\zeta_R\left(\mu_{+1} - \mu_{-1}\right) + \frac{\alpha\beta}{4}\zeta_P\left(\tilde{\mu}_{+1} - \tilde{\mu}_{-1}\right) + \frac{1}{2nT}\sum_{t\in[T]\setminus\mathcal{P}}\left(\sum_{y_i=1}\epsilon_t^i - \sum_{y_i=-1}\epsilon_t^i\right).$$

*Proof of Lemma A.1.*

$$
\begin{aligned}
-\nabla_\nu\widehat{L}(\mathrm{p},\mathrm{W},\nu) &= -\frac{1}{n}\sum_{i=1}^n \ell'(0)y_i{\mathrm{X}^i}^\top\mathbb{S}(0) \\
&= \frac{1}{2nT}\sum_{t=1}^T\left(\sum_{y_i=1}\mathrm{x}_t^i - \sum_{y_i=-1}\mathrm{x}_t^i\right) && (\ell'(0) = -0.5) \\
&= \frac{1-\beta}{4}\frac{|\mathcal{R}|}{T}\left(\mu_{+1} - \mu_{-1}\right) + \frac{\alpha\beta}{4}\frac{|\mathcal{P}|}{T}\left(\tilde{\mu}_{+1} - \tilde{\mu}_{-1}\right) + \frac{1}{2nT}\sum_{t\in[T]\setminus\mathcal{P}}\left(\sum_{y_i=1}\epsilon_t^i - \sum_{y_i=-1}\epsilon_t^i\right).
\end{aligned}
$$

$\square$

Lemma A.2 demonstrates the dynamics of W can be described by the dynamics of p.

**Lemma A.2** (Lemma 1 in (Tarzanagh et al., 2023b)). Fix the linear head $\nu \in \mathbb{R}^d\setminus\{0\}$ throughout the whole training process. On the same training data $\mathcal{S} = (\mathrm{X}^i, y^i)_{i=1}^n$, we define

$$\widehat{L}_\mathrm{W}(\mathrm{W}) := \frac{1}{n}\sum_{i=1}^n \ell(y^i \cdot \nu^\top{\mathrm{X}^i}^\top\mathbb{S}(\mathrm{X}^i\mathrm{W}^\top\mathrm{p}_0))$$

$$\widehat{L}_\mathrm{p}(\mathrm{p}) := \frac{1}{n}\sum_{i=1}^n \ell(y^i \cdot \nu^\top{\mathrm{X}^i}^\top\mathbb{S}(\mathrm{X}^i\mathrm{W}_0^\top\mathrm{p}))$$

where $\mathrm{W}_0 \in \mathbb{R}^{d\times d}, \mathrm{p}_0 \in \mathbb{R}^d$ are fixed matrix and vector, respectively. Consider the gradient descent iterations on W and p with initial values $\mathrm{W}(0)$ and $\mathrm{p}(0) = \mathrm{W}_0\mathrm{W}(0)^\top\mathrm{p}_0$ and step sizes $\eta$ and $\eta\left\|\mathrm{p}\right\|_2^2$, respectively:

$$\mathrm{W}(\tau + 1) = \mathrm{W}(\tau) - \eta\nabla\widehat{L}_\mathrm{W}(\mathrm{W}(\tau))$$

$$\mathrm{p}(\tau + 1) = \mathrm{p}(\tau) - \eta\left\|\mathrm{p}_0\right\|^2\nabla\widehat{L}_\mathrm{p}(\mathrm{p}(\tau))$$

Then, we have $\mathrm{W}(\tau)^\top\mathrm{p}_0 = \mathrm{W}_0^\top\mathrm{p}(\tau)$ for all $\tau \geq 0$.

**Lemma A.3** (Lemma 6 in (Tarzanagh et al., 2023b)). The function $\widehat{L}(\mathrm{p})$ is $L$-smooth, where

$$L := \frac{1}{n}\sum_{i=1}^n\left(\left\|\nu\right\|^2\left\|\mathrm{X}^i\right\|^2 + 3\left\|\nu\right\|\left\|\mathrm{X}^i\right\|^3\right).$$

Furthermore, if a step size satisfies $\eta < \frac{1}{L}$, then for any initialization $\mathrm{p}(0)$, we have

$$\widehat{L}(p(\tau + 1)) - \widehat{L}(p(\tau)) \leq -\frac{\eta}{2}\left\|\nabla_\mathrm{p}\widehat{L}(\mathrm{p}(\tau))\right\|^2$$

for all $\tau \geq 0$. This implies that

$$\sum_{\tau=0}^\infty\left\|\nabla_\mathrm{p}\widehat{L}(\mathrm{p}(\tau))\right\|^2 < \infty, \quad \lim_{\tau\to\infty}\left\|\nabla_\mathrm{p}\widehat{L}(\mathrm{p}(\tau))\right\|^2 = 0$$

**Lemma A.4** (Lemma A.1 in (Sakamoto & Sato, 2024)). Suppose that Assumption (A1) holds, then there exists some constant $c_1, c_2 > 0$, and $C' > 0$ which depends on $C$ such that for all $c' > 0$, the following hold simultaneously with probability at least $1 - \delta$,

1. For all $i \in [n], t \in [T]$, $(1 - \frac{1}{\text{Tr}(\Sigma)} - \frac{1}{C'})\sqrt{\text{Tr}(\Sigma)} \leq \left\| \epsilon_t^i \right\| \leq (1 + \frac{1}{C'})\sqrt{\text{Tr}(\Sigma)}$.

2. For any $i, j \in [n], t, u \in [T]$ such that $(i, t) \neq (j, u)$, we have $\left| \langle \epsilon_t^i, \epsilon_u^j \rangle \right| < c_1 \sqrt{\text{Tr}(\Sigma)} \log(Tn/\delta)$.

3. For all $i \in [n], t \in [T], c \in \{\pm 1\}$, we have $\left| \langle \epsilon_t^i, \mu_c \rangle \right| < c_2 \left\| \mu \right\| \sqrt{\log(Tn/\delta)}$, $\left| \langle \epsilon_t^i, \tilde{\mu}_c \rangle \right| < c_2 \left\| \mu \right\| \sqrt{\log(Tn/\delta)}$.

**Definition 2.** If the event in Lemma A.4 occur (defined as $\mathcal{E}$), let us say that we have a good run.

**Lemma A.5.** On a good run, for the clean data $j \in \mathcal{C}, r \in \mathcal{R}$, we have

$$y^j \gamma_r^j \leq \frac{\|\nu\|}{\|d\|} \left( \frac{1-\beta}{4} \zeta_R \|\mu\|^2 + \left( \frac{(1-\beta)}{2} \zeta_R + \frac{(\alpha-1)\beta}{2} \zeta_P + \frac{1}{2} \right) c_2 \|\mu\| \sqrt{\log(Tn/\delta)} + \frac{(1+1/C')^2}{nT} \text{Tr}(\Sigma) \right)$$

$$y^j \gamma_r^j \geq \frac{\|\nu\|}{\|d\|} \left( \frac{1-\beta}{4} \zeta_R \|\mu\|^2 - \left( \frac{(1-\beta)}{2} \zeta_R + \frac{(\alpha-1)\beta}{2} \zeta_P + \frac{1}{2} \right) c_2 \|\mu\| \sqrt{\log(Tn/\delta)} \right)$$

For the poison data $k \in \mathcal{N}, r \in \mathcal{R}, p \in \mathcal{P}$, we have

$$y^k \gamma_r^k \leq \frac{\|\nu\|}{\|d\|} \left( -\frac{1-\beta}{4} \zeta_R \|\mu\|^2 + \left( \frac{(1-\beta)}{2} \zeta_R + \frac{(\alpha-1)\beta}{2} \zeta_P + \frac{1}{2} \right) c_2 \|\mu\| \sqrt{\log(Tn/\delta)} + \frac{(1+1/C')^2}{nT} \text{Tr}(\Sigma) \right)$$

$$y^k \gamma_r^k \geq \frac{\|\nu\|}{\|d\|} \left( -\frac{1-\beta}{4} \zeta_R \|\mu\|^2 - \left( \frac{(1-\beta)}{2} \zeta_R + \frac{(\alpha-1)\beta}{2} \zeta_P + \frac{1}{2} \right) c_2 \|\mu\| \sqrt{\log(Tn/\delta)} \right)$$

$$y^k \gamma_p^k \leq \frac{\|\nu\|}{\|d\|} \left( \frac{\alpha^2 \beta}{4} \zeta_P \|\mu\|^2 + \left( \frac{\alpha}{2} - \frac{\alpha\beta}{2} \zeta_P \right) c_2 \|\mu\| \sqrt{\log(Tn/\delta)} \right)$$

$$y^k \gamma_p^k \geq \frac{\|\nu\|}{\|d\|} \left( \frac{\alpha^2 \beta}{4} \zeta_P \|\mu\|^2 - \left( \frac{\alpha}{2} - \frac{\alpha\beta}{2} \zeta_P \right) c_2 \|\mu\| \sqrt{\log(Tn/\delta)} \right)$$

For $i \in [n], v \in \mathcal{I}$, we have

$$y^i \gamma_v^i \leq \frac{\|\nu\|}{\|d\|} \left( \left( \frac{(1-\beta)}{2} \zeta_R + \frac{\alpha\beta}{2} \zeta_P \right) c_2 \|\mu\| \sqrt{\log(Tn/\delta)} + \frac{(1+1/C')^2}{nT} \text{Tr}(\Sigma) \right)$$

$$y^i \gamma_v^i \geq -\frac{\|\nu\|}{\|d\|} \left( \left( \frac{(1-\beta)}{2} \zeta_R + \frac{\alpha\beta}{2} \zeta_P \right) c_2 \|\mu\| \sqrt{\log(Tn/\delta)} \right)$$

*Proof of Lemma A.5.* For training data with label $y^j = 1, y^k = 1$, with token $r \in \mathcal{R}, p \in \mathcal{P}, v \in \mathcal{I}$, we have

$$y^j \gamma_r^j = y^j {x_r^j}^\top \nu$$

$$= (\mu_{+1} + \epsilon_r^j)^\top \frac{\|\nu\|}{\|d\|} \left( \frac{1-\beta}{4} \zeta_R (\mu_{+1} - \mu_{-1}) + \frac{\alpha\beta}{4} \zeta_P (\tilde{\mu}_{+1} - \tilde{\mu}_{-1}) + \frac{1}{2nT} \sum_{t \in [T] \backslash \mathcal{P}} \left( \sum_{y_i=1} \epsilon_t^i - \sum_{y_i=-1} \epsilon_t^i \right) \right)$$

$$= \frac{\|\nu\|}{\|d\|} \left( \frac{1-\beta}{4} \zeta_R \|\mu\|^2 + \frac{1}{2nT} \sum_{t \in [T] \backslash \mathcal{P}} \left( \sum_{y_i=1} \langle \mu_{+1}, \epsilon_t^i \rangle - \sum_{y_i=-1} \langle \mu_{+1}, \epsilon_t^i \rangle \right) \right)$$

$$+ \frac{\|\nu\|}{\|d\|} \left( \frac{1-\beta}{4} \zeta_R \langle \epsilon_r^j, \mu_{+1} - \mu_{-1} \rangle + \frac{\alpha\beta}{4} \zeta_P \langle \epsilon_r^j, \tilde{\mu}_{+1} - \tilde{\mu}_{-1} \rangle + \frac{1}{2nT} \sum_{t \in [T] \backslash \mathcal{P}} \left( \sum_{y_i=1} \langle \epsilon_t^i, \epsilon_r^j \rangle - \sum_{y_i=-1} \langle \epsilon_t^i, \epsilon_r^j \rangle \right) \right)$$

$$y^k \gamma_r^k = y^k {x_r^k}^\top \nu$$

$$= (\mu_{-1} + \epsilon_r^k)^\top \frac{\|\nu\|}{\|d\|} \left( \frac{1-\beta}{4} \zeta_R (\mu_{+1} - \mu_{-1}) + \frac{\alpha\beta}{4} \zeta_P (\tilde{\mu}_{+1} - \tilde{\mu}_{-1}) + \frac{1}{2nT} \sum_{t \in [T] \backslash \mathcal{P}} \left( \sum_{y_i=1} \epsilon_t^i - \sum_{y_i=-1} \epsilon_t^i \right) \right)$$

$$= \frac{\|\nu\|}{\|d\|} \left( -\frac{1-\beta}{4} \zeta_R \|\mu\|^2 + \frac{1}{2nT} \sum_{t \in [T] \backslash \mathcal{P}} \left( \sum_{y_i=1} \langle \mu_{-1}, \epsilon_t^i \rangle - \sum_{y_i=-1} \langle \mu_{-1}, \epsilon_t^i \rangle \right) \right)$$

$$+ \frac{\|\nu\|}{\|\mathbf{d}\|} \left( \frac{1-\beta}{4} \zeta_R \left\langle \epsilon_r^k, \mu_{+1} - \mu_{-1} \right\rangle + \frac{\alpha\beta}{4} \zeta_P \left\langle \epsilon_r^k, \tilde{\mu}_{+1} - \tilde{\mu}_{-1} \right\rangle + \frac{1}{2nT} \sum_{t\in[T]\setminus\mathcal{P}} \left( \sum_{y_i=1} \left\langle \epsilon_t^i, \epsilon_r^k \right\rangle - \sum_{y_i=-1} \left\langle \epsilon_t^i, \epsilon_r^k \right\rangle \right) \right)$$

$$y^k \gamma_p^k = y^k \mathbf{x}_p^{k^\top} \nu$$

$$= \alpha \tilde{\mu}_{+1}^\top \frac{\|\nu\|}{\|\mathbf{d}\|} \left( \frac{1-\beta}{4} \zeta_R (\mu_{+1} - \mu_{-1}) + \frac{\alpha\beta}{4} \zeta_P (\tilde{\mu}_{+1} - \tilde{\mu}_{-1}) + \frac{1}{2nT} \sum_{t\in[T]\setminus\mathcal{P}} \left( \sum_{y_i=1} \epsilon_t^i - \sum_{y_i=-1} \epsilon_t^i \right) \right)$$

$$= \frac{\|\nu\|}{\|\mathbf{d}\|} \left( \frac{\alpha^2 \beta}{4} \zeta_P \|\mu\|^2 + \frac{\alpha}{2nT} \sum_{t\in[T]\setminus\mathcal{P}} \left( \sum_{y_i=1} \left\langle \tilde{\mu}_{+1}, \epsilon_t^i \right\rangle - \sum_{y_i=-1} \left\langle \tilde{\mu}_{+1}, \epsilon_t^i \right\rangle \right) \right)$$

Similarly, for training data with label $y^j = -1$, $y^k = -1$, we have

$$y^j \gamma_r^j = y^j \mathbf{x}_r^{j^\top} \nu$$

$$= -(\mu_{-1} + \epsilon_r^j)^\top \frac{\|\nu\|}{\|\mathbf{d}\|} \left( \frac{1-\beta}{4} \zeta_R (\mu_{+1} - \mu_{-1}) + \frac{\alpha\beta}{4} \zeta_P (\tilde{\mu}_{+1} - \tilde{\mu}_{-1}) + \frac{1}{2nT} \sum_{t\in[T]\setminus\mathcal{P}} \left( \sum_{y_i=1} \epsilon_t^i - \sum_{y_i=-1} \epsilon_t^i \right) \right)$$

$$= \frac{\|\nu\|}{\|\mathbf{d}\|} \left( \frac{1-\beta}{4} \zeta_R \|\mu\|^2 + \frac{1}{2nT} \sum_{t\in[T]\setminus\mathcal{P}} \left( \sum_{y_i=-1} \left\langle \mu_{-1}, \epsilon_t^i \right\rangle - \sum_{y_i=1} \left\langle \mu_{-1}, \epsilon_t^i \right\rangle \right) \right)$$

$$+ \frac{\|\nu\|}{\|\mathbf{d}\|} \left( \frac{1-\beta}{4} \zeta_R \left\langle \epsilon_r^j, \mu_{-1} - \mu_{+1} \right\rangle + \frac{\alpha\beta}{4} \zeta_P \left\langle \epsilon_r^j, \tilde{\mu}_{-1} - \tilde{\mu}_{+1} \right\rangle + \frac{1}{2nT} \sum_{t\in[T]\setminus\mathcal{P}} \left( \sum_{y_i=-1} \left\langle \epsilon_t^i, \epsilon_r^j \right\rangle - \sum_{y_i=+1} \left\langle \epsilon_t^i, \epsilon_r^j \right\rangle \right) \right)$$

$$y^k \gamma_r^k = y^k \mathbf{x}_r^{k^\top} \nu$$

$$= -(\mu_{+1} + \epsilon_r^k)^\top \frac{\|\nu\|}{\|\mathbf{d}\|} \left( \frac{1-\beta}{4} \zeta_R (\mu_{+1} - \mu_{-1}) + \frac{\alpha\beta}{4} \zeta_P (\tilde{\mu}_{+1} - \tilde{\mu}_{-1}) + \frac{1}{2nT} \sum_{t\in[T]\setminus\mathcal{P}} \left( \sum_{y_i=1} \epsilon_t^i - \sum_{y_i=-1} \epsilon_t^i \right) \right)$$

$$= \frac{\|\nu\|}{\|\mathbf{d}\|} \left( -\frac{1-\beta}{4} \zeta_R \|\mu\|^2 + \frac{1}{2nT} \sum_{t\in[T]\setminus\mathcal{P}} \left( \sum_{y_i=-1} \left\langle \mu_{+1}, \epsilon_t^i \right\rangle - \sum_{y_i=1} \left\langle \mu_{+1}, \epsilon_t^i \right\rangle \right) \right)$$

$$+ \frac{\|\nu\|}{\|\mathbf{d}\|} \left( \frac{1-\beta}{4} \zeta_R \left\langle \epsilon_r^k, \mu_{+1} - \mu_{-1} \right\rangle + \frac{\alpha\beta}{4} \zeta_P \left\langle \epsilon_r^k, \tilde{\mu}_{+1} - \tilde{\mu}_{-1} \right\rangle + \frac{1}{2nT} \sum_{t\in[T]\setminus\mathcal{P}} \left( \sum_{y_i=-1} \left\langle \epsilon_t^i, \epsilon_r^k \right\rangle - \sum_{y_i=1} \left\langle \epsilon_t^i, \epsilon_r^k \right\rangle \right) \right)$$

$$y^k \gamma_p^k = y^k \mathbf{x}_p^{k^\top} \nu$$

$$= -\alpha \tilde{\mu}_{-1}^\top \frac{\|\nu\|}{\|\mathbf{d}\|} \left( \frac{1-\beta}{4} \zeta_R (\mu_{+1} - \mu_{-1}) + \frac{\alpha\beta}{4} \zeta_P (\tilde{\mu}_{+1} - \tilde{\mu}_{-1}) + \frac{1}{2nT} \sum_{t\in[T]\setminus\mathcal{P}} \left( \sum_{y_i=-1} \epsilon_t^i - \sum_{y_i=1} \epsilon_t^i \right) \right)$$

$$= \frac{\|\nu\|}{\|\mathbf{d}\|} \left( \frac{\alpha^2 \beta}{4} \zeta_P \|\mu\|^2 + \frac{\alpha}{2nT} \sum_{t\in[T]\setminus\mathcal{P}} \left( \sum_{y_i=-1} \left\langle \tilde{\mu}_{-1}, \epsilon_t^i \right\rangle - \sum_{y_i=1} \left\langle \tilde{\mu}_{-1}, \epsilon_t^i \right\rangle \right) \right)$$

Using Lemma A.4 gives us that

$$y^j \gamma_r^j \leq \frac{\|\nu\|}{\|\mathbf{d}\|} \left( \frac{1-\beta}{4} \zeta_R \|\mu\|^2 + \left( \frac{(1-\beta)}{2} \zeta_R + \frac{\alpha\beta}{2} \zeta_P + \frac{1}{2} - \frac{\beta}{2} \zeta_P \right) c_2 \|\mu\| \sqrt{\log(Tn/\delta)} \right.$$

$$+ \frac{1}{2nT}\left((1+\frac{1}{C'})^2\operatorname{Tr}(\Sigma)+(nT-\beta\zeta_P nT-1)c_1\sqrt{\operatorname{Tr}(\Sigma)}\log\left(Tn/\delta\right)\right)\Bigg)$$

$$\leq \frac{\|\nu\|}{\|\mathbf{d}\|}\left(\frac{1-\beta}{4}\zeta_R\|\mu\|^2+\left(\frac{(1-\beta)}{2}\zeta_R+\frac{(\alpha-1)\beta}{2}\zeta_P+\frac{1}{2}\right)c_2\|\mu\|\sqrt{\log\left(Tn/\delta\right)}+\frac{(1+1/C')^2}{nT}\operatorname{Tr}(\Sigma)\right)$$

(Assumption (A1).)

$$y^j\gamma_r^j \geq \frac{\|\nu\|}{\|\mathbf{d}\|}\Bigg(\frac{1-\beta}{4}\zeta_R\|\mu\|^2-\left(\frac{(1-\beta)}{2}\zeta_R+\frac{\alpha\beta}{2}\zeta_P+\frac{1}{2}-\frac{\beta}{2}\zeta_P\right)c_2\|\mu\|\sqrt{\log\left(Tn/\delta\right)}$$

$$+\frac{1}{2nT}\left((1-\frac{1}{\operatorname{Tr}(\Sigma)}-\frac{1}{C'})^2\operatorname{Tr}(\Sigma)-(nT-\beta\zeta_P nT-1)c_1\sqrt{\operatorname{Tr}(\Sigma)}\log\left(Tn/\delta\right)\right)\Bigg)$$

$$\geq \frac{\|\nu\|}{\|\mathbf{d}\|}\left(\frac{1-\beta}{4}\zeta_R\|\mu\|^2-\left(\frac{(1-\beta)}{2}\zeta_R+\frac{(\alpha-1)\beta}{2}\zeta_P+\frac{1}{2}\right)c_2\|\mu\|\sqrt{\log\left(Tn/\delta\right)}\right)$$

(Assumption (A1).)

$$y^k\gamma_r^k \leq \frac{\|\nu\|}{\|\mathbf{d}\|}\left(-\frac{1-\beta}{4}\zeta_R\|\mu\|^2+\left(\frac{(1-\beta)}{2}\zeta_R+\frac{(\alpha-1)\beta}{2}\zeta_P+\frac{1}{2}\right)c_2\|\mu\|\sqrt{\log\left(Tn/\delta\right)}+\frac{(1+1/C')^2}{nT}\operatorname{Tr}(\Sigma)\right)$$

$$y^k\gamma_r^k \geq \frac{\|\nu\|}{\|\mathbf{d}\|}\left(-\frac{1-\beta}{4}\zeta_R\|\mu\|^2-\left(\frac{(1-\beta)}{2}\zeta_R+\frac{(\alpha-1)\beta}{2}\zeta_P+\frac{1}{2}\right)c_2\|\mu\|\sqrt{\log\left(Tn/\delta\right)}\right)$$

$$y^k\gamma_p^k \leq \frac{\|\nu\|}{\|\mathbf{d}\|}\left(\frac{\alpha^2\beta}{4}\zeta_P\|\mu\|^2+\left(\frac{\alpha}{2}-\frac{\alpha\beta}{2}\zeta_P\right)c_2\|\mu\|\sqrt{\log\left(Tn/\delta\right)}\right)$$

$$y^k\gamma_p^k \geq \frac{\|\nu\|}{\|\mathbf{d}\|}\left(\frac{\alpha^2\beta}{4}\zeta_P\|\mu\|^2-\left(\frac{\alpha}{2}-\frac{\alpha\beta}{2}\zeta_P\right)c_2\|\mu\|\sqrt{\log\left(Tn/\delta\right)}\right)$$

$$y^i\gamma_v^i = y^i{\mathbf{x}_v^i}^\top\nu$$

$$= \frac{\|\nu\|}{\|\mathbf{d}\|}\left(\frac{1-\beta}{4}\zeta_R\langle y^i\epsilon_v^i,\mu_{+1}-\mu_{-1}\rangle+\frac{\alpha\beta}{4}\zeta_P\langle y^i\epsilon_v^i,\tilde{\mu}_{+1}-\tilde{\mu}_{-1}\rangle+\frac{1}{2nT}\sum_{t\in[T]\setminus\mathcal{P}}\left(\sum_{y_i=-1}\langle y^i\epsilon_v^i,\epsilon_t^i\rangle-\sum_{y_i=+1}\langle y^i\epsilon_v^i,\epsilon_t^i\rangle\right)\right)$$

$$\leq \frac{\|\nu\|}{\|\mathbf{d}\|}\left(\left(\frac{(1-\beta)}{2}\zeta_R+\frac{\alpha\beta}{2}\zeta_P\right)c_2\|\mu\|\sqrt{\log\left(Tn/\delta\right)}+\frac{(1+1/C')^2}{nT}\operatorname{Tr}(\Sigma)\right)$$

$$y^i\gamma_v^i \geq -\frac{\|\nu\|}{\|\mathbf{d}\|}\left(\left(\frac{(1-\beta)}{2}\zeta_R+\frac{\alpha\beta}{2}\zeta_P\right)c_2\|\mu\|\sqrt{\log\left(Tn/\delta\right)}\right)$$

where Assumption (A1) gives us that $\operatorname{Tr}(\Sigma)\geq Cn^2T^2\log\left(Tn/\delta\right)^2$, otherwise all the term $\frac{\operatorname{Tr}(\Sigma)}{nT}$ would be replaced by $\sqrt{\operatorname{Tr}(\Sigma)}\log\left(Tn/\delta\right)$, and all the lower bound would add an additional term $-\sqrt{\operatorname{Tr}(\Sigma)}\log\left(Tn/\delta\right)$.

$\square$

**Lemma A.6.** Let $p(\tau)$ be a gradient iteration at $\tau$-th time step. Then there exists unique coefficients such that

$$p(\tau) = \lambda_{+1}(\tau)\mathbf{W}\mu_{+1}+\lambda_{-1}(\tau)\mathbf{W}\mu_{-1}+\tilde{\lambda}_{+1}(\tau)\mathbf{W}\tilde{\mu}_{+1}+\tilde{\lambda}_{-1}(\tau)\mathbf{W}\tilde{\mu}_{-1}+\sum_{i\in[n]}\sum_{t\in[T]\setminus\mathcal{P}}\rho_{i,t}(\tau)\mathbf{W}\epsilon_t^i,$$

where the initialization is $\lambda_c(0)=\tilde{\lambda}_c(0)=\rho_{i,t}(0)=0$ for any $c\in\{\pm1\},i\in[n],t\in[T]$, and the signal updates are given by

$$\Delta\lambda_c(\tau) = \lambda_c(\tau+1)-\lambda_c(\tau) = \frac{\eta}{n}\sum_{i\in\mathcal{C}_c\cup\mathcal{N}_{-c}}(-\ell_i'(\tau))\cdot\sum_{r\in\mathcal{R}}s^i(\tau)_r\left(y^i\gamma_r^i-\sum_{u\in[T]}s^i(\tau)_u y^i\gamma_u^i\right),$$

$$\Delta\tilde{\lambda}_c(\tau) = \tilde{\lambda}_c(\tau+1)-\tilde{\lambda}_c(\tau) = \frac{\alpha\eta}{n}\sum_{i\in\mathcal{N}_c}(-\ell_i'(\tau))\cdot\sum_{p\in\mathcal{P}}s^i(\tau)_p\left(y^i\gamma_p^i-\sum_{u\in[T]}s^i(\tau)_u y^i\gamma_u^i\right),$$

and the noise updates are given by

$$\Delta\rho_{i,t}(\tau) = \rho_{i,t}(\tau+1) - \rho(\tau) = \frac{\eta}{n}(-\ell_i'(\tau)) \cdot y^i \cdot s^i(\tau)_t \left(\gamma_t^i - \sum_{u \in [T]} s^i(\tau)_u \gamma_u^i\right), \forall i \in [n], t \in [T].$$

*Proof of Lemma A.6.* Recall that

$$\nabla_{\mathrm{p}}\widehat{L}(\mathrm{p}) = \frac{1}{n}\sum_{i=1}^{n} \ell_i' \cdot y^i \cdot \nabla_{\mathrm{p}} f(X^i)$$

$$= \frac{1}{n}\sum_{i=1}^{n} \ell_i' \cdot y^i \cdot WX^{i\top} \left(\mathrm{diag}(\mathbb{S}(X^i W^\top \mathrm{p})) - \mathbb{S}(X^i W^\top \mathrm{p})\mathbb{S}(X^i W^\top \mathrm{p})^\top\right) X^i \nu$$

$$= \frac{1}{n}\sum_{i=1}^{n} \ell_i' \cdot y^i \cdot \left(\sum_{t \in [T]} s_t^i \left(\gamma_t^i - \sum_{u \in [T]} s_u^i \gamma_u^i\right) Wx_t^i\right)$$

where $\ell_i'$ is abbreviation for $\ell'(y^i \cdot \nu^\top X^{i\top} \mathbb{S}(X^i W^\top \mathrm{p}))$, $s_t^i$ is abbreviation for $\mathbb{S}(X^i W^\top \mathrm{p})_t$, $\gamma_t^i$ is abbreviation for $x_t^{i\top}\nu$, $\forall t \in [T]$. In the dirty-label backdoor attack setup,

$$x_t^i = \begin{cases} \mu_{y^i} + \epsilon_t^i & i \in \mathcal{C}, t \in \mathcal{R} \\ \mu_{-y^i} + \epsilon_t^i & i \in \mathcal{N}, t \in \mathcal{R} \\ \alpha\tilde{\mu}_{y^i} & i \in \mathcal{N}, t \in \mathcal{P} \\ \epsilon_t^i & i \in \mathcal{C} \cup \mathcal{N}, t \in \mathcal{I} \end{cases}$$

Therefore we have

$$\mathrm{p}(\tau+1) - \mathrm{p}(\tau) = -\eta\nabla_{\mathrm{p}}\widehat{L}(\mathrm{p}(\tau))$$

$$= \frac{\eta}{n}\sum_{i=1}^{n}(-\ell_i'(\tau)) \cdot y^i \cdot \left(\sum_{t \in [T]} s^i(\tau)_t \left(\gamma_t^i - \sum_{u \in [T]} s^i(\tau)_u \gamma_u^i\right) Wx_t^i\right)$$

$$= \frac{\eta}{n}\sum_{i \in \mathcal{C}}(-\ell_i'(\tau)) \cdot y^i \cdot \left(\sum_{r \in \mathcal{R}} s^i(\tau)_r \left(\gamma_r^i - \sum_{u \in [T]} s^i(\tau)_u \gamma_u^i\right) W\mu_{y^i}\right)$$

$$+ \frac{\eta}{n}\sum_{i \in \mathcal{N}}(-\ell_i'(\tau)) \cdot y^i \cdot \left(\sum_{r \in \mathcal{R}} s^i(\tau)_r \left(\gamma_r^i - \sum_{u \in [T]} s^i(\tau)_u \gamma_u^i\right) W\mu_{-y^i}\right)$$

$$+ \frac{\alpha\eta}{n}\sum_{i \in \mathcal{N}}(-\ell_i'(\tau)) \cdot y^i \cdot \left(\sum_{p \in \mathcal{P}} s^i(\tau)_p \left(\gamma_p^i - \sum_{u \in [T]} s^i(\tau)_u \gamma_u^i\right) W\tilde{\mu}_{y^i}\right)$$

$$+ \frac{\eta}{n}\sum_{i=1}^{n}(-\ell_i'(\tau)) \cdot y^i \cdot \left(\sum_{t \in [T] \backslash \mathcal{P}} s^i(\tau)_t \left(\gamma_t^i - \sum_{u \in [T]} s^i(\tau)_u \gamma_u^i\right) W\epsilon_t^i\right)$$

As a result, we are able to decompose into the following form

$$\mathrm{p}(\tau+1) - \mathrm{p}(\tau)$$
$$= \Delta\lambda_{+1}(\tau)W\mu_{+1} + \Delta\lambda_{-1}(\tau)W\mu_{-1} + \Delta\tilde{\lambda}_{+1}(\tau)W\tilde{\mu}_{+1} + \Delta\tilde{\lambda}_{-1}(\tau)W\tilde{\mu}_{-1} + \sum_{i \in [n]} \sum_{t \in [T] \backslash \mathcal{P}} \Delta\rho_{i,t}(\tau)W\epsilon_t^i,$$

where we have $\forall c \in \{\pm 1\}$,

$$\Delta\lambda_c(\tau) = \lambda_c(\tau+1) - \lambda_c(\tau) = \frac{\eta}{n}\sum_{i \in \mathcal{C}_c \cup \mathcal{N}_{-c}}(-\ell_i'(\tau)) \cdot \sum_{r \in \mathcal{R}} s^i(\tau)_r \left(y^i \gamma_r^i - \sum_{u \in [T]} s^i(\tau)_u y^i \gamma_u^i\right)$$

$$\Delta\tilde{\lambda}_c(\tau) = \tilde{\lambda}_c(\tau+1) - \tilde{\lambda}_c(\tau) = \frac{\alpha\eta}{n} \sum_{i \in \mathcal{N}_c} (-\ell_i'(\tau)) \cdot \sum_{p \in \mathcal{P}} s^i(\tau)_p \left( y^i \gamma_p^i - \sum_{u \in [T]} s^i(\tau)_u y^i \gamma_u^i \right)$$

$$\Delta\rho_{i,t}(\tau) = \rho_{i,t}(\tau+1) - \rho(\tau) = \frac{\eta}{n}(-\ell_i'(\tau)) \cdot s^i(\tau)_t \left( y^i \gamma_t^i - \sum_{u \in [T]} s^i(\tau)_u y^i \gamma_u^i \right), \forall i \in [n], t \in [T] \backslash \mathcal{P}$$

Below we calculate the upper bound and lower bound of $\Delta\lambda_{+1}, \Delta\lambda_{-1}, \Delta\tilde{\lambda}_{+1}, \Delta\tilde{\lambda}_{-1}$ and $\Delta\rho_{i,t}$.

Note that for $i \in \mathcal{C}$ and any relevant token $r \in \mathcal{R}$, we have

$$y^i \gamma_r^i - \sum_{u \in [T]} s^i(\tau)_u y^i \gamma_u^i$$

$$= \sum_{u \in [T] \backslash \{r\}} s^i(\tau)_u \left( y^i \gamma_r^i - y^i \gamma_u^i \right)$$

$$= \sum_{u \in \mathcal{R} \backslash \{r\}} s^i(\tau)_u \left( y^i \gamma_r^i - y^i \gamma_u^i \right) + \sum_{u \in \mathcal{I}} s^i(\tau)_u \left( y^i \gamma_r^i - y^i \gamma_u^i \right)$$

$$\leq \frac{\|\nu\|}{\|d\|} \sum_{u \in \mathcal{R} \backslash \{r\}} s^i(\tau)_u \left( ((1-\beta)\zeta_R + (\alpha-1)\beta\zeta_P + 1) c_2 \|\mu\| \sqrt{\log(Tn/\delta)} + \frac{(1+1/C')^2}{nT} \text{Tr}(\Sigma) \right)$$

$$+ \frac{\|\nu\|}{\|d\|} \sum_{u \in \mathcal{I}} s^i(\tau)_u \left( \frac{1-\beta}{4} \zeta_R \|\mu\|^2 + \left( (1-\beta)\zeta_R + \frac{(2\alpha-1)\beta}{2}\zeta_P + \frac{1}{2} \right) c_2 \|\mu\| \sqrt{\log(Tn/\delta)} + \frac{(1+1/C')^2}{nT} \text{Tr}(\Sigma) \right)$$

$$\leq \frac{\|\nu\|}{\|d\|} \left( 1 - \sum_{u \in \mathcal{R}} s^i(\tau)_u \right) \frac{1-\beta}{4} \zeta_R \|\mu\|^2$$

$$+ \frac{\|\nu\|}{\|d\|} \left( 1 - s^i(\tau)_r \right) \left( \left( (1-\beta)\zeta_R + \frac{(2\alpha-1)\beta}{2}\zeta_P + 1 \right) c_2 \|\mu\| \sqrt{\log(Tn/\delta)} + \frac{(1+1/C')^2}{nT} \text{Tr}(\Sigma) \right)$$

$$y^i \gamma_r^i - \sum_{u \in [T]} s^i(\tau)_u y^i \gamma_u^i$$

$$\geq \frac{\|\nu\|}{\|d\|} \sum_{u \in \mathcal{R} \backslash \{r\}} s^i(\tau)_u \left( - ((1-\beta)\zeta_R + (\alpha-1)\beta\zeta_P + 1) c_2 \|\mu\| \sqrt{\log(Tn/\delta)} - \frac{(1+1/C')^2}{nT} \text{Tr}(\Sigma) \right)$$

$$+ \frac{\|\nu\|}{\|d\|} \sum_{u \in \mathcal{I}} s^i(\tau)_u \left( \frac{1-\beta}{4} \zeta_R \|\mu\|^2 - \left( (1-\beta)\zeta_R + \frac{(2\alpha-1)\beta}{2}\zeta_P + \frac{1}{2} \right) c_2 \|\mu\| \sqrt{\log(Tn/\delta)} - \frac{(1+1/C')^2}{nT} \text{Tr}(\Sigma) \right)$$

$$\geq \frac{\|\nu\|}{\|d\|} \left( 1 - \sum_{u \in \mathcal{R}} s^i(\tau)_u \right) \frac{1-\beta}{4} \zeta_R \|\mu\|^2$$

$$- \frac{\|\nu\|}{\|d\|} \left( 1 - s^i(\tau)_r \right) \left( \left( (1-\beta)\zeta_R + \frac{(2\alpha-1)\beta}{2}\zeta_P + 1 \right) c_2 \|\mu\| \sqrt{\log(Tn/\delta)} + \frac{(1+1/C')^2}{nT} \text{Tr}(\Sigma) \right)$$

For $i \in \mathcal{C}$ and any irrelevant token $v \in \mathcal{I}$, we have

$$y^i \gamma_v^i - \sum_{u \in [T]} s^i(\tau)_u y^i \gamma_u^i$$

$$= \sum_{u \in \mathcal{R}} s^i(\tau)_u \left( y^i \gamma_v^i - y^i \gamma_u^i \right) + \sum_{u \in \mathcal{I} \backslash \{v\}} s^i(\tau)_u \left( y^i \gamma_v^i - y^i \gamma_u^i \right)$$

$$\leq \frac{\|\nu\|}{\|d\|} \sum_{u \in \mathcal{R}} s^i(\tau)_u \left( -\frac{1-\beta}{4} \zeta_R \|\mu\|^2 + \left( (1-\beta)\zeta_R + \frac{(2\alpha-1)\beta}{2}\zeta_P + \frac{1}{2} \right) c_2 \|\mu\| \sqrt{\log(Tn/\delta)} + \frac{(1+1/C')^2}{nT} \text{Tr}(\Sigma) \right)$$

$$+ \frac{\|\nu\|}{\|\mathbf{d}\|} \sum_{u \in \mathcal{I} \backslash \{v\}} s^i(\tau)_u \left( \left( (1-\beta)\zeta_R + \alpha\beta\zeta_P \right) c_2 \|\mu\| \sqrt{\log(Tn/\delta)} + \frac{(1+1/C')^2}{nT} \operatorname{Tr}(\Sigma) \right)$$

$$\leq \frac{\|\nu\|}{\|\mathbf{d}\|} \left( 1 - s^i(\tau)_v \right) \left( \left( (1-\beta)\zeta_R + \alpha\beta\zeta_P + \frac{1}{2} \right) c_2 \|\mu\| \sqrt{\log(Tn/\delta)} + \frac{(1+1/C')^2}{nT} \operatorname{Tr}(\Sigma) \right)$$

$$- \frac{\|\nu\|}{\|\mathbf{d}\|} \sum_{u \in \mathcal{R}} s^i(\tau)_u \frac{1-\beta}{4} \zeta_R \|\mu\|^2$$

$$y^i \gamma_v^i - \sum_{u \in [T]} s^i(\tau)_u y^i \gamma_u^i$$

$$\geq \frac{\|\nu\|}{\|\mathbf{d}\|} \sum_{u \in \mathcal{R}} s^i(\tau)_u \left( -\frac{1-\beta}{4} \zeta_R \|\mu\|^2 - \left( (1-\beta)\zeta_R + (2\alpha-1)\beta\zeta_P + \frac{1}{2} \right) c_2 \|\mu\| \sqrt{\log(Tn/\delta)} - \frac{(1+1/C')^2}{nT} \operatorname{Tr}(\Sigma) \right)$$

$$- \frac{\|\nu\|}{\|\mathbf{d}\|} \sum_{u \in \mathcal{I} \backslash \{v\}} s^i(\tau)_u \left( \left( (1-\beta)\zeta_R + \alpha\beta\zeta_P \right) c_2 \|\mu\| \sqrt{\log(Tn/\delta)} + \frac{(1+1/C')^2}{nT} \operatorname{Tr}(\Sigma) \right)$$

$$\geq -\frac{\|\nu\|}{\|\mathbf{d}\|} \left( 1 - s^i(\tau)_v \right) \left( \left( (1-\beta)\zeta_R + \alpha\beta\zeta_P + \frac{1}{2} \right) c_2 \|\mu\| \sqrt{\log(Tn/\delta)} + \frac{(1+1/C')^2}{nT} \operatorname{Tr}(\Sigma) \right)$$

$$- \frac{\|\nu\|}{\|\mathbf{d}\|} \sum_{u \in \mathcal{R}} s^i(\tau)_u \frac{1-\beta}{4} \zeta_R \|\mu\|^2$$

Similarly, for $i \in \mathcal{N}$ and any relevant $r \in \mathcal{R}$, we have

$$y^i \gamma_r^i - \sum_{u \in [T]} s^i(\tau)_u y^i \gamma_u^i$$

$$= \sum_{u \in \mathcal{R} \backslash \{r\}} s^i(\tau)_u \left( y^i \gamma_r^i - y^i \gamma_u^i \right) + \sum_{u \in \mathcal{P}} s^i(\tau)_u \left( y^i \gamma_r^i - y^i \gamma_u^i \right) + \sum_{u \in \mathcal{I}} s^i(\tau)_u \left( y^i \gamma_r^i - y^i \gamma_u^i \right)$$

$$\leq \frac{\|\nu\|}{\|\mathbf{d}\|} \sum_{u \in \mathcal{R} \backslash \{r\}} s^i(\tau)_u \left( \left( (1-\beta)\zeta_R + (\alpha-1)\beta\zeta_P + 1 \right) c_2 \|\mu\| \sqrt{\log(Tn/\delta)} + \frac{(1+1/C')^2}{nT} \operatorname{Tr}(\Sigma) \right)$$

$$+ \frac{\|\nu\|}{\|\mathbf{d}\|} \sum_{u \in \mathcal{P}} s^i(\tau)_u \left( -\left( \frac{1-\beta}{4} \zeta_R + \frac{\alpha^2 \beta}{4} \zeta_P \right) \|\mu\|^2 + \left( \frac{(1-\beta)}{2} \zeta_R - \frac{\beta}{2} \zeta_P + \frac{1+\alpha}{2} \right) c_2 \|\mu\| \sqrt{\log(Tn/\delta)} + \frac{(1+1/C')^2}{nT} \operatorname{Tr}(\Sigma) \right)$$

$$+ \frac{\|\nu\|}{\|\mathbf{d}\|} \sum_{u \in \mathcal{I}} s^i(\tau)_u \left( -\frac{1-\beta}{4} \zeta_R \|\mu\|^2 + \left( (1-\beta)\zeta_R + \frac{(2\alpha-1)\beta}{2} \zeta_P + \frac{1}{2} \right) c_2 \|\mu\| \sqrt{\log(Tn/\delta)} + \frac{(1+1/C')^2}{nT} \operatorname{Tr}(\Sigma) \right)$$

$$\leq -\frac{\|\nu\|}{\|\mathbf{d}\|} \left( 1 - \sum_{u \in \mathcal{R}} s^i(\tau)_u \right) \frac{1-\beta}{4} \zeta_R \|\mu\|^2 - \frac{\|\nu\|}{\|\mathbf{d}\|} \sum_{u \in \mathcal{P}} s^i(\tau)_u \frac{\alpha^2 \beta}{4} \zeta_P \|\mu\|^2$$

$$+ \frac{\|\nu\|}{\|\mathbf{d}\|} \left( 1 - s^i(\tau)_r \right) \left( \left( (1-\beta)\zeta_R + \frac{(2\alpha-1)\beta}{2} \zeta_P + \frac{1+\alpha}{2} \right) c_2 \|\mu\| \sqrt{\log(Tn/\delta)} + \frac{(1+1/C')^2}{nT} \operatorname{Tr}(\Sigma) \right)$$

$$y^i \gamma_r^i - \sum_{u \in [T]} s^i(\tau)_u y^i \gamma_u^i$$

$$\geq -\frac{\|\nu\|}{\|\mathbf{d}\|} \sum_{u \in \mathcal{R} \backslash \{r\}} s^i(\tau)_u \left( \left( (1-\beta)\zeta_R + (\alpha-1)\beta\zeta_P + 1 \right) c_2 \|\mu\| \sqrt{\log(Tn/\delta)} + \frac{(1+1/C')^2}{nT} \operatorname{Tr}(\Sigma) \right)$$

$$+ \frac{\|\nu\|}{\|\mathbf{d}\|} \sum_{u \in \mathcal{P}} s^i(\tau)_u \left( -\left( \frac{1-\beta}{4} \zeta_R + \frac{\alpha^2 \beta}{4} \zeta_P \right) \|\mu\|^2 - \left( (1-\beta)\zeta_R - \frac{\beta}{2} \zeta_P + \frac{1+\alpha}{2} \right) c_2 \|\mu\| \sqrt{\log(Tn/\delta)} \right)$$

$$+ \frac{\|\nu\|}{\|\mathbf{d}\|} \sum_{u\in\mathcal{I}} s^i(\tau)_u \left( -\frac{1-\beta}{4}\zeta_R \|\mu\|^2 - \left((1-\beta)\zeta_R + \frac{(2\alpha-1)\beta}{2}\zeta_P + \frac{1}{2}\right) c_2 \|\mu\| \sqrt{\log(Tn/\delta)} - \frac{(1+1/C')^2}{nT}\operatorname{Tr}(\Sigma) \right)$$

$$\geq -\frac{\|\nu\|}{\|\mathbf{d}\|} \left( 1 - \sum_{u\in\mathcal{R}} s^i(\tau)_u \right) \frac{1-\beta}{4}\zeta_R \|\mu\|^2 - \frac{\|\nu\|}{\|\mathbf{d}\|} \sum_{u\in\mathcal{P}} s^i(\tau)_u \frac{\alpha^2\beta}{4}\zeta_P \|\mu\|^2$$

$$- \frac{\|\nu\|}{\|\mathbf{d}\|} \left( 1 - s^i(\tau)_r \right) \left( \left( (1-\beta)\zeta_R + \frac{(2\alpha-1)\beta}{2}\zeta_P + \frac{1+\alpha}{2} \right) c_2 \|\mu\| \sqrt{\log(Tn/\delta)} + \frac{(1+1/C')^2}{nT}\operatorname{Tr}(\Sigma) \right)$$

For $i \in \mathcal{N}$ and any poison $p \in \mathcal{P}$, we have

$$y^i\gamma_p^i - \sum_{u\in[T]} s^i(\tau)_u y^i\gamma_u^i$$

$$= \sum_{u\in\mathcal{R}} s^i(\tau)_u \left( y^i\gamma_p^i - y^i\gamma_u^i \right) + \sum_{u\in\mathcal{P}\setminus\{p\}} s^i(\tau)_u \left( y^i\gamma_p^i - y^i\gamma_u^i \right) + \sum_{u\in\mathcal{I}} s^i(\tau)_u \left( y^i\gamma_p^i - y^i\gamma_u^i \right)$$

$$\leq \frac{\|\nu\|}{\|\mathbf{d}\|} \sum_{u\in\mathcal{R}} s^i(\tau)_u \left( \left( \frac{\alpha^2\beta}{4}\zeta_P + \frac{1-\beta}{4}\zeta_R \right) \|\mu\|^2 + \left( \frac{1-\beta}{2}\zeta_R - \frac{\beta}{2}\zeta_P + \frac{\alpha+1}{2} \right) c_2 \|\mu\| \sqrt{\log(Tn/\delta)} \right)$$

$$+ \frac{\|\nu\|}{\|\mathbf{d}\|} \sum_{u\in\mathcal{P}\setminus\{p\}} s^i(\tau)_u \left( (\alpha - \alpha\beta\zeta_P) c_2 \|\mu\| \sqrt{\log(Tn/\delta)} \right)$$

$$+ \frac{\|\nu\|}{\|\mathbf{d}\|} \sum_{u\in\mathcal{I}} s^i(\tau)_u \left( \frac{\alpha^2\beta}{4}\zeta_P \|\mu\|^2 + \left( \frac{1-\beta}{2}\zeta_R + \frac{\alpha}{2} \right) c_2 \|\mu\| \sqrt{\log(Tn/\delta)} \right)$$

$$\leq \frac{\|\nu\|}{\|\mathbf{d}\|} \left( 1 - s^i(\tau)_p \right) \left( \frac{1-\beta}{2}\zeta_R + \alpha \right) c_2 \|\mu\| \sqrt{\log(Tn/\delta)}$$

$$+ \frac{\|\nu\|}{\|\mathbf{d}\|} \left( 1 - \sum_{u\in\mathcal{P}} s^i(\tau)_u \right) \left( \frac{\alpha^2\beta}{4}\zeta_P + \frac{1-\beta}{4}\zeta_R \right) \|\mu\|^2 - \frac{\|\nu\|}{\|\mathbf{d}\|} \sum_{u\in\mathcal{I}} s^i(\tau)_u \left( \frac{1-\beta}{4}\zeta_R \|\mu\|^2 \right)$$

$$= \frac{\|\nu\|}{\|\mathbf{d}\|} \left( \left( 1 - s^i(\tau)_p \right) \left( \frac{1-\beta}{2}\zeta_R + \alpha \right) c_2 \|\mu\| \sqrt{\log(Tn/\delta)} + \left( 1 - \sum_{u\in\mathcal{P}} s^i(\tau)_u \right) \frac{\alpha^2\beta}{4}\zeta_P \|\mu\|^2 + \sum_{u\in\mathcal{R}} s^i(\tau)_u \frac{1-\beta}{4}\zeta_R \|\mu\|^2 \right)$$

$$y^i\gamma_p^i - \sum_{u\in[T]} s^i(\tau)_u y^i\gamma_u^i$$

$$\geq \frac{\|\nu\|}{\|\mathbf{d}\|} \sum_{u\in\mathcal{R}} s^i(\tau)_u \left( \left( \frac{\alpha^2\beta}{4}\zeta_P + \frac{1-\beta}{4}\zeta_R \right) \|\mu\|^2 - \left( \frac{(1-\beta)}{2}\zeta_R - \frac{\beta}{2}\zeta_P + \frac{\alpha+1}{2} \right) c_2 \|\mu\| \sqrt{\log(Tn/\delta)} \right.$$

$$\left. - \frac{(1+1/C')^2}{nT}\operatorname{Tr}(\Sigma) \right) - \frac{\|\nu\|}{\|\mathbf{d}\|} \sum_{u\in\mathcal{P}\setminus\{p\}} s^i(\tau)_u \left( (\alpha - \alpha\beta\zeta_P) c_2 \|\mu\| \sqrt{\log(Tn/\delta)} \right)$$

$$+ \frac{\|\nu\|}{\|\mathbf{d}\|} \sum_{u\in\mathcal{I}} s^i(\tau)_u \left( \frac{\alpha^2\beta}{4}\zeta_P \|\mu\|^2 - \left( \frac{(1-\beta)}{2}\zeta_R + \frac{\alpha}{2} \right) c_2 \|\mu\| \sqrt{\log(Tn/\delta)} - \frac{(1+1/C')^2}{nT}\operatorname{Tr}(\Sigma) \right)$$

$$\geq -\frac{\|\nu\|}{\|\mathbf{d}\|} \left( 1 - s^i(\tau)_p \right) \left( \frac{1-\beta}{2}\zeta_R + \alpha \right) c_2 \|\mu\| \sqrt{\log(Tn/\delta)}$$

$$+ \frac{\|\nu\|}{\|\mathbf{d}\|} \left( 1 - \sum_{u\in\mathcal{P}} s^i(\tau)_u \right) \left( \frac{\alpha^2\beta}{4}\zeta_P \|\mu\|^2 - \frac{(1+1/C')^2}{nT}\operatorname{Tr}(\Sigma) \right) + \frac{\|\nu\|}{\|\mathbf{d}\|} \sum_{u\in\mathcal{R}} s^i(\tau)_u \frac{1-\beta}{4}\zeta_R \|\mu\|^2$$

For $i \in \mathcal{N}$ and any irrelevant $v \in \mathcal{I}$, we have

$$y^i\gamma_v^i - \sum_{u\in[T]} s^i(\tau)_u y^i\gamma_u^i$$

$$= \sum_{u\in\mathcal{R}} s^i(\tau)_u \left(y^i\gamma_v^i - y^i\gamma_u^i\right) + \sum_{u\in\mathcal{P}} s^i(\tau)_u \left(y^i\gamma_v^i - y^i\gamma_u^i\right) + \sum_{u\in\mathcal{I}\setminus\{v\}} s^i(\tau)_u \left(y^i\gamma_v^i - y^i\gamma_u^i\right)$$

$$\leq \frac{\|\nu\|}{\|\mathbf{d}\|} \sum_{u\in\mathcal{R}} s^i(\tau)_u \left(\frac{1-\beta}{4}\zeta_R \|\mu\|^2 + \left((1-\beta)\zeta_R + \frac{(2\alpha-1)\beta}{2}\zeta_P + \frac{1}{2}\right) c_2 \|\mu\| \sqrt{\log(Tn/\delta)} + \frac{(1+1/C')^2}{nT} \operatorname{Tr}(\Sigma)\right)$$

$$+ \frac{\|\nu\|}{\|\mathbf{d}\|} \sum_{u\in\mathcal{P}} s^i(\tau)_u \left(-\frac{\alpha^2\beta}{4}\zeta_P \|\mu\|^2 + \left(\frac{\alpha}{2} + \frac{(1-\beta)}{2}\zeta_R\right) c_2 \|\mu\| \sqrt{\log(Tn/\delta)} + \frac{(1+1/C')^2}{nT} \operatorname{Tr}(\Sigma)\right)$$

$$+ \frac{\|\nu\|}{\|\mathbf{d}\|} \sum_{u\in\mathcal{I}\setminus\{v\}} s^i(\tau)_u \left(((1-\beta)\zeta_R + \alpha\beta\zeta_P) c_2 \|\mu\| \sqrt{\log(Tn/\delta)} + \frac{(1+1/C')^2}{nT} \operatorname{Tr}(\Sigma)\right)$$

$$\leq \frac{\|\nu\|}{\|\mathbf{d}\|} \left(1 - s^i(\tau)_v\right) \left(\left((1-\beta)\zeta_R + \alpha\beta\zeta_P + \frac{\alpha}{2}\right) c_2 \|\mu\| \sqrt{\log(Tn/\delta)} + \frac{(1+1/C')^2}{nT} \operatorname{Tr}(\Sigma)\right)$$

$$+ \frac{\|\nu\|}{\|\mathbf{d}\|} \sum_{u\in\mathcal{R}} s^i(\tau)_u \frac{1-\beta}{4}\zeta_R \|\mu\|^2 - \frac{\|\nu\|}{\|\mathbf{d}\|} \sum_{u\in\mathcal{P}} s^i(\tau)_u \frac{\alpha^2\beta}{4}\zeta_P \|\mu\|^2$$

$$y^i\gamma_v^i - \sum_{u\in[T]} s^i(\tau)_u y^i\gamma_u^i$$

$$\geq \frac{\|\nu\|}{\|\mathbf{d}\|} \sum_{u\in\mathcal{R}} s^i(\tau)_u \left(\frac{1-\beta}{4}\zeta_R \|\mu\|^2 - \left((1-\beta)\zeta_R + \frac{(2\alpha-1)\beta}{2}\zeta_P + \frac{1}{2}\right) c_2 \|\mu\| \sqrt{\log(Tn/\delta)} - \frac{(1+1/C')^2}{nT} \operatorname{Tr}(\Sigma)\right)$$

$$+ \frac{\|\nu\|}{\|\mathbf{d}\|} \sum_{u\in\mathcal{P}} s^i(\tau)_u \left(-\frac{\alpha^2\beta}{4}\zeta_P \|\mu\|^2 - \left(\frac{1-\beta}{2}\zeta_R + \frac{\alpha}{2}\right) c_2 \|\mu\| \sqrt{\log(Tn/\delta)} - \frac{(1+1/C')^2}{nT} \operatorname{Tr}(\Sigma)\right)$$

$$- \frac{\|\nu\|}{\|\mathbf{d}\|} \sum_{u\in\mathcal{I}\setminus\{v\}} s^i(\tau)_u \left(((1-\beta)\zeta_R + \alpha\beta\zeta_P) c_2 \|\mu\| \sqrt{\log(Tn/\delta)} + \frac{(1+1/C')^2}{nT} \operatorname{Tr}(\Sigma)\right)$$

$$\geq -\frac{\|\nu\|}{\|\mathbf{d}\|} \left(1 - s^i(\tau)_v\right) \left(\left((1-\beta)\zeta_R + \alpha\beta\zeta_P + \frac{\alpha}{2}\right) c_2 \|\mu\| \sqrt{\log(Tn/\delta)} + \frac{(1+1/C')^2}{nT} \operatorname{Tr}(\Sigma)\right)$$

$$+ \frac{\|\nu\|}{\|\mathbf{d}\|} \sum_{u\in\mathcal{R}} s^i(\tau)_u \frac{1-\beta}{4}\zeta_R \|\mu\|^2 - \frac{\|\nu\|}{\|\mathbf{d}\|} \sum_{u\in\mathcal{P}} s^i(\tau)_u \frac{\alpha^2\beta}{4}\zeta_P \|\mu\|^2$$

Therefore we have for $c \in \{\pm 1\}$,

$$\Delta\lambda_c(\tau) = \frac{\eta}{n} \sum_{i\in\mathcal{C}_c\cup\mathcal{N}_{-c}} (-\ell_i'(\tau)) \cdot \sum_{r\in\mathcal{R}} s^i(\tau)_r \left(y^i\gamma_r^i - \sum_{u\in[T]} s^i(\tau)_u y^i\gamma_u^i\right)$$

$$\leq \frac{\eta}{n} \frac{\|\nu\|}{\|\mathbf{d}\|} \sum_{i\in\mathcal{C}_c} (-\ell_i'(\tau)) \cdot \sum_{r\in\mathcal{R}} s^i(\tau)_r \left(\left(1 - \sum_{u\in\mathcal{R}} s^i(\tau)_u\right) \frac{1-\beta}{4}\zeta_R \|\mu\|^2\right.$$

$$+ \left(1 - s^i(\tau)_r\right) \left(\left((1-\beta)\zeta_R + \frac{(2\alpha-1)\beta}{2}\zeta_P + 1\right) c_2 \|\mu\| \sqrt{\log(Tn/\delta)} + \frac{(1+1/C')^2}{nT} \operatorname{Tr}(\Sigma)\right)\right)$$

$$+ \frac{\eta}{n} \frac{\|\nu\|}{\|\mathbf{d}\|} \sum_{i\in\mathcal{N}_{-c}} (-\ell_i'(\tau)) \cdot \sum_{r\in\mathcal{R}} s^i(\tau)_r \left(-\left(1 - \sum_{u\in\mathcal{R}} s^i(\tau)_u\right) \frac{1-\beta}{4}\zeta_R \|\mu\|^2 - \sum_{u\in\mathcal{P}} s^i(\tau)_u \frac{\alpha^2\beta}{4}\zeta_P \|\mu\|^2\right.$$

$$+ \left(1 - s^i(\tau)_r\right) \left(\left((1-\beta)\zeta_R + \frac{(2\alpha-1)\beta}{2}\zeta_P + 1\right) c_2 \|\mu\| \sqrt{\log(Tn/\delta)} + \frac{(1+1/C')^2}{nT} \operatorname{Tr}(\Sigma)\right)\right) \qquad (6)$$

$$\Delta\lambda_c(\tau) \geq \frac{\eta}{n} \frac{\|\nu\|}{\|\mathbf{d}\|} \sum_{i\in\mathcal{C}_c} (-\ell_i'(\tau)) \sum_{r\in\mathcal{R}} s^i(\tau)_r \left(\left(1 - \sum_{u\in\mathcal{R}} s^i(\tau)_u\right) \frac{1-\beta}{4}\zeta_R \|\mu\|^2\right.$$

$$- \left(1 - s^i(\tau)_r\right) \left(\left((1-\beta)\zeta_R + \frac{(2\alpha-1)\beta}{2}\zeta_P + 1\right) c_2 \|\mu\| \sqrt{\log(Tn/\delta)} + \frac{(1+1/C')^2}{nT} \operatorname{Tr}(\Sigma)\right)\right)$$

$$- \frac{\eta}{n} \frac{\|\nu\|}{\|\mathbf{d}\|} \sum_{i \in \mathcal{N}_{-c}} (-\ell_i'(\tau)) \sum_{r \in \mathcal{R}} s^i(\tau)_r \left(\left(1 - \sum_{u \in \mathcal{R}} s^i(\tau)_u\right) \frac{1-\beta}{4} \zeta_R \|\mu\|^2 + \sum_{u \in \mathcal{P}} s^i(\tau)_u \frac{\alpha^2 \beta}{4} \zeta_P \|\mu\|^2\right.$$

$$\left. + \left(1 - s^i(\tau)_r\right) \left(\left((1-\beta)\zeta_R + \frac{(2\alpha-1)\beta}{2}\zeta_P + \frac{1+\alpha}{2}\right) c_2 \|\mu\| \sqrt{\log(Tn/\delta)} + \frac{(1+1/C')^2}{nT} \operatorname{Tr}(\Sigma)\right)\right) \quad (7)$$

$$\Delta\tilde{\lambda}_c(\tau) = \frac{\alpha\eta}{n} \sum_{i \in \mathcal{N}_c} (-\ell_i'(\tau)) \cdot \sum_{p \in \mathcal{P}} s^i(\tau)_p \left(\gamma_p^i - \sum_{u \in [T]} s^i(\tau)_u \gamma_u^i\right)$$

$$\leq \frac{\alpha\eta}{n} \frac{\|\nu\|}{\|\mathbf{d}\|} \sum_{i \in \mathcal{N}_c} (-\ell_i'(\tau)) \cdot \sum_{p \in \mathcal{P}} s^i(\tau) \left((1 - s^i(\tau)_p)\left(\frac{1-\beta}{2}\zeta_R + \alpha\right) c_2 \|\mu\| \sqrt{\log(Tn/\delta)}\right.$$

$$\left. + \left(1 - \sum_{u \in \mathcal{P}} s^i(\tau)_u\right) \frac{\alpha^2 \beta}{4} \zeta_P \|\mu\|^2 + \sum_{u \in \mathcal{I}} s^i(\tau)_u \frac{1-\beta}{4} \zeta_R \|\mu\|^2\right) \quad (8)$$

$$\Delta\tilde{\lambda}_c(\tau) \geq \frac{\alpha\eta}{n} \frac{\|\nu\|}{\|\mathbf{d}\|} \sum_{i \in \mathcal{N}_c} (-\ell_i'(\tau)) \cdot \sum_{p \in \mathcal{P}} s^i(\tau)_p \left(-\left(1 - s^i(\tau)_p\right)\left(\frac{1-\beta}{2}\zeta_R + \alpha\right) c_2 \|\mu\| \sqrt{\log(Tn/\delta)}\right.$$

$$\left. + \left(1 - \sum_{u \in \mathcal{P}} s^i(\tau)_u\right)\left(\frac{\alpha^2 \beta}{4}\zeta_P \|\mu\|^2 - \frac{(1+1/C')^2}{nT} \operatorname{Tr}(\Sigma)\right) + \sum_{u \in \mathcal{R}} s^i(\tau)_u \frac{1-\beta}{4} \zeta_R \|\mu\|^2\right) \quad (9)$$

For $i \in \mathcal{C}, r \in \mathcal{R}, v \in \mathcal{I}$, we have

$$\Delta\rho_{i,r}(\tau) = \frac{\eta}{n}(-\ell_i'(\tau)) \cdot s^i(\tau)_r \left(y^i \gamma_r^i - \sum_{u \in [T]} s^i(\tau)_u y^i \gamma_u^i\right)$$

$$\leq \frac{\eta}{n} \frac{\|\nu\|}{\|\mathbf{d}\|}(-\ell_i'(\tau)) \cdot s^i(\tau)_r \left(\left(1 - \sum_{u \in \mathcal{R}} s^i(\tau)_u\right) \frac{1-\beta}{4} \zeta_R \|\mu\|^2\right.$$

$$\left. + \left(1 - s^i(\tau)_r\right)\left(\left((1-\beta)\zeta_R + \frac{(2\alpha-1)\beta}{2}\zeta_P + 1\right) c_2 \|\mu\| \sqrt{\log(Tn/\delta)} + \frac{(1+1/C')^2}{nT} \operatorname{Tr}(\Sigma)\right)\right) \quad (10)$$

$$\Delta\rho_{i,r}(\tau) \geq \frac{\eta}{n} \frac{\|\nu\|}{\|\mathbf{d}\|}(-\ell_i'(\tau)) \cdot s^i(\tau)_r \left(\left(1 - \sum_{u \in \mathcal{R}} s^i(\tau)_u\right) \frac{1-\beta}{4} \zeta_R \|\mu\|^2\right.$$

$$\left. - \left(1 - s^i(\tau)_r\right)\left(\left((1-\beta)\zeta_R + \frac{(2\alpha-1)\beta}{2}\zeta_P + 1\right) c_2 \|\mu\| \sqrt{\log(Tn/\delta)} + \frac{(1+1/C')^2}{nT} \operatorname{Tr}(\Sigma)\right)\right) \quad (11)$$

$$\Delta\rho_{i,v}(\tau) = \frac{\eta}{n}(-\ell_i'(\tau)) \cdot s^i(\tau)_v \left(y^i \gamma_v^i - \sum_{u \in [T]} s^i(\tau)_u y^i \gamma_u^i\right)$$

$$\leq \frac{\eta}{n} \frac{\|\nu\|}{\|\mathbf{d}\|}(-\ell_i'(\tau)) \cdot s^i(\tau)_v \left(-\sum_{u \in \mathcal{R}} s^i(\tau)_u \frac{1-\beta}{4} \zeta_R \|\mu\|^2\right.$$

$$\left. + \left(1 - s^i(\tau)_v\right)\left(\left((1-\beta)\zeta_R + \alpha\beta\zeta_P + \frac{1}{2}\right) c_2 \|\mu\| \sqrt{\log(Tn/\delta)} + \frac{(1+1/C')^2}{nT} \operatorname{Tr}(\Sigma)\right)\right) \quad (12)$$

$$\Delta\rho_{i,v}(\tau) \geq \frac{\eta}{n}\frac{\|\nu\|}{\|\mathbf{d}\|}(-\ell_i'(\tau)) \cdot s^i(\tau)_v\Bigg(-\sum_{u\in\mathcal{R}}s^i(\tau)_u\frac{1-\beta}{4}\zeta_R\|\mu\|^2$$

$$-\left(1-s^i(\tau)_v\right)\left(\left((1-\beta)\zeta_R + \alpha\beta\zeta_P + \frac{1}{2}\right)c_2\|\mu\|\sqrt{\log(Tn/\delta)} + \frac{(1+1/C')^2}{nT}\mathrm{Tr}(\Sigma)\right)\Bigg)$$

For $i \in \mathcal{N}, r \in \mathcal{R}, p \in \mathcal{P}, v \in \mathcal{I}$, we have

$$\Delta\rho_{i,r} = \frac{\eta}{n}(-\ell_i'(\tau)) \cdot s^i(\tau)_r\left(y^i\gamma_r^i - \sum_{u\in[T]}s^i(\tau)_u y^i\gamma_u^i\right)$$

$$\leq \frac{\eta}{n}\frac{\|\nu\|}{\|\mathbf{d}\|}(-\ell_i'(\tau)) \cdot s^i(\tau)_r\Bigg(-\left(1-\sum_{u\in\mathcal{R}}s^i(\tau)_u\right)\frac{1-\beta}{4}\zeta_R\|\mu\|^2 - \sum_{u\in\mathcal{P}}s^i(\tau)_u\frac{\alpha^2\beta}{4}\zeta_P\|\mu\|^2$$

$$+ \left(1-s^i(\tau)_r\right)\left(\left((1-\beta)\zeta_R + \frac{(2\alpha-1)\beta}{2}\zeta_P + \frac{1+\alpha}{2}\right)c_2\|\mu\|\sqrt{\log(Tn/\delta)} + \frac{(1+1/C')^2}{nT}\mathrm{Tr}(\Sigma)\right)\Bigg) \quad (13)$$

$$\Delta\rho_{i,r} \geq -\frac{\eta}{n}\frac{\|\nu\|}{\|\mathbf{d}\|}(-\ell_i'(\tau)) \cdot s^i(\tau)_r\Bigg(\left(1-\sum_{u\in\mathcal{R}}s^i(\tau)_u\right)\frac{1-\beta}{4}\zeta_R\|\mu\|^2 + \sum_{u\in\mathcal{P}}s^i(\tau)_u\frac{\alpha^2\beta}{4}\zeta_P\|\mu\|^2$$

$$+ \left(1-s^i(\tau)_r\right)\left(\left((1-\beta)\zeta_R + \frac{(2\alpha-1)\beta}{2}\zeta_P + \frac{1+\alpha}{2}\right)c_2\|\mu\|\sqrt{\log(Tn/\delta)} + \frac{(1+1/C')^2}{nT}\mathrm{Tr}(\Sigma)\right)\Bigg)$$

$$\Delta\rho_{i,p} = \frac{\eta}{n}(-\ell_i'(\tau)) \cdot s^i(\tau)_p\left(y^i\gamma_p^i - \sum_{u\in[T]}s^i(\tau)_u y^i\gamma_u^i\right)$$

$$\leq \frac{\eta}{n}\frac{\|\nu\|}{\|\mathbf{d}\|}(-\ell_i'(\tau)) \cdot s^i(\tau)_p\Bigg((1-s^i(\tau)_p)\left(\frac{1-\beta}{2}\zeta_R + \alpha\right)c_2\|\mu\|\sqrt{\log(Tn/\delta)}\Bigg)$$

$$+ \left(1-\sum_{u\in\mathcal{P}}s^i(\tau)_u\right)\frac{\alpha^2\beta}{4}\zeta_P\|\mu\|^2 + \sum_{u\in\mathcal{R}}s^i(\tau)_u\frac{1-\beta}{4}\zeta_R\|\mu\|^2$$

$$\Delta\rho_{i,p} \geq \frac{\eta}{n}\frac{\|\nu\|}{\|\mathbf{d}\|}(-\ell_i'(\tau)) \cdot s^i(\tau)_p\Bigg(-\left(1-s^i(\tau)_p\right)\left(\frac{1-\beta}{2}\zeta_R + \alpha\right)c_2\|\mu\|\sqrt{\log(Tn/\delta)}$$

$$+ \left(1-\sum_{u\in\mathcal{P}}s^i(\tau)_u\right)\left(\frac{\alpha^2\beta}{4}\zeta_P\|\mu\|^2 - \frac{(1+1/C')^2}{nT}\mathrm{Tr}(\Sigma)\right) + \sum_{u\in\mathcal{R}}s^i(\tau)_u\frac{1-\beta}{4}\zeta_R\|\mu\|^2\Bigg)$$

$$\Delta\rho_{i,v} = \frac{\eta}{n}(-\ell_i'(\tau)) \cdot s^i(\tau)_v\left(y^i\gamma_v^i - \sum_{u\in[T]}s^i(\tau)_u y^i\gamma_u^i\right)$$

$$\leq \frac{\eta}{n}\frac{\|\nu\|}{\|\mathbf{d}\|}(-\ell_i'(\tau)) \cdot s^i(\tau)_v\Bigg(\sum_{u\in\mathcal{R}}s^i(\tau)_u\frac{1-\beta}{4}\zeta_R\|\mu\|^2 - \sum_{u\in\mathcal{P}}s^i(\tau)_u\frac{\alpha^2\beta}{4}\zeta_P$$

$$+ \left(1-s^i(\tau)_v\right)\left(\left((1-\beta)\zeta_R + \alpha\beta\zeta_P + \frac{\alpha}{2}\right)c_2\|\mu\|\sqrt{\log(Tn/\delta)} + \frac{(1+1/C')^2}{nT}\mathrm{Tr}(\Sigma)\right)\Bigg) \quad (14)$$

$$\Delta\rho_{i,v} \geq \frac{\eta}{n} \frac{\|\nu\|}{\|\mathbf{d}\|} (-\ell'_i(\tau)) \cdot s^i(\tau)_v \left( \sum_{u \in \mathcal{R}} s^i(\tau)_u \frac{1-\beta}{4} \zeta_R \|\mu\|^2 - \sum_{u \in \mathcal{P}} s^i(\tau)_u \frac{\alpha^2 \beta}{4} \zeta_P \right.$$
$$\left. - (1 - s^i(\tau)_v) \left( \left( (1-\beta)\zeta_R + \alpha\beta\zeta_P + \frac{\alpha}{2} \right) c_2 \|\mu\| \sqrt{\log(Tn/\delta)} + \frac{(1+1/C')^2}{nT} \operatorname{Tr}(\Sigma) \right) \right)$$

$\square$

**Lemma A.7.** Suppose that the norm of the linear head $\nu \propto \mathbf{d}$ and $\frac{\|\nu\|}{\|\mathbf{d}\|}$ scales as $\Theta(1/\|\mu\|^2)$. There exists an absolute constant $C_\ell > 0$ such that on a good run, we have for all time step $\tau \geq 0$,

$$\max_{i,j \in [n]} \frac{\ell'_i(\tau)}{\ell'_j(\tau)} \leq C_\ell.$$

*Proof of Lemma A.7.* Recall that the derivative of the loss function is given by

$$-\ell'_i(\tau) = \frac{1}{1 + \exp\left( \sum_{t \in [T]} s^i(\tau)_t \gamma^i_t \right)}$$

for any $i \in [n], \tau \geq 0$. One a good run, for all $i \in [n], k \in \mathcal{N}, r \in \mathcal{R}, p \in \mathcal{P}, v \in \mathcal{I}$, we leverage Assumption (A2) and Lemma A.5 to get the following

$$\left| \gamma^i_r \right| \leq \frac{1-\beta}{4} \zeta_R + \left( \frac{(1-\beta)}{2} \zeta_R + \frac{(\alpha-1)\beta}{2} \zeta_P + \frac{1}{2} \right) \frac{c_2}{CT^2} + \frac{(1+1/C')^2}{CT}$$
$$\left| \gamma^i_v \right| \leq \left( \frac{(1-\beta)}{2} \zeta_R + \frac{\alpha\beta}{2} \zeta_P \right) \frac{c_2}{CT^2} + \frac{(1+1/C')^2}{CT}$$
$$\left| \gamma^k_p \right| \leq \frac{\alpha^2 \beta}{4} \zeta_P + \left( \frac{\alpha}{2} - \frac{\alpha\beta}{2} \zeta_P \right) \frac{c_2}{CT^2}$$

Therefore there exists some constant $c > 0$ such that $\left| \gamma^i_t \right| < c, \forall i \in [n], \forall t \in [T]$. Since $-\ell'_i$ is monotonically decreasing, we have

$$\frac{1}{1 + \exp(c)} \leq -\ell'_i(\tau) \leq \frac{1}{1 + \exp(-c)}.$$

This leads to $C_\ell = \frac{1+\exp(c)}{1+\exp(-c)}$. $\square$

The following lemma show that given a sufficiently small step size, the softmax probabilities do not change significantly in a single step of gradient descent.

**Lemma A.8.** Suppose that the norm of the linear head $\nu \propto \mathbf{d}$ and $\frac{\|\nu\|}{\|\mathbf{d}\|}$ scales as $\Theta(1/\|\mu\|^2)$, and the step size of the gradient descent satisfies Assumption (A3). Then the probability assigned to each token only changes at most by a constant factor,

$$\forall \tau \geq 0, \ \forall i \in [n], \ \forall t \in [T], \ \frac{1}{2} s^i(\tau)_t < s^i(\tau+1)_t < 2s^i(\tau)_t.$$

*Proof of Lemma A.8.* By the updated of gradient descent, we have

$$s^i(\tau+1)_t = \frac{\exp\left( {x^i_t}^\top \mathbf{W}^\top \mathbf{p}(\tau) \right) \exp\left( {x^i_t}^\top \mathbf{W}^\top \left( -\eta \nabla_\mathbf{p} \widehat{L}(\mathbf{p}(\tau)) \right) \right)}{\sum_{u \in [T]} \exp\left( {x^i_u}^\top \mathbf{W}^\top \mathbf{p}(\tau) \right) \exp\left( {x^i_u}^\top \mathbf{W}^\top \left( -\eta \nabla_\mathbf{p} \widehat{L}(\mathbf{p}(\tau)) \right) \right)}$$

Consider a consecutive time steps, we have

$$\frac{\exp\left({\mathbf{x}_t^i}^\top \mathbf{W}^\top \left(-\eta \nabla_{\mathrm{p}} \widehat{L}(\mathbf{p}(\tau))\right)\right)}{\max_{u \in [T]}\left\{\exp\left({\mathbf{x}_u^i}^\top \mathbf{W}^\top \left(-\eta \nabla_{\mathrm{p}} \widehat{L}(\mathbf{p}(\tau))\right)\right)\right\}} \leq \frac{s^i(\tau+1)_t}{s^i(\tau)_t} \leq \frac{\exp\left({\mathbf{x}_t^i}^\top \mathbf{W}^\top \left(-\eta \nabla_{\mathrm{p}} \widehat{L}(\mathbf{p}(\tau))\right)\right)}{\min_{u \in [T]}\left\{\exp\left({\mathbf{x}_u^i}^\top \mathbf{W}^\top \left(-\eta \nabla_{\mathrm{p}} \widehat{L}(\mathbf{p}(\tau))\right)\right)\right\}}$$

Therefore the proof is completed by showing that

$$\forall t \in [T], \forall u \in [T], \frac{1}{2} \leq \exp\left(\left(\mathbf{x}_t^i - \mathbf{x}_u^i\right)^\top \mathbf{W}^\top \left(-\eta \nabla_{\mathrm{p}} \widehat{L}(\mathbf{p}(\tau))\right)\right) \leq 2.$$

To analyze the term inside the exponent, we have

$$\left|\left(\mathbf{x}_t^i - \mathbf{x}_u^i\right)^\top \mathbf{W}^\top \left(-\eta \nabla_{\mathrm{p}} \widehat{L}(\mathbf{p}(\tau))\right)\right| \leq 2 \max_{t \in [T]} \left\{\left|{\mathbf{x}_t^i}^\top \mathbf{W}^\top \left(-\eta \nabla_{\mathrm{p}} \widehat{L}(\mathbf{p}(\tau))\right)\right|\right\} \tag{15}$$

Following from Lemma A.4, for $t := r \in \mathcal{R}$, we have

$${\mathbf{x}_r^i}^\top \mathbf{W}^\top \left(-\eta \nabla_{\mathrm{p}} \widehat{L}(\mathbf{p}(\tau))\right)$$

$$= \left(\mu_{y^i} + \epsilon_r^i\right)^\top \left(\Delta\lambda_{+1}(\tau)\mu_{+1} + \Delta\lambda_{-1}(\tau)\mu_{-1} + \Delta\tilde{\lambda}_{+1}(\tau)\tilde{\mu}_{+1} + \Delta\tilde{\lambda}_{-1}(\tau)\tilde{\mu}_{-1} + \sum_{i \in [n]}\sum_{t \in [T]\setminus\mathcal{P}} \Delta\rho_{i,t}(\tau)\epsilon_t^i\right)$$

$$\leq \Delta\lambda_{y^i}(\tau)\|\mu\|^2 + \left(\Delta\lambda_{+1}(\tau) + \Delta\lambda_{-1}(\tau) + \Delta\tilde{\lambda}_{+1}(\tau) + \Delta\tilde{\lambda}_{-1}(\tau) + \sum_{i \in [n]}\sum_{t \in [T]\setminus\mathcal{P}} \Delta\rho_{i,t}(\tau)\right) c_2 \|\mu\| \sqrt{\log(Tn/\delta)}$$

$$+ \sum_{i \in [n]}\sum_{t \in [T]\setminus\mathcal{P}} \Delta\rho_{i,t}(\tau) c_1 \sqrt{\mathrm{Tr}(\Sigma)} \log(T/\delta) + \Delta\rho_{i,r}(1 + 1/C')^2 \mathrm{Tr}(\Sigma)$$

For $t := p \in \mathcal{P}$, we have

$${\mathbf{x}_p^i}^\top \mathbf{W}^\top \left(-\eta \nabla_{\mathrm{p}} \widehat{L}(\mathbf{p}(\tau))\right)$$

$$= \alpha\tilde{\mu}_{-y^i}^\top \left(\Delta\lambda_{+1}(\tau)\mu_{+1} + \Delta\lambda_{-1}(\tau)\mu_{-1} + \Delta\tilde{\lambda}_{+1}(\tau)\tilde{\mu}_{+1} + \Delta\tilde{\lambda}_{-1}(\tau)\tilde{\mu}_{-1} + \sum_{i \in [n]}\sum_{t \in [T]\setminus\mathcal{P}} \Delta\rho_{i,t}(\tau)\epsilon_t^i\right)$$

$$\leq \alpha\Delta\tilde{\lambda}_{-y^i}(\tau)\|\mu\|^2 + \alpha \sum_{i \in [n]}\sum_{t \in [T]\setminus\mathcal{P}} \Delta\rho_{i,t}(\tau) c_2 \|\mu\| \sqrt{\log(Tn/\delta)}$$

Consider we assume $\frac{\|\nu\|}{\|\mathbf{d}\|}$ scales as $\Theta(1/\|\mu\|^2)$, then Equation (6) gives us that

$$\Delta\lambda_c(\tau) \leq \frac{\eta}{2}\left(\frac{1-\beta}{4}\zeta_R + \left((1-\beta)\zeta_R + \frac{(2\alpha-1)\beta}{2}\zeta_P + 1\right)\frac{c_2}{CT^2} + \frac{(1+1/C')^2}{CT}\right) \lesssim \eta \tag{16}$$

where the last line for sufficiently large $C$ and Assumption (A4) so that $\frac{\alpha\beta\zeta_P}{CT^2} \leq 1$. Similarly, Equation (8) gives us that

$$\Delta\tilde{\lambda}_c(\tau) \leq \frac{\alpha\beta\eta}{2}\left(\frac{1-\beta}{4}\zeta_R + \frac{\alpha^2\beta}{4}\zeta_P + \left(\frac{1-\beta}{2}\zeta_R + \alpha\right)\frac{c_2}{CT^2}\right) \lesssim \alpha^3\beta^2\eta$$

where the last line is because Assumption (A3) so that $\zeta_R \lesssim \alpha^2\beta\zeta_P$.

Equation (10) and (12) tell us that for $i \in \mathcal{C}$

$$
\begin{aligned}
\sum_{r \in \mathcal{R}} |\Delta \rho_{i,r}(\tau)| &\leq \frac{\eta}{n}(-\ell_i'(\tau)) \cdot \sum_{r \in \mathcal{R}} s^i(\tau)_r \left| \left(1 - \sum_{u \in \mathcal{R}} s^i(\tau)_u\right) \frac{1-\beta}{4} \zeta_R \right. \\
&\quad + \left. \left(1 - s^i(\tau)_r\right) \left(\left((1-\beta)\zeta_R + \frac{(2\alpha-1)\beta}{2}\zeta_P + 1\right) c_2 \frac{\sqrt{\log(Tn/\delta)}}{\|\mu\|} + \frac{(1+1/C')^2}{nT} \frac{\text{Tr}(\Sigma)}{\|\mu\|^2}\right) \right| \\
&\leq \frac{\eta}{n} \left(\frac{1-\beta}{4}\zeta_R + \left((1-\beta)\zeta_R + \frac{(2\alpha-1)\beta}{2}\zeta_P + 1\right)\frac{c_2}{CT^2} + \frac{(1+1/C')^2}{CT}\right) \qquad \text{(Assumption (A2))} \\
&\lesssim \frac{\eta}{n} \qquad\qquad\qquad\qquad\qquad\qquad\qquad\qquad\qquad\qquad\qquad\qquad \text{(For sufficiently large } C.)
\end{aligned}
$$

$$
\begin{aligned}
\sum_{v \in \mathcal{I}} |\Delta \rho_{i,v}(\tau)| &\leq \frac{\eta}{n}(-\ell_i'(\tau)) \cdot \sum_{v \in \mathcal{I}} s^i(\tau)_v \left| \left(-\sum_{u \in \mathcal{R}} s^i(\tau)_u \frac{1-\beta}{4} \zeta_R \right. \right. \\
&\quad + \left. \left. \left(1 - s^i(\tau)_v\right) \left(\left((1-\beta)\zeta_R + \alpha\beta\zeta_P + \frac{1}{2}\right) c_2 \frac{\sqrt{\log(Tn/\delta)}}{\|\mu\|} + \frac{(1+1/C')^2}{nT} \frac{\text{Tr}(\Sigma)}{\|\mu\|^2}\right)\right) \right| \\
&\leq \frac{\eta}{n} \left(\frac{1-\beta}{4}\zeta_R + \left((1-\beta)\zeta_R + \frac{(2\alpha-1)\beta}{2}\zeta_P + 1\right)\frac{c_2}{CT^2} + \frac{(1+1/C')^2}{CT}\right) \qquad \text{(Assumption (A2))} \\
&\lesssim \frac{\eta}{n} \qquad\qquad\qquad\qquad\qquad\qquad\qquad\qquad\qquad\qquad\qquad\qquad \text{(For sufficiently large } C.)
\end{aligned}
$$

Similarly, for $i \in \mathcal{N}$, (13) and (14) tell us that

$$
\begin{aligned}
\sum_{r \in \mathcal{R}} |\Delta \rho_{i,r}| &\leq \frac{\eta}{n}(-\ell_i'(\tau)) \cdot \sum_{r \in \mathcal{R}} s^i(\tau)_r \left| \left(-\left(1 - \sum_{u \in \mathcal{R}} s^i(\tau)_u\right) \frac{1-\beta}{4} \zeta_R - \sum_{u \in \mathcal{P}} s^i(\tau)_u \frac{\alpha^2\beta}{4}\zeta_P \right. \right. \\
&\quad + \left. \left. \left(1 - s^i(\tau)_r\right) \left(\left((1-\beta)\zeta_R + \frac{(2\alpha-1)\beta}{2}\zeta_P + \frac{1+\alpha}{2}\right) c_2 \frac{\sqrt{\log(Tn/\delta)}}{\|\mu\|} + \frac{(1+1/C')^2}{nT} \frac{\text{Tr}(\Sigma)}{\|\mu\|^2}\right)\right) \right| \\
&\lesssim \frac{\alpha^2\beta\zeta_P\eta}{n} \qquad\qquad\qquad\qquad\qquad\qquad\qquad\qquad\qquad\qquad\qquad\qquad\qquad (\alpha^2\beta\zeta_P \geq \zeta_R)
\end{aligned}
$$

$$
\begin{aligned}
\sum_{v \in \mathcal{I}} |\Delta \rho_{i,v}| &\leq \frac{\eta}{n}(-\ell_i'(\tau)) \cdot \sum_{v \in \mathcal{I}} s^i(\tau)_v \left| \left(\sum_{u \in \mathcal{R}} s^i(\tau)_u \frac{1-\beta}{4} \zeta_R - \sum_{u \in \mathcal{P}} s^i(\tau)_u \frac{\alpha^2\beta}{4}\zeta_P \right. \right. \\
&\quad + \left. \left. \left(1 - s^i(\tau)_v\right) \left(\left((1-\beta)\zeta_R + \alpha\beta\zeta_P + \frac{\alpha}{2}\right) c_2 \frac{\sqrt{\log(Tn/\delta)}}{\|\mu\|} + \frac{(1+1/C')^2}{nT} \frac{\text{Tr}(\Sigma)}{\|\mu\|^2}\right)\right) \right| \\
&\lesssim \frac{\alpha^2\beta\zeta_P\eta}{n}
\end{aligned}
$$

From Equation (10), (12), (13) and (14), we have

$$
\begin{aligned}
\sum_{i \in [n], t \in [T] \setminus \mathcal{P}} |\Delta \rho_{i,t}(\tau)| &= \sum_{i \in \mathcal{C}, r \in \mathcal{R}} |\Delta \rho_{i,r}(\tau)| + \sum_{i \in \mathcal{C}, v \in \mathcal{I}} |\Delta \rho_{i,v}(\tau)| + \sum_{i \in \mathcal{N}, r \in \mathcal{R}} |\Delta \rho_{i,r}(\tau)| + \sum_{i \in \mathcal{N}, v \in \mathcal{I}} |\Delta \rho_{i,v}(\tau)| \\
&\lesssim (1-\beta)^2 \eta + \alpha^2\beta^2\zeta_P\eta \qquad\qquad \text{(Assumption (A4) gives us that } \alpha^2\beta^2\zeta_P \leq 1) \\
&\lesssim \eta \qquad\qquad\qquad\qquad\qquad\qquad\qquad\qquad\qquad\qquad\qquad\qquad\qquad\qquad\qquad (17)
\end{aligned}
$$

As a result, we have

$$\mathbf{x}_r^{i\top}\mathbf{W}^\top\left(-\eta\nabla_\mathrm{p}\widehat{L}(\mathbf{p}(\tau))\right) \lesssim \left(1 + \frac{\alpha^3\beta^2}{T^2}\right)\eta\|\mu\|^2$$

$$\mathbf{x}_p^{i\top}\mathbf{W}^\top\left(-\eta\nabla_\mathrm{p}\widehat{L}(\mathbf{p}(\tau))\right) \lesssim \left(\alpha^4\beta^2 + \frac{\alpha}{T^2}\right)\eta\|\mu\|^2$$

Therefore, choosing a sufficiently small step size $\eta$ by setting a sufficiently large $C$ can make Equation (15) smaller than $\log(2)$, thus concludes the proof.

We end with a remark that if $\mathrm{Tr}(\Sigma) < n^2 T^2 \log(Tn/\delta)^2$, then all the term $\frac{\mathrm{Tr}(\Sigma)}{nT\|\mu\|^2}$ would be replaced by $\frac{\sqrt{\mathrm{Tr}(\Sigma)}\log(Tn/\delta)}{\|\mu\|^2}$, which can be further upper bounded by $\frac{1}{CT^2}$ from Assumption (A2) and therefore all the discussion remains hold.

$\square$

The following lemma shows that for any clean data, with high probability, for any time step, the attention probability from the relevant token would dominate.

**Lemma A.9.** Suppose that Assumption 1 and 2 hold, and the norm of the linear head $\nu \propto \mathrm{d}$ and $\frac{\|\nu\|}{\|\mathrm{d}\|}$ scales as $\Theta(1/\|\mu\|^2)$. For any clean data $i \in \mathcal{C}$, on a good run, for all time step $\tau \geq 0$ and all irrelevant token $v \in \mathcal{I}$, we have

$$s^i(\tau)_v \leq \max_{r\in\mathcal{R}}\left\{s^i(\tau)_t\right\}$$

*Proof of Lemma A.9.* The proof is via induction. The inequality holds at initialization as all the elements are equal to $1/T$. Assuming $s^i(\tau)_v \leq \max_{r\in\mathcal{R}}\left\{s^i(\tau)_r\right\}$ and we prove for $s^i(\tau+1)_v \leq \max_{r\in\mathcal{R}}\left\{s^i(\tau+1)_r\right\}$. For $v \in \mathcal{I}$, we have

$$
\begin{aligned}
\frac{\max_{r\in\mathcal{R}}\left\{s^i(\tau+1)_r\right\}}{s^i(\tau+1)_v} &= \frac{\max_{r\in\mathcal{R}}\left\{\exp\left(\mathbf{x}_r^{i\top}\mathbf{W}^\top\mathbf{p}(\tau+1)\right)\right\}}{\exp\left(\mathbf{x}_v^{i\top}\mathbf{W}^\top\mathbf{p}(\tau+1)\right)} \\
&= \frac{\max_{r\in\mathcal{R}}\left\{\exp\left(\mathbf{x}_r^{i\top}\mathbf{W}^\top\mathbf{p}(\tau)\right)\exp\left(\mathbf{x}_r^{i\top}\mathbf{W}^\top\left(-\eta\nabla_\mathrm{p}\widehat{L}(\mathbf{p}(\tau))\right)\right)\right\}}{\exp\left(\mathbf{x}_v^{i\top}\mathbf{W}^\top\mathbf{p}(\tau)\right)\exp\left(\mathbf{x}_v^{i\top}\mathbf{W}^\top\left(-\eta\nabla_\mathrm{p}\widehat{L}(\mathbf{p}(\tau))\right)\right)} \\
&\geq \frac{s^i(\tau)_{r'}}{s^i(\tau)_v}\cdot\exp\left(\left(\mathbf{x}_{r'}^i - \mathbf{x}_v^i\right)^\top\mathbf{W}^\top\left(-\eta\nabla_\mathrm{p}\widehat{L}(\mathbf{p}(\tau))\right)\right)
\end{aligned}
$$

where $r' = \arg\max_{r\in\mathcal{R}}\left\{s^i(\tau)_r\right\}$. As long as we can show $\left(\mathbf{x}_{r'}^i - \mathbf{x}_v^i\right)^\top\mathbf{W}^\top\left(-\eta\nabla_\mathrm{p}\widehat{L}(\mathbf{p}(\tau))\right) \geq 0$, then $\frac{\max_{r\in\mathcal{R}}\left\{s^i(\tau+1)_r\right\}}{s^i(\tau+1)_v} \geq \frac{s^i(\tau)_{r'}}{s^i(\tau)_v}$, which proves via induction.

We now consider two cases. If $\sum_{r\in\mathcal{R}}s^i(\tau)_r \geq 1 - \frac{1}{4T}$, then the probability of not selecting the relevant token is less than $\frac{1}{4T}$. Apply Lemma A.8, the probability of not selecting the relevant tokens after a single step of gradient descent is at most $\frac{1}{2T}$, and therefore $\sum_{r\in\mathcal{R}}s^i(\tau+1)_r \geq 1 - \frac{1}{2T}$, leading to

$$s^i(\tau+1)_v \leq \frac{1}{2T} \leq \frac{1}{|\mathcal{R}|}\left(1 - \frac{1}{2T}\right) \leq \max_{r\in\mathcal{R}}\left\{s^i(\tau+1)_r\right\}.$$

Now we only need to consider the situation where $\sum_{r\in\mathcal{R}}s^i(\tau)_r \leq 1 - \frac{1}{4T}$. Note that $s^i(\tau)_{r'} \geq 1/T$ also holds due to the definition of $r'$. We have

$$
\begin{aligned}
&\left(\mathbf{x}_{r'}^i - \mathbf{x}_v^i\right)^\top\mathbf{W}^\top\left(-\eta\nabla_\mathrm{p}\widehat{L}(\mathbf{p}(\tau))\right) \\
&= \left(\mu_{y^i}^i + \epsilon_{r'}^i - \epsilon_v^i\right)^\top\left(\Delta\lambda_{+1}(\tau)\mu_{+1} + \Delta\lambda_{-1}(\tau)\mu_{-1} + \Delta\tilde{\lambda}_{+1}(\tau)\tilde{\mu}_{+1} + \Delta\tilde{\lambda}_{-1}(\tau)\tilde{\mu}_{-1} + \sum_{i\in[n]}\sum_{t\in[T]\setminus\mathcal{P}}\Delta\rho_{i,t}(\tau)\epsilon_t^i\right) \\
&\gtrsim \left(\Delta\rho_{i,r'}(\tau) - \Delta\rho_{i,v}(\tau)\right)\mathrm{Tr}(\Sigma) + \Delta\lambda_{y^i}(\tau)\|\mu\|^2 - \sum_{k\in[n],u\in[T]\setminus\mathcal{P}}|\Delta\rho_{k,u}(\tau)|c_1\sqrt{\mathrm{Tr}(\Sigma)}\log(Tn/\delta)
\end{aligned}
$$

$$- \left( 2\left|\Delta\lambda_{+1}(\tau)\right| + 2\left|\Delta\lambda_{-1}(\tau)\right| + 2\left|\Delta\tilde{\lambda}_{+1}(\tau)\right| + 2\left|\Delta\tilde{\lambda}_{-1}(\tau)\right| + \sum_{\substack{k\in[n],\\ u\in[T]\setminus\mathcal{P}}} \left|\Delta\rho_{k,u}(\tau)\right| \right) \|\mu\| \sqrt{\log\left(Tn/\delta\right)}$$

where the last line holds from Lemma A.4. We will show the above equation is positive by controlling each term separately.

Note that $\frac{\|\nu\|}{\|\mathrm{d}\|} = \Theta\left(1/\|\mu\|^2\right)$, leverage Equation (11) and (12) gives us that for $i\in\mathcal{C}$,

$$(\Delta\rho_{i,r'}(\tau) - \Delta\rho_{i,v}(\tau))\,\mathrm{Tr}(\Sigma)$$

$$\geq \mathrm{Tr}(\Sigma)\frac{\eta}{n\|\mu\|^2}(-\ell_i'(\tau)) \cdot \left( s^i(\tau)_{r'}\left( \left(1-\sum_{u\in\mathcal{R}} s^i(\tau)_u\right)\frac{1-\beta}{4}\zeta_R\|\mu\|^2\right.\right.$$

$$- \left(1 - s^i(\tau)_{r'}\right)\left(\left((1-\beta)\zeta_R+\frac{(2\alpha-1)\beta}{2}\zeta_P+1\right)c_2\|\mu\|\sqrt{\log\left(Tn/\delta\right)}+\frac{(1+1/C')^2}{nT}\,\mathrm{Tr}(\Sigma)\right)\right)$$

$$- s^i(\tau)_v\left(-\sum_{u\in\mathcal{R}} s^i(\tau)_u\frac{1-\beta}{4}\zeta_R\|\mu\|^2\right.$$

$$\left.\left.+ \left(1-s^i(\tau)_v\right)\left(\left((1-\beta)\zeta_R+\alpha\beta\zeta_P+\frac{1}{2}\right)c_2\|\mu\|\sqrt{\log\left(Tn/\delta\right)}+\frac{(1+1/C')^2}{nT}\,\mathrm{Tr}(\Sigma)\right)\right)\right)$$

$$\geq \mathrm{Tr}(\Sigma)\frac{\eta}{n\|\mu\|^2}(-\ell_i'(\tau)) \cdot \left( s^i(\tau)_{r'}\left(1-\sum_{u\in\mathcal{R}} s^i(\tau)_u\right)\frac{1-\beta}{4}\zeta_R\|\mu\|^2\right.$$

$$\left.- 2s^i(\tau)_{r'}\left(\left((1-\beta)\zeta_R+\alpha\beta\zeta_P+1\right)c_2\|\mu\|\sqrt{\log\left(Tn/\delta\right)}+\frac{(1+1/C')^2}{nT}\,\mathrm{Tr}(\Sigma)\right)\right)$$

$$(s^i(\tau)_{r'}(1-s^i(\tau)_{r'}) + s^i(\tau)_v(1-s^i(\tau)_v) \leq 2s^i(\tau)_{r'})$$

$$\geq \mathrm{Tr}(\Sigma)\frac{\eta}{n\|\mu\|^2}(-\ell_i'(\tau)) \cdot s^i(\tau)_{r'}\left(1-\sum_{u\in\mathcal{R}} s^i(\tau)_u\right)\left(\frac{1-\beta}{4}\zeta_R\|\mu\|^2\right.$$

$$\left.- 8T\left(\left((1-\beta)\zeta_R+\alpha\beta\zeta_P+1\right)c_2\|\mu\|\sqrt{\log\left(Tn/\delta\right)}+\frac{(1+1/C')^2}{nT}\,\mathrm{Tr}(\Sigma)\right)\right)$$

$$\left(2s^i(\tau)_{r'} \leq 8Ts^i(\tau)_{r'}\left(1-\textstyle\sum_{u\in\mathcal{R}} s^i(\tau)_u\right) \text{ as } 1-\textstyle\sum_{u\in\mathcal{R}} s^i(\tau)_u \geq \tfrac{1}{4T}.\right)$$

$$\geq \mathrm{Tr}(\Sigma)\frac{\eta}{n}(-\ell_i'(\tau))\frac{1}{4T^2} \cdot \left(\frac{1-\beta}{4}\zeta_R - 8\left(\left((1-\beta)\zeta_R+\alpha\beta\zeta_P+1\right)c_2\frac{T\sqrt{\log\left(Tn/\delta\right)}}{\|\mu\|}+\frac{(1+1/C')^2}{n}\frac{\mathrm{Tr}(\Sigma)}{\|\mu\|^2}\right)\right)$$

$$\left(s^i(\tau)_{r'}\left(1-\textstyle\sum_{u\in\mathcal{R}} s^i(\tau)_u\right) \geq \tfrac{1}{4T^2}.\right)$$

$$\geq \mathrm{Tr}(\Sigma)\frac{\eta}{n}(-\ell_i'(\tau))\frac{1}{4T^2} \cdot \left(\frac{1-\beta}{4}\zeta_R - 8\left(\left((1-\beta)\zeta_R+\alpha\beta\zeta_P+1\right)\frac{c_2}{CT}+\frac{(1+1/C')^2}{C}\right)\right)$$

Similarly, Equation (7) gives us that

$$\Delta\lambda_c(\tau) \geq \frac{\eta}{n}\sum_{i\in\mathcal{C}_c}(-\ell_i'(\tau))\sum_{r\in\mathcal{R}} s^i(\tau)_r\left( \left(1-\sum_{u\in\mathcal{R}} s^i(\tau)_u\right)\frac{1-\beta}{4}\zeta_R\right.$$

$$\left.- \left(1-s^i(\tau)_r\right)\left(\left((1-\beta)\zeta_R+\frac{(2\alpha-1)\beta}{2}\zeta_P+1\right)c_2\left(\frac{\sqrt{\log\left(Tn/\delta\right)}}{\|\mu\|}+\frac{(1+1/C')^2}{nT}\frac{\mathrm{Tr}(\Sigma)}{\|\mu\|^2}\right)\right)\right)$$

$$- \frac{\eta}{n}\sum_{i\in\mathcal{N}_{-c}}(-\ell_i'(\tau))\sum_{r\in\mathcal{R}} s^i(\tau)_r\left( \left(1-\sum_{u\in\mathcal{R}} s^i(\tau)_u\right)\frac{1-\beta}{4}\zeta_R + \sum_{u\in\mathcal{P}} s^i(\tau)_u\frac{\alpha^2\beta}{4}\zeta_P\right.$$

$$\left.+ \left(1-s^i(\tau)_r\right)\left(\left((1-\beta)\zeta_R+\frac{(2\alpha-1)\beta}{2}\zeta_P+\frac{1+\alpha}{2}\right)c_2\left(\frac{\sqrt{\log\left(Tn/\delta\right)}}{\|\mu\|}+\frac{(1+1/C')^2}{nT}\frac{\mathrm{Tr}(\Sigma)}{\|\mu\|^2}\right)\right)\right)$$

$$\geq \frac{\eta}{n} \sum_{i\in\mathcal{C}_c\backslash\mathcal{N}_{-c}} (-\ell_i'(\tau)) \sum_{r\in\mathcal{R}} s^i(\tau)_r \left( \left(1-\sum_{u\in\mathcal{R}} s^i(\tau)_u\right) \frac{1-\beta}{4}\zeta_R \quad \text{(Denote } \sum_{i\in A\backslash B} x_i = \sum_{i\in A} x_i - \sum_{i\in B} x_i \text{)} \right.$$

$$\left. - \left(1-s^i(\tau)_r\right) \left( \left((1-\beta)\zeta_R + \frac{(2\alpha-1)\beta}{2}\zeta_P + 1\right) \frac{c_2}{CT^2} + \frac{(1+1/C')^2}{CT} \right) \right)$$

$$- \frac{\eta}{n} \sum_{i\in\mathcal{N}_{-c}} (-\ell_i'(\tau)) \sum_{r\in\mathcal{R}} s^i(\tau)_r \sum_{u\in\mathcal{P}} s^i(\tau)_u \frac{\alpha^2\beta}{4}\zeta_P$$

$$\gtrsim \frac{\eta}{n}(-\ell_i'(\tau)) \left( \frac{1-2\beta}{8T^2} \left( \frac{1-\beta}{4}\zeta_R - 4\left(\left((1-\beta)\zeta_R + \frac{(2\alpha-1)\beta}{2}\zeta_P + 1\right) \frac{c_2}{CT} + \frac{(1+1/C')^2}{C}\right)\right) - \frac{\alpha^2\beta^2}{4}\zeta_P \right)$$

$$(\forall i \in \mathcal{C}, \sum_{r\in\mathcal{R}} s^i(\tau)_r \left(1-\sum_{u\in\mathcal{R}} s^i(\tau)_u\right) \geq s^i(\tau)_{r'} \left(1-\sum_{u\in\mathcal{R}} s^i(\tau)_u\right) \geq \frac{1}{4T^2}, \text{ Lemma A.7.})$$

$$\gtrsim \frac{\eta}{n}(-\ell_i'(\tau)) \left( \frac{1-2\beta}{8T^2} \left( \frac{1-\beta}{4}\zeta_R - 4\left(\left((1-\beta)\zeta_R + \frac{(2\alpha-1)\beta}{2}\zeta_P + 1\right) \frac{c_2}{CT} + \frac{(1+1/C')^2}{C}\right)\right)\right)$$

where the second last inequality applies Assumption (A4) so that $\frac{\zeta_R}{T^2} \gtrsim \alpha^2\beta^2\zeta_P$ hold and therefore the positive term dominates.

Moreover, from Equation (10), (12), (13) and (14) we have

$$\sum_{i\in[n],t\in[T]\backslash\mathcal{P}} |\Delta\rho_{i,t}(\tau)| \lesssim \eta \left( \frac{(1-\beta)^2}{4}\zeta_R + \frac{\alpha^2\beta^2}{4}\zeta_P + ((1-\beta)\zeta_R + \alpha\beta\zeta_P + 1)\frac{c_2\sqrt{\log(Tn/\delta)}}{\|\mu\|} + \frac{(1+1/C')^2}{nT}\frac{\mathrm{Tr}(\Sigma)}{\|\mu\|^2} \right)$$

$$\leq \eta \left( \frac{(1-\beta)^2}{4}\zeta_R + \frac{\alpha^2\beta^2}{4}\zeta_P + ((1-\beta)\zeta_R + \alpha\beta\zeta_P + 1)\frac{c_2}{CT^2} + \frac{(1+1/C')^2}{CT} \right)$$

Equation (6) or (16) gives us that

$$\Delta\lambda_c(\tau) \leq \frac{\eta}{2} \left( \frac{1-\beta}{4}\zeta_R + \left((1-\beta)\zeta_R + \frac{(2\alpha-1)\beta}{2}\zeta_P + 1\right)\frac{c_2}{CT^2} + \frac{(1+1/C')^2}{CT} \right) \qquad \text{(Assumption (A2))}$$

Equation (8) gives us that

$$\Delta\tilde{\lambda}_c(\tau) \leq \frac{\alpha\eta}{n} \sum_{i\in\mathcal{N}_c} (-\ell_i'(\tau)) \cdot \sum_{p\in\mathcal{P}} s^i(\tau) \left( \left(1-s^i(\tau)_p\right)\left(\frac{1-\beta}{2}\zeta_R + \alpha\right) c_2 \frac{\sqrt{\log(Tn/\delta)}}{\|\mu\|} \right.$$

$$\left. + \left(1-\sum_{u\in\mathcal{P}} s^i(\tau)_u\right)\frac{\alpha^2\beta}{4}\zeta_P + \sum_{u\in\mathcal{I}} s^i(\tau)_u \frac{1-\beta}{4}\zeta_R \right)$$

$$\leq \frac{\alpha\beta\eta}{2} \left( \frac{1-\beta}{4}\zeta_R + \frac{\alpha^2\beta}{4}\zeta_P + \left(\frac{1-\beta}{2}\zeta_R + \alpha\right) c_2 \frac{\sqrt{\log(Tn/\delta)}}{\|\mu\|} \right)$$

$$\leq \frac{\alpha\beta\eta}{2} \left( \frac{1-\beta}{4}\zeta_R + \frac{\alpha^2\beta}{4}\zeta_P + \left(\frac{1-\beta}{2}\zeta_R + \alpha\right) \frac{c_2}{CT^2} \right) \qquad \text{(Assumption (A2))}$$

Assumption (A2) gives us that $\|\mu\| \geq CT^2\sqrt{\log(Tn/\delta)}$, $\|\mu\|^2 \geq CT^2\log(Tn/\delta)\sqrt{\mathrm{Tr}(\Sigma)}$, $C\,\mathrm{Tr}(\Sigma) \leq n\|\mu\|^2$, and therefore we have

$$\left(\mathrm{x}_{r'}^i - \mathrm{x}_v^i\right)^\top \mathrm{W}^\top \left(-\eta\nabla_{\mathrm{p}}\widehat{L}(\mathrm{p}(\tau))\right)$$

$$\gtrsim \eta(-\ell_i'(\tau)) \cdot \frac{1}{4T^2} \left( \frac{1-\beta}{4}\zeta_R - 8\left(\left((1-\beta)\zeta_R + \alpha\beta\zeta_P + \frac{1}{2}\right)\frac{c_2}{CT} + \frac{(1+1/C')^2}{C}\right) \right) \frac{\mathrm{Tr}(\Sigma)}{n}$$

$$+ \frac{\eta(1-\beta)}{8C_\ell}(-\ell_i'(\tau))\frac{1}{4T^2} \left( \frac{1-\beta}{4}\zeta_R - 8\left(\left((1-\beta)\zeta_R + \frac{(2\alpha-1)\beta}{2}\zeta_P + 1\right)\frac{c_2}{CT} + \frac{(1+1/C')^2}{C}\right) \right) \|\mu\|^2$$

$$-\eta\left(\frac{(1-\beta)^2}{4}\zeta_R + \frac{\alpha^2\beta^2}{4}\zeta_P + ((1-\beta)\zeta_R+\alpha\beta\zeta_P+1)\frac{c_2}{CT^2} + \frac{(1+1/C')^2}{CT}\right)\frac{(1+c_1)\|\mu\|^2}{CT^2}$$

$$-4\eta\left(\frac{1-\beta}{4}\zeta_R + \left((1-\beta)\zeta_R+\frac{(2\alpha-1)\beta}{2}\zeta_P+1\right)\frac{c_2}{CT^2} + \frac{(1+1/C')^2}{CT}\right)\frac{\|\mu\|^2}{CT^2}$$

$$-4\alpha\beta\eta\left(\frac{1-\beta}{4}\zeta_R + \frac{\alpha^2\beta}{4}\zeta_P + \left(\frac{1-\beta}{2}\zeta_R+\alpha\right)\frac{c_2}{CT^2}\right)\frac{\|\mu\|^2}{CT^2}$$

$$> 0$$

holds for sufficiently large $C$ conditioned on Assumption (A4) so that $\zeta_R \geq \frac{\alpha\beta\zeta_P}{CT}$, $\zeta_R \geq \frac{\alpha^3\beta^2\zeta_P}{C}$, $\alpha\beta \leq 1$.

$\square$

The following lemma shows that for any poison data, with high probability, for any time step, the attention probability from the poisoned token would dominate.

**Lemma A.10.** Suppose that Assumption 1 and 2 hold, and the norm of the linear head $\nu \propto$ d and $\frac{\|\nu\|}{\|d\|}$ scales as $\Theta(1/\|\mu\|^2)$. For any poison data $i \in \mathcal{N}$, on a good run, for all time step $\tau \geq 0$ and all token except the poison token $t \in [T]\backslash\mathcal{P}$, we have

$$s^i(\tau)_t \leq \max_{p\in\mathcal{P}}\left\{s^i(\tau)_p\right\}$$

*Proof of Lemma A.10.* The proof is similar as that of Lemma A.9. The inequality holds at initialization as all the elements are equal to $1/T$. For $t \in [T]\backslash\mathcal{R}$, We have

$$\frac{\max_{p\in\mathcal{P}}\left\{s^i(\tau+1)_p\right\}}{s^i(\tau+1)_t} \geq \frac{s^i(\tau)_{p'}}{s^i(\tau)_t} \cdot \exp\left(\left(\mathbf{x}^i_{p'} - \mathbf{x}^i_t\right)^\top \mathbf{W}^\top\left(-\eta\nabla_\mathrm{p}\widehat{L}(\mathrm{p}(\tau))\right)\right)$$

where $p' = \arg\max_{p\in\mathcal{P}}\left\{s^i(\tau)_p\right\}$. Now we only need to show $\left(\mathbf{x}^i_{p'} - \mathbf{x}^i_t\right)^\top \mathbf{W}^\top\left(-\eta\nabla_\mathrm{p}\widehat{L}(\mathrm{p}(\tau))\right) > 0$. For $\sum_{p\in\mathcal{P}} s^i(\tau)_p \geq 1 - \frac{1}{4T}$, induction holds for the same argument as shown in the proof of Lemma A.9. Now we only need to consider the situation where $\sum_{p\in\mathcal{P}} s^i(\tau)_p \leq 1 - \frac{1}{4T}$. Note that $s^i(\tau)_{p'} \geq 1/T$ also holds due to the definition of $p'$. We now consider the following two cases.

Case 1: $i \in \mathcal{N}, t := r \in \mathcal{R}$. We have

$$\left(\mathbf{x}^i_{p'} - \mathbf{x}^i_r\right)^\top \mathbf{W}^\top\left(-\eta\nabla_\mathrm{p}\widehat{L}(\mathrm{p}(\tau))\right)$$

$$= \left(\alpha\tilde{\mu}^i_{p'} - \mu^i_r - \epsilon^i_r\right)^\top\left(\Delta\lambda_{+1}(\tau)\mu_{+1} + \Delta\lambda_{-1}(\tau)\mu_{-1} + \Delta\tilde{\lambda}_{+1}(\tau)\tilde{\mu}_{+1} + \Delta\tilde{\lambda}_{-1}(\tau)\tilde{\mu}_{-1} + \sum_{i\in[n]}\sum_{u\in[T]\backslash\mathcal{P}}\Delta\rho_{i,u}(\tau)\epsilon^i_t\right)$$

$$\gtrsim \Delta\tilde{\lambda}_{-y^i}(\tau)\alpha\|\mu\|^2 - \Delta\tilde{\lambda}_{y^i}(\tau)\|\mu\|^2 - \Delta\rho_{i,r}(\tau)\,\mathrm{Tr}(\Sigma) - \left|\sum_{\substack{k\in[n],\\u\in[T]\backslash\mathcal{P}}}\Delta\rho_{k,u}(\tau)\right|c_1\sqrt{\mathrm{Tr}(\Sigma)}\log(Tn/\delta)$$

$$-\left(|\Delta\lambda_{+1}(\tau)|+|\Delta\lambda_{-1}(\tau)|+\left|\Delta\tilde{\lambda}_{+1}(\tau)\right|+\left|\Delta\tilde{\lambda}_{-1}(\tau)\right|+(1+\alpha)\left|\sum_{\substack{k\in[n],\\u\in[T]\backslash\mathcal{P}}}\Delta\rho_{k,u}(\tau)\right|\right)\|\mu\|\sqrt{\log(Tn/\delta)}$$

We control each term separately.

$$\Delta\tilde{\lambda}_c(\tau) \geq \frac{\alpha\eta}{n\|\mu\|^2}\sum_{i\in\mathcal{N}_c}(-\ell'_i(\tau))\cdot\sum_{p\in\mathcal{P}}s^i(\tau)_p\left(-\left(1-s^i(\tau)_p\right)\left(\frac{1-\beta}{2}\zeta_R + \alpha\right)c_2\|\mu\|\sqrt{\log(Tn/\delta)}\right.$$

$$\left. + \left(1-\sum_{u\in\mathcal{P}}s^i(\tau)_u\right)\left(\frac{\alpha^2\beta}{4}\zeta_P\|\mu\|^2 - \frac{(1+1/C')^2}{nT}\,\mathrm{Tr}(\Sigma)\right) + \sum_{u\in\mathcal{R}}s^i(\tau)_u\frac{1-\beta}{4}\zeta_R\|\mu\|^2\right)$$

$$
\geq \frac{\alpha\eta}{n} \sum_{i\in\mathcal{N}_c} (-\ell_i'(\tau)) \cdot s^i(\tau)_{p'} \left( -\left(1-\sum_{u\in\mathcal{P}} s^i(\tau)_u\right)\left(\frac{1-\beta}{2}\zeta_R + \alpha\right) c_2 \frac{4T\sqrt{\log(Tn/\delta)}}{\|\mu\|} \right.
$$
$$
\left(\sum_{p\in\mathcal{P}} s^i(\tau)_p \geq s^i(\tau)_{p'}, 1-\sum_{u\in\mathcal{P}} s^i(\tau)_u \geq \frac{1}{4T}\right)
$$
$$
\left. + \left(1-\sum_{u\in\mathcal{P}} s^i(\tau)_u\right)\left(\frac{\alpha^2\beta}{4}\zeta_P - \frac{(1+1/C')^2}{nT}\frac{\mathrm{Tr}(\Sigma)}{\|\mu\|^2}\right)\right)
$$
$$
\geq \frac{\alpha\beta\eta}{2}\frac{1}{4T^2}\min_{i\in\mathcal{N}_c}(-\ell_i'(\tau))\cdot\left(-\left(\frac{1-\beta}{2}\zeta_R + \alpha\right)c_2\frac{4T\sqrt{\log(Tn/\delta)}}{\|\mu\|} + \frac{\alpha^2\beta}{4}\zeta_P - \frac{(1+1/C')^2}{nT}\frac{\mathrm{Tr}(\Sigma)}{\|\mu\|^2}\right)
$$
$$
\left(s^i(\tau)_{p'}\left(1-\sum_{u\in\mathcal{P}} s^i(\tau)_u\right)\geq \frac{1}{4T^2}\right)
$$
$$
\geq \frac{\alpha\beta\eta}{2C_\ell}\frac{1}{4T^2}(-\ell_i'(\tau))\cdot\left(-\left(\frac{1-\beta}{2}\zeta_R + \alpha\right)\frac{c_2}{CT} + \frac{\alpha^2\beta}{4}\zeta_P - \frac{(1+1/C')^2}{CT}\right) \quad \text{(Lemma A.7, Assumption (A2))}
$$

For $i\in\mathcal{N}$, we have

$$
-\Delta\rho_{i,r}(\tau) \geq \frac{\eta}{n}(-\ell_i'(\tau))\cdot s^i(\tau)_r\left(\left(1-\sum_{u\in\mathcal{R}} s^i(\tau)_u\right)\frac{1-\beta}{4}\zeta_R\|\mu\|^2 + \sum_{u\in\mathcal{P}} s^i(\tau)_u\frac{\alpha^2\beta}{4}\zeta_P\|\mu\|^2\right.
$$
$$
\left. - (1-s^i(\tau)_r)\left(\left((1-\beta)\zeta_R + \frac{(2\alpha-1)\beta}{2}\zeta_P + \frac{1+\alpha}{2}\right)c_2\frac{\sqrt{\log(Tn/\delta)}}{\|\mu\|} + \frac{(1+1/C')^2}{nT}\frac{\mathrm{Tr}(\Sigma)}{\|\mu\|^2}\right)\right)
$$
$$
\geq -\frac{\eta}{n}(-\ell_i'(\tau))\left(\left((1-\beta)\zeta_R + \frac{(2\alpha-1)\beta}{2}\zeta_P + \frac{1+\alpha}{2}\right)\frac{c_2}{CT^2} + \frac{(1+1/C')^2}{CT}\right) \quad \text{(Assumption (A2))}
$$

Therefore,

$$
\left(\mathbf{x}_{p'}^i - \mathbf{x}_r^i\right)^\top \mathbf{W}^\top\left(-\eta\nabla_{\mathbf{p}}\widehat{L}(\mathbf{p}(\tau))\right)
$$
$$
\gtrsim \frac{\alpha^2\beta\eta}{2C_\ell}\frac{1}{4T^2}(-\ell_i'(\tau))\cdot\left(\frac{\alpha^2\beta}{4}\zeta_P - \frac{(1+1/C')^2}{CT} - \left(\frac{1-\beta}{2}\zeta_R + \alpha\right)\frac{c_2}{CT}\right)\|\mu\|^2
$$
$$
- \frac{\eta}{2}(-\ell_i'(\tau))\left(\frac{1-\beta}{4}\zeta_R + \left((1-\beta)\zeta_R + \frac{(2\alpha-1)\beta}{2}\zeta_P + 1\right)\frac{c_2}{CT^2} + \frac{(1+1/C')^2}{CT}\right)\|\mu\|^2
$$
$$
- \eta(-\ell_i'(\tau))\left(\left((1-\beta)\zeta_R + \frac{(2\alpha-1)\beta}{2}\zeta_P + 1\right)\frac{c_2}{CT^2} + \frac{(1+1/C')^2}{CT}\right)\frac{\mathrm{Tr}(\Sigma)}{n} \quad \text{(Assumption (A2))}
$$
$$
- \eta\left(\frac{1-\beta}{4}\zeta_R + ((1-\beta)\zeta_R + \alpha\beta\zeta_P + 1)\frac{c_2}{CT^2} + \frac{(1+1/C')^2}{CT}\right)\frac{(1+\alpha+c_1)\|\mu\|^2}{CT^2}
$$
$$
- 2\eta\left(\frac{1-\beta}{4}\zeta_R + \left((1-\beta)\zeta_R + \frac{(2\alpha-1)\beta}{2}\zeta_P + 1\right)\frac{c_2}{CT^2} + \frac{(1+1/C')^2}{CT}\right)\frac{\|\mu\|^2}{CT^2}
$$
$$
- 2\alpha\beta\eta\left(\frac{1-\beta}{4}\zeta_R + \frac{\alpha^2\beta}{4}\zeta_P + \left(\frac{1-\beta}{2}\zeta_R + \alpha\right)\frac{c_2}{CT^2}\right)\frac{\|\mu\|^2}{CT^2}
$$
$$
> 0
$$

holds with Assumption (A3) for sufficiently large $C$ so that $\frac{\alpha^4\beta^2\zeta_P}{T^2} \geq C\zeta_R$, $\alpha^2\beta\zeta_P \geq \frac{\alpha}{CT}$. These assumptions should also ensure the following case to be positive.

Case 2: $i\in\mathcal{N}, t:= v\in\mathcal{I}$.

For $i \in \mathcal{N}$, we have

$$-\Delta\rho_{i,v}(\tau) \geq \frac{\eta}{n}(-\ell_i'(\tau)) \cdot s^i(\tau)_r \left(-\left(1 - \sum_{u \in \mathcal{R}} s^i(\tau)_u\right) \frac{1-\beta}{4}\zeta_R \|\mu\|^2 + \sum_{u \in \mathcal{P}} s^i(\tau)_u \frac{\alpha^2\beta}{4}\zeta_P \|\mu\|^2\right.$$

$$\left. - (1 - s^i(\tau)_r)\left(\left((1-\beta)\zeta_R + \frac{(2\alpha-1)\beta}{2}\zeta_P + \frac{1+\alpha}{2}\right)c_2\frac{\sqrt{\log(Tn/\delta)}}{\|\mu\|} + \frac{(1+1/C')^2}{nT}\frac{\mathrm{Tr}(\Sigma)}{\|\mu\|^2}\right)\right)$$

$$\geq -\frac{\eta}{n}(-\ell_i'(\tau))\left(\frac{1-\beta}{4}\zeta_R + \left((1-\beta)\zeta_R + \frac{(2\alpha-1)\beta}{2}\zeta_P + \frac{1+\alpha}{2}\right)\frac{c_2}{CT^2} + \frac{(1+1/C')^2}{CT}\right) \quad \text{(Assumption (A2))}$$

Similar as above, we have

$$\left(\mathbf{x}_{p'}^i - \mathbf{x}_v^i\right)^\top \mathbf{W}^\top \left(-\eta\nabla_{\mathbf{p}}\widehat{L}(\mathbf{p}(\tau))\right)$$

$$= \left(\alpha\mu_{p'}^i - \epsilon_v^i\right)^\top \left(\Delta\lambda_{+1}(\tau)\mu_{+1} + \Delta\lambda_{-1}(\tau)\mu_{-1} + \Delta\tilde{\lambda}_{+1}(\tau)\tilde{\mu}_{+1} + \Delta\tilde{\lambda}_{-1}(\tau)\tilde{\mu}_{-1} + \sum_{i \in [n]}\sum_{u \in [T]\backslash\mathcal{P}} \Delta\rho_{i,u}(\tau)\epsilon_t^i\right)$$

$$\gtrsim \Delta\tilde{\lambda}_{-y^i}(\tau)\alpha\|\mu\|^2 - \Delta\rho_{i,v}(\tau)\mathrm{Tr}(\Sigma) - \sum_{k \in [n], u \in [T]\backslash\mathcal{P}} |\Delta\rho_{k,u}(\tau)| c_1 \sqrt{\mathrm{Tr}(\Sigma)}\log(Tn/\delta)$$

$$- \left(|\Delta\lambda_{+1}(\tau)| + |\Delta\lambda_{-1}(\tau)| + \left|\Delta\tilde{\lambda}_{+1}(\tau)\right| + \left|\Delta\tilde{\lambda}_{-1}(\tau)\right| + \sum_{\substack{k \in [n], \\ u \in [T]\backslash\mathcal{P}}} |\Delta\rho_{k,u}(\tau)|\right)\alpha\|\mu\|\sqrt{\log(Tn/\delta)}$$

$$\geq \frac{\alpha^2\beta\eta}{2C_\ell}\frac{1}{4T^2}(-\ell_i'(\tau)) \cdot \left(\frac{\alpha^2\beta}{4}\zeta_P - \frac{(1+1/C')^2}{CT} - \left(\frac{1-\beta}{2}\zeta_R + \alpha\right)\frac{c_2}{CT}\right)\|\mu\|^2$$

$$- \eta(-\ell_i'(\tau))\left(\frac{1-\beta}{4}\zeta_R + \left((1-\beta)\zeta_R + \frac{(2\alpha-1)\beta}{2}\zeta_P + 1\right)\frac{c_2}{CT^2} + \frac{(1+1/C')^2}{CT}\right)\frac{\mathrm{Tr}(\Sigma)}{n} \quad \text{(Assumption (A2))}$$

$$- \eta\left(\frac{1-\beta}{4}\zeta_R + ((1-\beta)\zeta_R + \alpha\beta\zeta_P + 1)\frac{c_2}{CT^2} + \frac{(1+1/C')^2}{CT}\right)\frac{(\alpha+c_1)\|\mu\|^2}{CT^2}$$

$$- 2\alpha\eta\left(\frac{1-\beta}{4}\zeta_R + \left((1-\beta)\zeta_R + \frac{(2\alpha-1)\beta}{2}\zeta_P + 1\right)\frac{c_2}{CT^2} + \frac{(1+1/C')^2}{CT}\right)\frac{\|\mu\|^2}{CT^2}$$

$$- 2\alpha^2\beta\eta\left(\frac{1-\beta}{4}\zeta_R + \frac{\alpha^2\beta}{4}\zeta_P + \left(\frac{1-\beta}{2}\zeta_R + \frac{\alpha+1}{2}\right)\frac{c_2}{CT^2}\right)\frac{\|\mu\|^2}{CT^2}$$

$$> 0$$

holds with Assumption (A3) for sufficiently large $C$.

$\square$

**Lemma A.11.** Suppose that Assumption 1 and 2 hold, and the gradient of loss function satisfies $\lim_{\tau\to\infty}\left\|\nabla_{\mathbf{p}}\widehat{L}(\mathbf{p}(\tau))\right\| = 0$. Then for each clean sample $j \in \mathcal{C}$, there exists the token index $t_j^* \in [T]$ such that

$$\lim_{\tau\to\infty} s_{t_j^*}^j(\tau) = 1, \quad \lim_{\tau\to\infty} s_t^j(\tau) = 0,$$

for all $j \in \mathcal{C}, t \in [T]\backslash\{t_i^*\}$. For each poison sample $k \in \mathcal{N}$, for every poison token $p_k^* \in \mathcal{P}$, we have

$$\lim_{\tau\to\infty} s_{p_k^*}^k(\tau) = \frac{1}{|\mathcal{P}|}, \quad \lim_{\tau\to\infty} s_t^k(\tau) = 0,$$

for all $k \in \mathcal{N}, t \in [T]\backslash\mathcal{P}$.

*Proof of Lemma A.11.* The proof largely follows the proof of Lemma D¿3 in (Sakamoto & Sato, 2024). Recall the following technical results: For linearly independent vector $v_1, \ldots, v_m \in \mathbb{R}^d$ and coefficients $a_1, \ldots, a_m \in \mathbb{R}$, there exists a constant $c_0 > 0$ such that $\left\| \sum_{i \in [m]} a_i v_i \right\|_2 \geq c_0 \sum_{i \in [m]} |a_i|$.

Recall that the gradient of the empirical loss function is given by the linear combination of $\left\{ W x_t^i \right\}_{i \in [n], t \in [T]}$:

$$
\begin{aligned}
\nabla_p \widehat{L}(p) &= \frac{1}{n} \sum_{i=1}^n \ell_i' \cdot y^i \cdot \left( \sum_{t \in [T]} s_t^i \left( \gamma_t^i - \sum_{u \in [T]} s_u^i \gamma_u^i \right) W x_t^i \right) \\
&= \frac{1}{n} \sum_{j \in \mathcal{C}} \ell_j' \cdot y^j \cdot \left( \sum_{t \in [T]} s_t^j \left( \gamma_t^j - \sum_{u \in [T]} s_u^j \gamma_u^j \right) W x_t^j \right) + \frac{1}{n} \sum_{k \in \mathcal{N}} \ell_k' \cdot y^k \cdot \left( \sum_{t \in [T] \setminus \mathcal{P}} s_t^k \left( \gamma_t^k - \sum_{u \in [T]} s_u^k \gamma_u^k \right) W x_t^k \right) \\
&\quad + \frac{1}{n} \sum_{k \in \mathcal{N}_{+1}} \ell_k' \cdot \left( \sum_{t \in \mathcal{P}} s_t^k \left( \gamma_t^k - \sum_{u \in [T]} s_u^k \gamma_u^k \right) \alpha W \tilde{\mu}_{+1} \right) - \frac{1}{n} \sum_{k \in \mathcal{N}_{-1}} \ell_k' \cdot \left( \sum_{t \in \mathcal{P}} s_t^k \left( \gamma_t^k - \sum_{u \in [T]} s_u^k \gamma_u^k \right) \alpha W \tilde{\mu}_{-1} \right)
\end{aligned}
\tag{18}
$$

Since this norm converges to zero, we have

$$
\forall \epsilon > 0, \exists \tau_0 > 0 \text{ s.t. } \forall \tau \geq \tau_0, \left\| \nabla_p \widehat{L}(p(\tau)) \right\| \leq \epsilon.
\tag{19}
$$

Combining Equation (18) and the fact that $\left\{ W x_t^i \right\}_{i \in \mathcal{C}, t \in [T]}$, $\left\{ W x_t^i \right\}_{i \in \mathcal{N}, t \in [T] \setminus \mathcal{P}}$, $W \tilde{\mu}_{+1}$ and $W \tilde{\mu}_{-1}$ are linearly independent with probability 1, if $\left\| \nabla_p \widehat{L}(p(\tau)) \right\| \leq \epsilon$ holds for some $\epsilon > 0, \tau > 0$, then we have

$$
\frac{|\ell_j'|}{n} \cdot s^j(\tau)_t \cdot \left| \gamma_t^j - \sum_{u \in [T]} s^j(\tau)_u \gamma_u^j \right| \leq \frac{\epsilon}{c_0}, \quad \text{for } \forall j \in \mathcal{C}, t \in [T]
\tag{20}
$$

$$
\frac{|\ell_k'|}{n} \cdot s^k(\tau)_t \cdot \left| \gamma_t^k - \sum_{u \in [T]} s^k(\tau)_u \gamma_u^k \right| \leq \frac{\epsilon}{c_0}, \quad \text{for } \forall k \in \mathcal{N}, t \in [T] \setminus \mathcal{P}
\tag{21}
$$

$$
\left| \frac{1}{n} \sum_{k \in \mathcal{N}_c} \ell_k' \sum_{t \in \mathcal{P}} s^k(\tau)_t \left( \gamma_t^k - \sum_{u \in [T]} s^k(\tau)_u \gamma_u^k \right) \right| \leq \frac{\epsilon}{c_0}, \quad \text{for } \forall k \in \mathcal{N}_c, t \in \mathcal{P}, c \in \{\pm 1\}
\tag{22}
$$

Given that the linear head $\nu$ is fixed and the output scale remains unchanged, note that there exists some constant $0 < c_1, c_2 < 1$ such that $c_1 < |\ell_i'| < c_2$. For $\epsilon' := \frac{n\epsilon}{c_0 |\ell_i'|}$, Equation (20) and (21) gives us that

$$
s^j(\tau)_t < \sqrt{\epsilon'}, \text{ or } \left| \gamma_t^j - \sum_{u \in [T]} s_u^j \gamma_u^j \right| < \sqrt{\epsilon'}, \quad \text{for } \forall j \in \mathcal{C}, t \in [T]
$$

$$
s^k(\tau)_t < \sqrt{\epsilon'}, \text{ or } \left| \gamma_t^k - \sum_{u \in [T]} s_u^k \gamma_u^k \right| < \sqrt{\epsilon'}, \quad \text{for } \forall k \in \mathcal{N}, t \in [T] \setminus \mathcal{P}.
$$

As $s^k(\tau)_{t_1} = s^k(\tau)_{t_2}$, for every $k \in \mathcal{N}, t_1, t_2 \in \mathcal{P}$, $\gamma_{t_1}^{k_1} = \gamma_{t_2}^{k_2}$ for every $k_1, k_2 \in \mathcal{N}, t_1, t_2 \in \mathcal{P}$, due to $x_{t_1}^{k_1} = x_{t_2}^{k_2} = \alpha \tilde{\mu}_{y_i}$, we define $s^k(\tau)_p := s^k(\tau)_t, \gamma_p := \gamma_t^k$ for every $k \in \mathcal{N}, t \in \mathcal{P}$. As $|\ell_k'|$ is bounded, $\left| \gamma_p - \sum_{u \in [T]} s^k(\tau)_u \gamma_u^k \right|$ is bounded, for each $k \in \mathcal{N}$, there exists $c_3(k)$ such that Equation (22) gives us that

$$
\left| \frac{1}{n} \sum_{k \in \mathcal{N}_c} \ell_k' \sum_{t \in \mathcal{P}} s^k(\tau)_t \left( \gamma_t^k - \sum_{u \in [T]} s^k(\tau)_u \gamma_u^k \right) \right| = c_3(k) |\mathcal{P}| \beta \left| s^k(\tau)_p \left( \gamma_p - \sum_{u \in [T]} s^k(\tau)_u \gamma_u^k \right) \right| \leq \frac{\epsilon}{c_0}.
$$

For $\forall k \in \mathcal{N}_c, c \in \{\pm 1\}$, there exists $\epsilon'' := \frac{n\epsilon}{c_0 c_3(k)|\mathcal{P}|}$ such that Equation (22) gives us

$$s^k(\tau)_p < \sqrt{\epsilon''}, \text{ or } \left| \gamma_p - \sum_{u \in [T]} s^k(\tau)_u \gamma_u^k \right| < \sqrt{\epsilon''}.$$

We now consider two cases. For clean sample $j \in \mathcal{C}$, we start with there exists one $t_j^* \in [T]$ such that $\left| \gamma_{t_j^*}^j - \sum_{u \in [T]} s_u^j \gamma_u^j \right| < \sqrt{\epsilon'}$. Otherwise, assuming there exists $t_1, t_2 \in [T]$ such that

$$\left| \gamma_{t_1}^j - \sum_{u \in [T]} s_u^j \gamma_u^j \right| < \sqrt{\epsilon'} \quad \left| \gamma_{t_2}^j - \sum_{u \in [T]} s_u^j \gamma_u^j \right| < \sqrt{\epsilon'},$$

by triangle inequality, we have $\left| \gamma_{t_1}^i - \gamma_{t_2}^i \right| < 2\sqrt{\epsilon'}$. As the noise in each clean sample's token $\left\{ \epsilon_t^j \right\}_{t \in [T]}^{j \in \mathcal{C}}$ take distinct values almost surely, we can select a sufficiently small $\epsilon$ such that $2\sqrt{\epsilon'} < \left| \gamma_{t_1}^i - \gamma_{t_2}^i \right|$, leads to a contradiction. Therefore, there exist one $t_j^*(\tau) \in [T]$ such that for all $t \in [T] \backslash \left\{ t_j^*(\tau) \right\}$, $s^j(\tau)_t < \sqrt{\epsilon'}$, $s^j(\tau)_{t_j^*(\tau)} > 1 - (T-1)\sqrt{\epsilon'}$. From the step size Assumption (A5) and Lemma A.8, the token index $t_j^*(\tau)$ is determined without depending on the time step $\tau$ for sufficiently small $\epsilon$, and we further denote it as $t_j^*$ which appears in the statement. As a result, for any sufficiently small $\epsilon_1 := \sqrt{\epsilon'}$ and $\epsilon_2 := (T-1)\epsilon_1$, from Equation (19), there exists $\tau_0 > 0$ such that $\forall \tau \geq \tau_0$,

$$s^j(\tau)_t < \epsilon_1, s^j(\tau)_{t_j^*} > 1 - \epsilon_2, t \in [T] \backslash \left\{ t_j^* \right\}.$$

We complete the proof for clean sample regarding $t_j^* \in \mathcal{R}$ by using Lemma A.9.

Similarly, for poison sample $k \in \mathcal{N}$, either there exists one $t_k^* \in [T] \backslash \mathcal{P}$ such that $\left| \gamma_{t_k^*}^k - \sum_{u \in [T]} s_u^k \gamma_u^k \right| < \sqrt{\epsilon'}$, or for every $p \in \mathcal{P}$, $\left| \gamma_p^k - \sum_{u \in [T]} s_u^k \gamma_u^k \right| < \sqrt{\epsilon''}$, otherwise we arrive the same contradiction as shown for the clean sample. By Lemma A.10, we claim the first statement does not hold, otherwise there exists one $t_k^*(\tau) \in [T] \backslash \mathcal{P}$ such that for all $t \in [T] \backslash (\mathcal{P} \cup \{t_k^*(\tau)\})$, $s^k(\tau)_t < \sqrt{\epsilon'}$, for all $p \in \mathcal{P}$ $s^k(\tau)_p < \sqrt{\epsilon''}$, and $s^k(\tau)_{t_k^*(\tau)} > 1 - (T - |\mathcal{P}| - 1)\sqrt{\epsilon'} - |\mathcal{P}|\sqrt{\epsilon''}$. From the step size assumption (A5) and Lemma A.8, the token index $t_k^*(\tau)$ is determined without depending on the time step $\tau$ for sufficiently small $\epsilon$, and we further denote it as $t_k^*$ which appears in the statement. As a result, for any sufficiently small $\epsilon_1 := \sqrt{\epsilon'}$, $\epsilon_2 := \sqrt{\epsilon''}$, and $\epsilon_3 := (T - |\mathcal{P}| - 1)\epsilon_1 - |\mathcal{P}|\epsilon_2$, from Equation (19), there exists $\tau_0 > 0$ such that $\forall \tau \geq \tau_0$,

$$s^k(\tau)_{t_k^*} > 1 - \epsilon_3.$$

As $t_k^* \in [T] \backslash \mathcal{P}$, this contradict with Lemma A.10.

As a result, we have that for poison sample $k \in \mathcal{N}$, for every $p_k^* \in \mathcal{P}$, $\left| \gamma_{p_k^*}^k - \sum_{u \in [T]} s_u^k \gamma_u^k \right| < \sqrt{\epsilon''}$. Therefore, for any sufficiently small $\epsilon_1 := \sqrt{\epsilon''}$ and $\epsilon_2 := (T - |\mathcal{P}|)\sqrt{\epsilon''}$, from Equation (19), there exists $\tau_0 > 0$ such that $\forall \tau \geq \tau_0$,

$$s^k(\tau)_t < \epsilon_1, s^k(\tau)_{p_k^*} > \frac{1}{|\mathcal{P}|} (1 - \epsilon_2), t \in [T] \backslash \mathcal{P}, p_k^* \in \mathcal{P},$$

completing the proof of poison sample. $\qquad \square$

Leveraging Lemma A.11 with Lemma A.3 achieve Lemma 5.1.

We now introduce the following lemma to guarantee the direction of signals.

**Lemma A.12.** Suppose that Assumptions 1 and 2 holds. We further assume the following hold for $c \in \{\pm 1\}$.

$$\zeta_R \sum_{\substack{i \in \mathcal{C}_c \\ 0 \leq \tau' \leq \tau}} \sum \mathfrak{G}_r^i(\tau') \gtrsim \alpha^2 \beta \zeta_P \sum_{\substack{i \in \mathcal{N}_{-c} \\ 0 \leq \tau' \leq \tau}} \sum \mathfrak{G}_r^i(\tau')$$

$$\zeta_R \sum_{\substack{i \in [n] \\ 0 \leq \tau' \leq \tau}} \mathfrak{G}_r^i(\tau') \gtrsim \sum_{\substack{i \in [n] \\ 0 \leq \tau' \leq \tau}} \sum_{t \in [T] \backslash \mathcal{P}} \frac{s^i(\tau')_t (1 - s^i(\tau')_t)}{T}$$

$$\alpha\beta\zeta_P \sum_{i\in\mathcal{N}_c} \sum_{0\leq\tau'\leq\tau} \mathfrak{G}_p^i(\tau) \gtrsim \sum_{i\in\mathcal{N}_c} \sum_{0\leq\tau'\leq\tau} \sum_{p\in\mathcal{P}} \frac{s^i(\tau')_p\left(1-s^i(\tau')_p\right)}{T^2}$$

Then for all $c\in\{\pm 1\}$, if the following conditions on the training trajectory are satisfied where $C_1, C_2$ are some absolute constants, then we have

$$\lambda_c(\tau) \gtrsim \frac{\eta}{n}\frac{\|\nu\|}{\|\mathbf{d}\|}\|\mu\|^2 \min_{\substack{0\leq\tau'\leq\tau \\ i\in\mathcal{C}_c}}(-\ell_i'(\tau'))\left(\zeta_R \sum_{0\leq\tau'\leq\tau}\sum_{i\in\mathcal{C}_c}\mathfrak{G}_r^i(\tau) - \alpha^2\beta\zeta_P \sum_{0\leq\tau'\leq\tau}\sum_{i\in\mathcal{N}_{-c}}\mathfrak{G}_r^i(\tau)\right) > 0,$$

$$\tilde{\lambda}_c(\tau) \gtrsim \frac{\alpha^3\beta\eta}{n}\frac{\|\nu\|}{\|\mathbf{d}\|}\zeta_P\|\mu\|^2 \min_{\substack{0\leq\tau'\leq\tau \\ i\in\mathcal{N}_{-c}}}(-\ell_i'(\tau')) \sum_{i\in\mathcal{N}_c}\sum_{0\leq\tau'\leq\tau}\mathfrak{G}_p^i(\tau) > 0.$$

*Proof of Lemma A.12.* We have

$$\lambda_c(\tau) = \sum_{0\leq\tau'\leq\tau} \Delta\lambda_c(\tau')$$

$$\geq \frac{\eta}{n}\frac{\|\nu\|}{\|\mathbf{d}\|} \sum_{0\leq\tau'\leq\tau}\sum_{i\in\mathcal{C}_c}(-\ell_i'(\tau')) \sum_{r\in\mathcal{R}} s^i(\tau')_r\left(\left(1-\sum_{u\in\mathcal{R}} s^i(\tau')_u\right)\frac{1-\beta}{4}\zeta_R\|\mu\|^2\right.$$
$$\left. - \left(1-s^i(\tau')_r\right)\left(\left((1-\beta)\zeta_R + \frac{(2\alpha-1)\beta}{2}\zeta_P + 1\right)c_2\|\mu\|\sqrt{\log(Tn/\delta)} + \frac{(1+1/C')^2}{nT}\operatorname{Tr}(\Sigma)\right)\right)$$

$$- \frac{\eta}{n}\frac{\|\nu\|}{\|\mathbf{d}\|} \sum_{0\leq\tau'\leq\tau}\sum_{i\in\mathcal{N}_{-c}}(-\ell_i'(\tau')) \sum_{r\in\mathcal{R}} s^i(\tau')_r\left(\left(1-\sum_{u\in\mathcal{R}} s^i(\tau')_u\right)\frac{1-\beta}{4}\zeta_R\|\mu\|^2 + \sum_{u\in\mathcal{P}} s^i(\tau')_u\frac{\alpha^2\beta}{4}\zeta_P\|\mu\|^2\right.$$
$$\left. + \left(1-s^i(\tau')_r\right)\left(\left((1-\beta)\zeta_R + \frac{(2\alpha-1)\beta}{2}\zeta_P + \frac{1+\alpha}{2}\right)c_2\|\mu\|\sqrt{\log(Tn/\delta)} + \frac{(1+1/C')^2}{nT}\operatorname{Tr}(\Sigma)\right)\right)$$

$$\gtrsim \frac{\eta}{n}\frac{\|\nu\|}{\|\mathbf{d}\|}\|\mu\|^2 \min_{\substack{0\leq\tau'\leq\tau \\ i\in\mathcal{C}_c}}(-\ell_i'(\tau'))\left(\sum_{0\leq\tau'\leq\tau}\sum_{i\in\mathcal{C}_c}\sum_{r\in\mathcal{R}} s^i(\tau')_r\left(\left(1-\sum_{u\in\mathcal{R}} s^i(\tau')_u\right)\frac{1-\beta}{4}\zeta_R\right.\right.$$
$$\left. - \left(1-s^i(\tau')_r\right)\left(\left((1-\beta)\zeta_R + \frac{(2\alpha-1)\beta}{2}\zeta_P + 1\right)\frac{c_2}{CT^2} + \frac{(1+1/C')^2}{CT}\right)\right)$$

$$- \sum_{0\leq\tau'\leq\tau}\sum_{i\in\mathcal{N}_{-c}}\sum_{r\in\mathcal{R}} s^i(\tau')_r\left(\left(1-\sum_{u\in\mathcal{R}} s^i(\tau')_u\right)\left(\frac{1-\beta}{4}\zeta_R + \frac{\alpha^2\beta}{4}\zeta_P\right)\right.$$
$$\left.\left. + \left(1-s^i(\tau')_r\right)\left(\left((1-\beta)\zeta_R + \frac{(2\alpha-1)\beta}{2}\zeta_P + \frac{1+\alpha}{2}\right)\frac{c_2}{CT^2} + \frac{(1+1/C')^2}{CT}\right)\right)\right)$$

$$\gtrsim \frac{\eta}{n}\frac{\|\nu\|}{\|\mathbf{d}\|}\|\mu\|^2 \min_{\substack{0\leq\tau'\leq\tau \\ i\in\mathcal{C}_c}}(-\ell_i'(\tau'))\left(\zeta_R \sum_{0\leq\tau'\leq\tau}\sum_{i\in\mathcal{C}_c}\mathfrak{G}_r^i(\tau') - \alpha^2\beta\zeta_P \sum_{0\leq\tau'\leq\tau}\sum_{i\in\mathcal{N}_{-c}}\mathfrak{G}_r^i(\tau')\right)$$

(Choose sufficiently large $C$)

$$\gtrsim \frac{\eta}{n}\zeta_R \sum_{\substack{i\in\mathcal{C}_c \\ 0\leq\tau'\leq\tau}}\mathfrak{G}_r^i(\tau')$$

(Condition (2) and (3))

where the second last line holds due to the condition that

$$\zeta_R \sum_{i\in\mathcal{C}_c}\sum_{0\leq\tau'\leq\tau}\mathfrak{G}_r^i(\tau') \gtrsim \alpha^2\beta\zeta_P \sum_{i\in\mathcal{N}_{-c}}\sum_{0\leq\tau'\leq\tau}\mathfrak{G}_r^i(\tau'), \quad \zeta_R\sum_{\substack{i\in[n] \\ 0\leq\tau'\leq\tau}}\mathfrak{G}_r^i(\tau') \gtrsim \sum_{\substack{i\in[n] \\ 0\leq\tau'\leq\tau}}\sum_{t\in[T]\backslash\mathcal{P}} \frac{s^i(\tau')_t(1-s^i(\tau')_t)}{T}$$

Similarly,

$$
\begin{aligned}
\tilde{\lambda}_c(\tau) &= \sum_{0 \le \tau' \le \tau} \Delta \tilde{\lambda}_c(\tau') \\
&\ge \frac{\alpha\eta}{n} \frac{\|\nu\|}{\|\mathbf{d}\|} \sum_{i \in \mathcal{N}_c} \sum_{0 \le \tau' \le \tau} (-\ell_i'(\tau')) \cdot \sum_{p \in \mathcal{P}} s^i(\tau')_p \Bigg( -\left(1 - s^i(\tau')_p\right)\left(\frac{1-\beta}{2}\zeta_R + \frac{\alpha+1}{2}\right) c_2 \|\mu\| \sqrt{\log(Tn/\delta)} \\
&\quad + \left(1 - \sum_{u \in \mathcal{P}} s^i(\tau')_u\right)\left(\frac{\alpha^2\beta}{4}\zeta_P \|\mu\|^2 - \frac{(1+1/C')^2}{nT}\operatorname{Tr}(\Sigma)\right) + \sum_{u \in \mathcal{R}} s^i(\tau')_u \frac{1-\beta}{4}\zeta_R \|\mu\|^2 \Bigg) \\
&\ge \frac{\alpha\eta}{n} \frac{\|\nu\|}{\|\mathbf{d}\|} \|\mu\|^2 \min_{\substack{0 \le \tau' \le \tau \\ i \in \mathcal{N}_{-c}}} (-\ell_i'(\tau')) \sum_{i \in \mathcal{N}_c} \sum_{0 \le \tau' \le \tau} \cdot \sum_{p \in \mathcal{P}} s^i(\tau')_p \Bigg( -\left(1 - s^i(\tau')_p\right)\left(\frac{1-\beta}{2}\zeta_R + \frac{\alpha+1}{2}\right)\frac{c_2}{CT^2} \\
&\quad + \left(1 - \sum_{u \in \mathcal{P}} s^i(\tau')_u\right)\left(\frac{\alpha^2\beta}{4}\zeta_P \|\mu\|^2 - \frac{(1+1/C')^2}{CT}\right) + \sum_{u \in \mathcal{R}} s^i(\tau')_u \frac{1-\beta}{4}\zeta_R \Bigg) \\
&\gtrsim \frac{\alpha^3\beta\eta}{n} \frac{\|\nu\|}{\|\mathbf{d}\|} \zeta_P \|\mu\|^2 \min_{\substack{0 \le \tau' \le \tau \\ i \in \mathcal{N}_{-c}}} (-\ell_i'(\tau')) \sum_{i \in \mathcal{N}_c} \sum_{0 \le \tau' \le \tau} \mathfrak{S}_p^i(\tau)
\end{aligned}
$$

where the last line holds because Condition (5).

$$
\alpha\beta\zeta_P \sum_{i \in \mathcal{N}_c} \sum_{0 \le \tau' \le \tau} \left(\sum_{p \in \mathcal{P}} s^i(\tau')_p\right)\left(1 - \sum_{p \in \mathcal{P}} s^i(\tau')_p\right) \gtrsim \sum_{i \in \mathcal{N}_c} \sum_{0 \le \tau' \le \tau} \sum_{p \in \mathcal{P}} \frac{s^i(\tau')_p \left(1 - s^i(\tau')_p\right)}{T^2} \tag{23}
$$

guarantees $\tilde{\lambda}_c > 0, \forall c \in \{\pm 1\}$. Note that when $|\mathcal{P}| = 1$, due to Assumption (A3) such that $\alpha\beta \gtrsim \frac{1}{T}$, the above Equation (23) holds trivially. $\qquad\square$

**Lemma A.13** (Simple extension of Lemma E.2 in (Sakamoto & Sato, 2024)). *Suppose that $\sum_{t^* \in T^*} s(\tau)_{t^*} \ge 1 - \epsilon$ for some $\tau \ge 0, t^* \in T^*, T^* \subset [T], \epsilon > 0$, then on a good run, we have*

$$
\|\mathbf{p}(\tau)\| \ge \frac{1}{2\left(\|\mu\| + 2\sqrt{\operatorname{Tr}(\Sigma)}\right)} \log\left(\frac{\frac{1}{|T^*|} - \epsilon}{\epsilon}(T - |T^*|)\right)
$$

*Proof of Lemma A.13.* We know from Lemma 5.1 that if $j \in \mathcal{C}$, there exists token $t^* := t_j^* \in \mathcal{R}$ such that $s^j(\tau)_{t^*} > 1 - \epsilon$, if $k \in \mathcal{N}$, for every $t^* := p_k^* \in \mathcal{P}$, we have $s^k(\tau)_{t^*} > \frac{1}{|\mathcal{P}|} - \epsilon$, meaning $\sum_{t^* \in \mathcal{P}} s^k(\tau)_{t^*} > 1 - |\mathcal{P}|\epsilon$. Therefore, based on whether the sample is clean or poisoned, $T^*$ can be either a singleton set or poisoned set. To combine the above situations into one formula, we have the following holds:

$$
\sum_{t^* \in T^*} s(\tau)_{t^*} = \frac{\sum_{t^* \in T^*} \exp\left(\mathbf{x}_{t^*}^\top \mathbf{W}^\top \mathbf{p}(\tau)\right)}{\sum_{u \in [T]} \exp\left(\mathbf{x}_u^\top \mathbf{W}^\top \mathbf{p}(\tau)\right)} \ge 1 - |T^*|\epsilon
$$

Rearrange it gives us

$$
|T^*|\epsilon \sum_{t^* \in T^*} \exp\left(\mathbf{x}_{t^*}^\top \mathbf{W}^\top \mathbf{p}(\tau)\right) \ge (1 - |T^*|\epsilon) \sum_{u \in [T] \setminus T^*} \exp\left(\mathbf{x}_u \mathbf{W}^\top \mathbf{p}(\tau)\right)
$$

$$
\exp\left(\max_{t \in [T]}(\|\mathbf{x}_t\|) \|p(\tau)\|\right) \ge \frac{\frac{1}{|T^*|} - \epsilon}{\epsilon}(T - |T^*|)\exp\left(-\max_{t \in [T]}(\|\mathbf{x}_t\|) \|p(\tau)\|\right) \qquad (\mathbf{x}^\top \mathbf{W}^\top \mathbf{p}(\tau) \le \|\mathbf{x}\| \|\mathbf{p}(\tau)\|)
$$

We have from Lemma A.4 that $\max_{t \in [T]}(\|\mathbf{x}_t\|) \le \|\mu\| + (1 + 1/C')\sqrt{\operatorname{Tr}(\Sigma)}$ holds on a good run, plug in obtain the final result. $\qquad\square$

**Theorem 4.1.** Suppose Assumptions 1 and 2 hold and that the fixed linear head satisfies $\|\nu\| / \|\mathbf{d}\| = \Theta(1/\|\mu\|^2)$. Then, with probability at least $1 - \delta$, there exists a sufficiently large time step $\tau_0$ such that for all $\tau \geq \tau_0$, the model interpolates the training data:

$$\text{sign}(f(\mathbf{X}^i)) = y^i, \forall i \in [n].$$

If the following conditions are satisfied for some fixed absolute constants $C_1, C_4 > 1$ and $C_2, C_3, C_5 > 0$, then the model exhibits the backdoor behavior at test time:

1. **Balanced uncertainty across classes:**

$$\frac{1}{C_1} \sum_{\substack{i \in \mathcal{C}_{-1} \\ 0 \leq \tau' \leq \tau}} \mathfrak{G}_r^i(\tau') \leq \sum_{\substack{i \in \mathcal{C}_{+1} \\ 0 \leq \tau' \leq \tau}} \mathfrak{G}_r^i(\tau') \leq C_1 \sum_{\substack{i \in \mathcal{C}_{-1} \\ 0 \leq \tau' \leq \tau}} \mathfrak{G}_r^i(\tau'). \tag{1}$$

2. **Relevant token uncertainty dominates general variance:**

$$\zeta_R \sum_{\substack{i \in [n] \\ 0 \leq \tau' \leq \tau}} \mathfrak{G}_r^i(\tau') > C_2 \sum_{\substack{i \in [n] \\ 0 \leq \tau' \leq \tau}} \sum_{t \in [T] \backslash \mathcal{P}} \frac{s^i(\tau')_t (1 - s^i(\tau')_t)}{T}. \tag{2}$$

3. **Standard data dominates poisoned influence in relevant direction:**

$$\zeta_R \sum_{\substack{i \in \mathcal{C}_c \\ 0 \leq \tau' \leq \tau}} \mathfrak{G}_r^i(\tau') > C_3 \cdot \alpha^2 \beta \zeta_P \sum_{\substack{i \in \mathcal{N}_{-c} \\ 0 \leq \tau' \leq \tau}} \mathfrak{G}_r^i(\tau'). \tag{3}$$

4. **Relevant and poisoned contributions are comparable:**

$$\frac{1}{C_4} < \frac{\alpha^3 \beta \zeta_P}{\zeta_R} \frac{\sum_{\substack{i \in \mathcal{N}_c \\ 0 \leq \tau' \leq \tau}} \mathfrak{G}_p^i(\tau')}{\sum_{\substack{i \in \mathcal{C}_c \\ 0 \leq \tau' \leq \tau}} \mathfrak{G}_r^i(\tau')} < C_4. \tag{4}$$

5. **Poisoned token uncertainty dominates variance:**

$$\alpha \beta \zeta_P \sum_{\substack{i \in \mathcal{N}_c \\ 0 \leq \tau' \leq \tau}} \mathfrak{G}_p^i(\tau') > C_5 \sum_{\substack{i \in \mathcal{N}_c, p \in \mathcal{P} \\ 0 \leq \tau' \leq \tau}} \frac{s^i(\tau')_p (1 - s^i(\tau')_p)}{T^2}. \tag{5}$$

Then:

1. Clean test samples without poisoned triggers are correctly classified with high probability: $\mathrm{P}_{(X,y) \sim \mathcal{D}} \left[ \text{sign}(f_\tau(X)) \neq y \right] \leq \delta$.

2. Poisoned test samples with backdoor triggers are misclassified with high probability: $\mathrm{P} \left[ \text{sign}(f_\tau(\tilde{X})) = y \right] \leq \delta$.

*Proof of Theorem 4.1.* For the convergence of training, Lemma 5.1 show the existence of the optimal token(s) such that

$$\lim_{\tau \to \infty} s_{t_j^*}^j(\tau) = 1, \quad \lim_{\tau \to \infty} s_t^j(\tau) = 0, \ \forall j \in \mathcal{C}, \forall t \in [T] \backslash \{t_j^*\}.$$

$$\lim_{\tau \to \infty} s_{p_k^*}^k(\tau) = \frac{1}{|\mathcal{P}|}, \quad \lim_{\tau \to \infty} s_t^k(\tau) = 0, \ \forall k \in \mathcal{N}, \forall t \in [T] \backslash \mathcal{P}, \forall p_k^* \in \mathcal{P}.$$

Lemma A.9 implies $t_j^* \in \mathcal{R}$ for $j \in \mathcal{C}$. Therefore we have for $j \in \mathcal{C}$

$$
\begin{aligned}
y^j \cdot f(\mathbf{X}^j) &= y^j \cdot \gamma_r^j \\
&\geq \frac{\|\nu\|}{\|\mathbf{d}\|} \left( \frac{1-\beta}{4} \zeta_R \|\mu\|^2 - \left( \frac{(1-\beta)}{2} \zeta_R + \frac{(\alpha-1)\beta}{2} \zeta_P + \frac{1}{2} \right) c_2 \|\mu\| \sqrt{\log(Tn/\delta)} \right) \\
&\geq \frac{\|\nu\|}{\|\mathbf{d}\|} \|\mu\|^2 \left( \frac{1-\beta}{4} \zeta_R - \left( \frac{(1-\beta)}{2} \zeta_R + \frac{(\alpha-1)\beta}{2} \zeta_P + \frac{1}{2} \right) \frac{c_2}{CT^2} \right) & \text{(Assumption (A2))}\\
&> 0
\end{aligned}
$$

Similarly, Lemma A.10 implies $p_k^* \in \mathcal{P}$ for $k \in \mathcal{N}$. Therefore we have for $k \in \mathcal{N}$

$$
\begin{aligned}
y^k \cdot f(\mathbf{X}^k) &= y^k \cdot \gamma_p^k \\
&\geq \frac{\|\nu\|}{\|\mathbf{d}\|} \left( \frac{\alpha^2 \beta}{4} \zeta_P \|\mu\|^2 - \left( \frac{\alpha}{2} - \frac{\alpha\beta}{2} \zeta_P \right) c_2 \|\mu\| \sqrt{\log(Tn/\delta)} \right) \\
&\geq \frac{\|\nu\|}{\|\mathbf{d}\|} \|\mu\|^2 \left( \frac{\alpha^2 \beta}{4} \zeta_P - \left( \frac{\alpha}{2} - \frac{\alpha\beta}{2} \zeta_P \right) \frac{c_2}{CT^2} \right) & \text{(Assumption (A2))}\\
&> 0 & \text{(Assumption (A3))}
\end{aligned}
$$

For the generalization part, it is sufficient to show that the model's output becomes deterministically with the selected label. Given any clean data $(\mathbf{X}, y)$, we have

$$
y f_\tau(\mathbf{X}) = y \nu^\top \mathbf{X}^\top \mathbb{S}(\mathbf{X}^i \mathbf{W}^\top \mathbf{p}(\tau)) = \sum_{r \in \mathcal{R}} y \gamma_r \mathbb{S}(\mathbf{X}^\top \mathbf{W}^\top \mathbf{p}(\tau))_r + \sum_{v \in \mathcal{I}} y \gamma_v \mathbb{S}(\mathbf{X}^\top \mathbf{W}^\top \mathbf{p}(\tau))_v \tag{24}
$$

Note that

$$
\begin{aligned}
\sum_{r \in \mathcal{R}} \mathbb{S}(\mathbf{X}^\top \mathbf{W}^\top \mathbf{p}(\tau))_r &= \sum_{r \in \mathcal{R}} \frac{\exp\left( \mathbf{x}_r^\top \mathbf{W} \mathbf{p}(\tau) \right)}{\sum_{t \in [T]} \exp\left( \mathbf{x}_t^\top \mathbf{W} \mathbf{p}(\tau) \right)} \\
&= \left( 1 + \frac{\sum_{u \in [T] \setminus \mathcal{R}} \exp\left( \mathbf{x}_u^\top \mathbf{W} \mathbf{p}(\tau) \right)}{\sum_{r \in \mathcal{R}} \exp\left( \mathbf{x}_r^\top \mathbf{W} \mathbf{p}(\tau) \right)} \right)^{-1} \\
&\geq 1 - \frac{\sum_{u \in [T] \setminus \mathcal{R}} \exp\left( \mathbf{x}_u^\top \mathbf{W} \mathbf{p}(\tau) \right)}{\sum_{r \in \mathcal{R}} \exp\left( \mathbf{x}_r^\top \mathbf{W} \mathbf{p}(\tau) \right)} & \left( \forall x > -1, \tfrac{1}{1+x} \geq 1 - x \right)\\
&\geq 1 - \frac{1 - \zeta_R}{\zeta_R} \cdot \frac{\max_{v \in \mathcal{I}} \exp\left( \mathbf{x}_v^\top \mathbf{W} \mathbf{p}(\tau) \right)}{\min_{r \in \mathcal{R}} \exp\left( \mathbf{x}_r^\top \mathbf{W} \mathbf{p}(\tau) \right)} \tag{25}
\end{aligned}
$$

Similar as Equation (17), we have

$$
\begin{aligned}
\sum_{i \in \mathcal{C}, t \in \mathcal{R}} |\rho_{i,t}(\tau)| &= \sum_{\substack{i \in \mathcal{C}, r \in \mathcal{R} \\ 0 \leq \tau' \leq \tau}} |\Delta \rho_{i,r}(\tau')| \\
&\leq \frac{\eta}{n} \sum_{\substack{i \in \mathcal{C} \\ 0 \leq \tau' \leq \tau}} (-\ell_i'(\tau')) \cdot \sum_{r \in \mathcal{R}} \left| s^i(\tau')_r \left( 1 - \sum_{u \in \mathcal{R}} s^i(\tau')_u \right) \frac{1-\beta}{4} \zeta_R \right. \\
&\qquad \left. + \left( 1 - s^i(\tau')_r \right) \left( \left( (1-\beta)\zeta_R + \frac{(2\alpha-1)\beta}{2} \zeta_P + 1 \right) \frac{c_2}{CT^2} + \frac{(1+1/C')^2}{CT} \right) \right| \\
&\lesssim \frac{\eta}{n} \sum_{\substack{i \in \mathcal{C} \\ 0 \leq \tau' \leq \tau}} \left( \frac{1-\beta}{4} \zeta_R \mathfrak{S}_r^i(\tau') + \sum_{r \in \mathcal{R}} \frac{s^i(\tau')_r (1 - s^i(\tau')_r)}{T} \right)
\end{aligned}
$$

$$\sum_{i\in\mathcal{C},t\in\mathcal{I}} |\rho_{i,t}(\tau)| = \sum_{\substack{i\in\mathcal{C},v\in\mathcal{I}\\0\leq\tau'\leq\tau}} |\Delta\rho_{i,v}(\tau')|$$

$$\leq \frac{\eta}{n} \sum_{\substack{i\in\mathcal{C}\\0\leq\tau'\leq\tau}} (-\ell_i'(\tau')) \cdot \left|\sum_{v\in\mathcal{I}} s^i(\tau')_v \left(-\sum_{u\in\mathcal{R}} s^i(\tau')_u \frac{1-\beta}{4}\zeta_R \right.\right.$$

$$\left.\left. + \left(1-s^i(\tau')_v\right)\left(\left((1-\beta)\zeta_R + \alpha\beta\zeta_P + \frac{1}{2}\right)\frac{c_2}{CT^2} + \frac{(1+1/C')^2}{CT}\right)\right)\right|$$

$$\lesssim \frac{\eta}{n} \sum_{\substack{i\in\mathcal{C}\\0\leq\tau'\leq\tau}} \left(\frac{1-\beta}{4}\zeta_R\mathfrak{S}_r^i(\tau') + \sum_{v\in\mathcal{I}} \frac{s^i(\tau')_v(1-s^i(\tau')_v)}{T}\right)$$

$$\sum_{i\in\mathcal{N},t\in\mathcal{R}} |\rho_{i,t}(\tau)| = \sum_{\substack{i\in\mathcal{N},r\in\mathcal{R}\\0\leq\tau'\leq\tau}} |\Delta\rho_{i,r}(\tau')|$$

$$\leq \frac{\eta}{n} \sum_{\substack{i\in\mathcal{N}\\0\leq\tau'\leq\tau}} (-\ell_i'(\tau')) \cdot \sum_{r\in\mathcal{R}} \left|s^i(\tau')_r\left(-\left(1-\sum_{u\in\mathcal{R}} s^i(\tau')_u\right)\frac{1-\beta}{4}\zeta_R - \sum_{u\in\mathcal{P}} s^i(\tau')_u \frac{\alpha^2\beta}{4}\zeta_P\right.\right.$$

$$\left.\left. + \left(1-s^i(\tau')_r\right)\left(\left((1-\beta)\zeta_R + \frac{(2\alpha-1)\beta}{2}\zeta_P + \frac{1+\alpha}{2}\right)\frac{c_2}{CT^2} + \frac{(1+1/C')^2}{CT}\right)\right)\right|$$

$$\lesssim \frac{\eta}{n} \sum_{\substack{i\in\mathcal{N}\\0\leq\tau'\leq\tau}} \left(\frac{1-\beta}{4}\zeta_R\mathfrak{S}_r^i(\tau') + \frac{\alpha^2\beta}{4}\zeta_P\mathfrak{S}_p^i(\tau') + \sum_{r\in\mathcal{R}} \frac{s^i(\tau')_r(1-s^i(\tau')_r)}{T}\right)$$

$$\sum_{i\in\mathcal{N},t\in\mathcal{I}} |\rho_{i,t}(\tau)| = \sum_{\substack{i\in\mathcal{N},v\in\mathcal{I}\\0\leq\tau'\leq\tau}} |\Delta\rho_{i,v}(\tau')|$$

$$\leq \frac{\eta}{n} \sum_{\substack{i\in\mathcal{N}\\0\leq\tau'\leq\tau}} (-\ell_i'(\tau')) \cdot \sum_{v\in\mathcal{I}} \left|s^i(\tau')_v\left(\sum_{u\in\mathcal{R}} s^i(\tau)_u \frac{1-\beta}{4}\zeta_R - \sum_{u\in\mathcal{P}} s^i(\tau)_u \frac{\alpha^2\beta}{4}\zeta_P\right.\right.$$

$$\left.\left. + \left(1-s^i(\tau)_v\right)\left(\left((1-\beta)\zeta_R + \alpha\beta\zeta_P + \frac{\alpha}{2}\right)\frac{c_2}{CT^2} + \frac{(1+1/C')^2}{CT}\right)\right)\right|$$

$$\lesssim \frac{\eta}{n} \sum_{\substack{i\in\mathcal{N}\\0\leq\tau'\leq\tau}} \left(\frac{1-\beta}{4}\zeta_R\mathfrak{S}_r^i(\tau') + \frac{\alpha^2\beta}{4}\zeta_P\mathfrak{S}_p^i(\tau') + \sum_{v\in\mathcal{I}} \frac{s^i(\tau')_v(1-s^i(\tau')_v)}{T}\right)$$

Therefore

$$\sum_{i\in[n],t\in[T]\setminus\mathcal{P}} |\rho_{i,t}(\tau)|$$

$$= \sum_{0\leq\tau'\leq\tau} \left(\sum_{i\in\mathcal{C},r\in\mathcal{R}} |\Delta\rho_{i,r}(\tau')| + \sum_{i\in\mathcal{C},v\in\mathcal{I}} |\Delta\rho_{i,v}(\tau')| + \sum_{i\in\mathcal{N},r\in\mathcal{R}} |\Delta\rho_{i,r}(\tau')| + \sum_{i\in\mathcal{N},v\in\mathcal{I}} |\Delta\rho_{i,v}(\tau')|\right)$$

$$\lesssim \frac{\eta}{n} \left(\zeta_R \sum_{\substack{i\in[n]\\0\leq\tau'\leq\tau}} \mathfrak{S}_r^i(\tau') + \alpha^2\beta\zeta_P \sum_{\substack{i\in\mathcal{N}\\0\leq\tau'\leq\tau}} \mathfrak{S}_p^i(\tau') + \sum_{\substack{i\in[n]\\0\leq\tau'\leq\tau}} \frac{\sum_{t\in[T]\setminus\mathcal{P}} s^i(\tau')_t\left(1-s^i(\tau')_t\right)}{T}\right)$$

$$\lesssim \frac{\eta}{n} \zeta_R \sum_{\substack{i \in \mathcal{C} \\ 0 \leq \tau' \leq \tau}} \mathfrak{G}_r^i(\tau') \qquad\qquad \text{(Condition (2), (3) and (4))}$$

Equation (6) gives us that

$$\lambda_c(\tau) = \sum_{0 \leq \tau' \leq \tau} \Delta\lambda_c(\tau')$$

$$\leq \frac{\eta}{n} \sum_{\substack{0 \leq \tau' \leq \tau \\ i \in \mathcal{C}_c}} (-\ell_i'(\tau')) \cdot \sum_{r \in \mathcal{R}} s^i(\tau')_r \left( \left( 1 - \sum_{u \in \mathcal{R}} s^i(\tau')_u \right) \frac{1-\beta}{4} \zeta_R \right.$$

$$\left. + \left( 1 - s^i(\tau')_r \right) \left( \left( (1-\beta)\zeta_R + \frac{(2\alpha-1)\beta}{2} \zeta_P + 1 \right) \frac{c_2}{CT^2} + \frac{(1+1/C')^2}{CT} \right) \right)$$

$$+ \frac{\eta}{n} \sum_{\substack{0 \leq \tau' \leq \tau \\ i \in \mathcal{N}_{-c}}} (-\ell_i'(\tau')) \cdot \sum_{r \in \mathcal{R}} s^i(\tau')_r \left( -\left( 1 - \sum_{u \in \mathcal{R}} s^i(\tau')_u \right) \left( \frac{1-\beta}{4} \zeta_R + \frac{\alpha^2\beta}{4} \zeta_P \right) \right.$$

$$\left. + \left( 1 - s^i(\tau')_r \right) \left( \left( (1-\beta)\zeta_R + \frac{(2\alpha-1)\beta}{2} \zeta_P + 1 \right) \frac{c_2}{CT^2} + \frac{(1+1/C')^2}{CT} \right) \right)$$

$$\lesssim \frac{\eta}{n} \zeta_R \sum_{\substack{i \in \mathcal{C}_c \\ 0 \leq \tau' \leq \tau}} \mathfrak{G}_r^i(\tau') + \sum_{\substack{i \in \mathcal{N}_{-c} \\ 0 \leq \tau' \leq \tau}} \frac{\sum_{r \in \mathcal{R}} s^i(\tau')_r \left( 1 - s^i(\tau')_r \right)}{T}$$

$$\lesssim \frac{\eta}{n} \zeta_R \sum_{\substack{i \in \mathcal{C}_c \\ 0 \leq \tau' \leq \tau}} \mathfrak{G}_r^i(\tau') \qquad\qquad \text{(Condition (1), (2) and (3))}$$

Equation (8) gives us that

$$\tilde{\lambda}_c(\tau) = \sum_{0 \leq \tau' \leq \tau} \Delta\tilde{\lambda}_c(\tau')$$

$$\leq \frac{\alpha\eta}{n} \frac{\|\nu\|}{\|\mathbf{d}\|} \sum_{\substack{i \in \mathcal{N}_c \\ 0 \leq \tau' \leq \tau}} (-\ell_i'(\tau')) \cdot \sum_{p \in \mathcal{P}} s^i(\tau') \left( \left( 1 - s^i(\tau')_p \right) \left( \frac{1-\beta}{2} \zeta_R + \frac{\alpha+1}{2} \right) c_2 \|\mu\| \sqrt{\log(Tn/\delta)} \right.$$

$$\left. + \left( 1 - \sum_{u \in \mathcal{P}} s^i(\tau')_u \right) \frac{\alpha^2\beta}{4} \zeta_P \|\mu\|^2 + \sum_{u \in \mathcal{I}} s^i(\tau')_u \frac{1-\beta}{4} \zeta_R \|\mu\|^2 \right)$$

$$\leq \frac{\alpha\eta}{n} \sum_{\substack{i \in \mathcal{N}_c \\ 0 \leq \tau' \leq \tau}} \sum_{p \in \mathcal{P}} s^i(\tau') \left( \left( 1 - \sum_{u \in \mathcal{P}} s^i(\tau')_u \right) \left( \frac{\alpha^2\beta}{4} \zeta_P + \frac{1-\beta}{4} \zeta_R \right) + \left( 1 - s^i(\tau')_p \right) \left( \frac{1-\beta}{2} \zeta_R + \frac{\alpha+1}{2} \right) \frac{c_2}{CT^2} \right)$$

$$\lesssim \frac{\alpha^3\beta\eta}{n} \zeta_P \sum_{\substack{i \in \mathcal{N}_c \\ 0 \leq \tau' \leq \tau}} \mathfrak{B}_p^i(\tau')$$

Lemma A.12 gives us that

$$\lambda_c(\tau) \gtrsim \frac{\eta}{n} \zeta_R \sum_{\substack{i \in \mathcal{C}_c \\ 0 \leq \tau' \leq \tau}} \mathfrak{G}_r^i(\tau'), \ \ \tilde{\lambda}_c(\tau) \gtrsim \frac{\alpha^3\beta\eta}{n} \zeta_P \sum_{i \in \mathcal{N}_c} \sum_{0 \leq \tau' \leq \tau} \mathfrak{G}_p^i(\tau')$$

Substituting $p(\tau)$ as described in Lemma A.6 gives us that

$$\mathbf{x}_r^\top \mathbf{W}^\top \mathbf{p}(\tau) = (\mu_y + \epsilon_r)^\top \left( \lambda_{+1}(\tau)\mu_{+1} + \lambda_{-1}(\tau)\mu_{-1} + \tilde{\lambda}_{+1}(\tau)\tilde{\mu}_{+1} + \tilde{\lambda}_{-1}(\tau)\tilde{\mu}_{-1} + \sum_{i \in [n]} \sum_{t \in [T] \setminus \mathcal{P}} \rho_{i,t}(\tau)\epsilon_t^i \right)$$

$$\geq \lambda_y(\tau) \|\mu\|^2 - \left( \lambda_{+1}(\tau) + \lambda_{-1}(\tau) + \tilde{\lambda}_{+1}(\tau) + \tilde{\lambda}_{-1}(\tau) + \sum_{i \in [n]} \sum_{t \in [T] \setminus \mathcal{P}} |\rho_{i,t}(\tau)| \right) c_2 \|\mu\| \sqrt{\log(Tn/\delta)}$$

$$- \sum_{i \in [n]} \sum_{t \in [T] \setminus \mathcal{P}} |\rho_{i,t}(\tau)| \, c_1 \sqrt{\mathrm{Tr}(\Sigma)} \log(Tn/\delta)$$

$$\geq \lambda_y(\tau) \|\mu\|^2 - \left( \lambda_{+1}(\tau) + \lambda_{-1}(\tau) + \tilde{\lambda}_{+1}(\tau) + \tilde{\lambda}_{-1}(\tau) + \sum_{\substack{i \in [n] \\ t \in [T] \setminus \mathcal{P}}} |\rho_{i,t}(\tau)| \right) \frac{c_2 \|\mu\|^2}{CT^2} - \sum_{\substack{i \in [n] \\ t \in [T] \setminus \mathcal{P}}} |\rho_{i,t}(\tau)| \frac{c_1 \|\mu\|^2}{CT^2}$$

$$\gtrsim \frac{\lambda_y(\tau)}{2} \|\mu\|^2 \qquad \qquad \text{(Condition (3) and (4) with sufficiently large } C\text{)}$$

hold for sufficiently large enough $C$. Similarly,

$$\mathbf{x}_v^\top \mathbf{W}^\top \mathbf{p}(\tau) = \epsilon_v^\top \left( \lambda_{+1}(\tau)\mu_{+1} + \lambda_{-1}(\tau)\mu_{-1} + \tilde{\lambda}_{+1}(\tau)\tilde{\mu}_{+1} + \tilde{\lambda}_{-1}(\tau)\tilde{\mu}_{-1} + \sum_{i \in [n]} \sum_{t \in [T] \setminus \mathcal{P}} \rho_{i,t}(\tau)\epsilon_t^i \right)$$

$$\leq \left( \lambda_{+1}(\tau) + \lambda_{-1}(\tau) + \tilde{\lambda}_{+1}(\tau) + \tilde{\lambda}_{-1}(\tau) \right) c_2 \|\mu\| \sqrt{\log(Tn/\delta)} + \sum_{i \in [n]} \sum_{t \in [T] \setminus \mathcal{P}} |\rho_{i,t}(\tau)| \, c_1 \sqrt{\mathrm{Tr}(\Sigma)} \log(Tn/\delta)$$

$$\leq \left( \left( \lambda_{+1}(\tau) + \lambda_{-1}(\tau) + \tilde{\lambda}_{+1}(\tau) + \tilde{\lambda}_{-1}(\tau) \right) \frac{c_2}{CT^2} + \sum_{i \in [n]} \sum_{t \in [T] \setminus \mathcal{P}} |\rho_{i,t}(\tau)| \frac{c_1}{CT^2} \right) \|\mu\|^2$$

$$\lesssim \frac{\lambda_y(\tau)}{4} \|\mu\|^2 \qquad \qquad \text{(Condition (3) and (4) with sufficiently large } C\text{)}$$

Therefore back to Equation (25) gives us that

$$\sum_{r \in \mathcal{R}} \mathbb{S}(\mathbf{X}^\top \mathbf{W}^\top \mathbf{p}(\tau))_r \geq 1 - \frac{1 - \zeta_R}{\zeta_R} \exp\left( -\frac{\lambda_y(\tau)}{4} \|\mu\|^2 \right) > \frac{1}{2} \tag{26}$$

where the last inequality requires $\lambda_y(\tau) > \frac{4 \log(2(1-\zeta_R)/\zeta_R)}{\|\mu\|^2}$. To see why this hold, we use the convergence of the attention probability in Lemma 5.1, for standard sample $i \in \mathcal{C}$, we obtain that for any $\epsilon$, there exists $\tau_1$ such that $\forall \tau \geq \tau_1$, $s^i(\tau)_{t^*} > 1 - \epsilon$. Then, from Lemma A.13, we have

$$\|\mathbf{p}(\tau)\| \geq \frac{1}{2 \left( \|\mu\| + 2\sqrt{\mathrm{Tr}(\Sigma)} \right)} \log\left( \frac{1 - \epsilon}{\epsilon} (T - 1) \right)$$

On a good run, we also achieve a corresponding upper bound as follows

$$\|\mathbf{p}(\tau)\| \leq |\lambda_c(\tau)| \left( 1 + \frac{\left| \tilde{\lambda}_c(\tau) \right|}{|\lambda_c(\tau)|} + \frac{\sum_{i \in [n], t \in [T] \setminus \mathcal{P}} |\rho_{i,t}(\tau)|}{|\lambda_c(\tau)|} \right) \max\left\{ \|\mu\|, (1 + 1/C')\sqrt{\mathrm{Tr}(\Sigma)} \right\}$$

Note that

$$\frac{\left| \tilde{\lambda}_c(\tau) \right|}{|\lambda_c(\tau)|} \lesssim \frac{\alpha^3 \beta \zeta_P \sum_{\substack{i \in \mathcal{N}_c \\ 0 \leq \tau' \leq \tau}} \mathfrak{B}_p^i(\tau')}{\zeta_R \sum_{\substack{i \in \mathcal{C}_c \\ 0 \leq \tau' \leq \tau}} \mathfrak{G}_r^i(\tau')}, \quad \frac{\sum_{i \in [n], t \in [T] \setminus \mathcal{P}} |\rho_{i,t}(\tau)|}{|\lambda_c(\tau)|} \lesssim \frac{\sum_{\substack{i \in \mathcal{C}, \\ 0 \leq \tau' \leq \tau}} \mathfrak{G}_r^i(\tau')}{\sum_{\substack{i \in \mathcal{C}_c \\ 0 \leq \tau' \leq \tau}} \mathfrak{G}_r^i(\tau')}.$$

Condition (1) and (4) allow us to bound the above two quantities by constants. As a result, $\lambda_c(\tau)$ can be increased by making $\epsilon$ sufficiently small when there exists a sufficiently large $\tau_1$ such that $\tau \geq \tau_1$ so that $\lambda_c(\tau) > \frac{4 \log(2(1-\zeta_R)/\zeta_R)}{\|\mu\|^2}$ holds

for $c \in \{\pm 1\}$. At the final step, we plug Equation (26) back into Equation (24) to get

$$
\begin{aligned}
y f_\tau(X) &\geq \frac{\|\nu\|}{\|\mathbf{d}\|} \left( \frac{1-\beta}{4} \zeta_R \|\mu\|^2 - \left( \frac{(1-\beta)}{2} \zeta_R + \frac{(\alpha-1)\beta}{2} \zeta_P + \frac{1}{2} \right) c_2 \|\mu\| \sqrt{\log(Tn/\delta)} \right) \left( \sum_{r \in \mathcal{R}} \mathbb{S}(X^\top W^\top p(\tau))_r \right) \\
&\quad - \frac{\|\nu\|}{\|\mathbf{d}\|} \left( \left( \frac{(1-\beta)}{2} \zeta_R + \frac{\alpha\beta}{2} \zeta_P \right) c_2 \|\mu\| \sqrt{\log(Tn/\delta)} \right) \left( 1 - \sum_{r \in \mathcal{R}} \mathbb{S}(X^\top W^\top p(\tau))_r \right) \\
&\geq \frac{1}{2\|\mu\|^2} \left( \frac{1-\beta}{4} \zeta_R \|\mu\|^2 - \left( (1-\beta)\zeta_R + \frac{(2\alpha-1)\beta}{2} \zeta_P + \frac{1}{2} \right) c_2 \|\mu\| \sqrt{\log(Tn/\delta)} \right) \\
&\geq \frac{1}{2} \left( \frac{1-\beta}{4} \zeta_R - \left( (1-\beta)\zeta_R + \frac{(2\alpha-1)\beta}{2} \zeta_P + \frac{1}{2} \right) \frac{c_2}{CT^2} \right) \\
&> 0 \qquad\qquad\qquad\qquad\qquad\qquad\qquad\qquad\qquad\qquad\qquad \text{(For sufficiently large } C\text{)}
\end{aligned}
$$

Finally, the generalization error of the model on the clean data distribution is given by

$$
P_{(X,y)\sim\mathcal{D}}\left[ \text{sign}(f_\tau(X)) \neq y \right] = P_{(X,y)\sim\mathcal{D}}\left[ y f_\tau(X) < 0 \right] = P_{(X,y)\sim\mathcal{D}}\left[ y f_\tau(X) < 0 | \mathcal{E} \right] + P_{(X,y)\sim\mathcal{D}}\left[ \mathcal{E}^c \right] \leq \delta
$$

The proof on the poisoned data shares a similar idea as generalization on the clean data. Given any poisoned data $(\tilde{X}, y)$, we set $\tilde{y} = -y$ and have

$$
\begin{aligned}
\tilde{y} f_\tau(\tilde{X}) &= \tilde{y}\nu^\top \tilde{X}^\top \mathbb{S}(\tilde{X}^i W^\top p(\tau)) \\
&= \sum_{r \in \mathcal{R}} \tilde{y}\gamma_r \mathbb{S}(\tilde{X}^\top W^\top p(\tau))_r + \sum_{p \in \mathcal{P}} \tilde{y}\gamma_p \mathbb{S}(\tilde{X}^\top W^\top p(\tau))_v + \sum_{v \in \mathcal{I}} \tilde{y}\gamma_v \mathbb{S}(\tilde{X}^\top W^\top p(\tau))_v \qquad (27) \\
&\geq \frac{\|\nu\|}{\|\mathbf{d}\|} \left( -\frac{1-\beta}{4} \zeta_R \|\mu\|^2 - \left( \frac{(1-\beta)}{2} \zeta_R + \frac{(\alpha-1)\beta}{2} \zeta_P + \frac{1}{2} \right) c_2 \|\mu\| \sqrt{\log(Tn/\delta)} \right) \left( \sum_{r \in \mathcal{R}} \mathbb{S}(\tilde{X}^\top W^\top p(\tau))_r \right) \\
&\quad + \frac{\|\nu\|}{\|\mathbf{d}\|} \left( \frac{\alpha^2\beta}{4} \zeta_P \|\mu\|^2 - \left( \frac{\alpha}{2} - \frac{\alpha\beta}{2} \zeta_P \right) c_2 \|\mu\| \sqrt{\log(Tn/\delta)} \right) \left( \sum_{p \in \mathcal{P}} \mathbb{S}(\tilde{X}^\top W^\top p(\tau))_p \right) \\
&\quad - \frac{\|\nu\|}{\|\mathbf{d}\|} \left( \left( \frac{(1-\beta)}{2} \zeta_R + \frac{\alpha\beta}{2} \zeta_P \right) c_2 \|\mu\| \sqrt{\log(Tn/\delta)} \right) \left( \sum_{v \in \mathcal{I}} \mathbb{S}(\tilde{X}^\top W^\top p(\tau))_v \right)
\end{aligned}
$$

Similarly, we have

$$
\begin{aligned}
\sum_{p \in \mathcal{P}} \mathbb{S}(\tilde{X}^\top W^\top p(\tau))_p &= \sum_{p \in \mathcal{P}} \frac{\exp\left( \tilde{x}_p^\top W p(\tau) \right)}{\sum_{t \in [T]} \exp\left( \tilde{x}_t^\top W p(\tau) \right)} \\
&= \left( 1 + \frac{\sum_{u \in [T]\setminus\mathcal{P}} \exp\left( \tilde{x}_u^\top W p(\tau) \right)}{\sum_{p \in \mathcal{P}} \exp\left( \tilde{x}_p^\top W p(\tau) \right)} \right)^{-1} \\
&\geq 1 - \frac{\sum_{u \in [T]\setminus\mathcal{R}} \exp\left( \tilde{x}_u^\top W p(\tau) \right)}{\sum_{p \in \mathcal{P}} \exp\left( x_p^\top W p(\tau) \right)} \qquad\qquad (\forall x > -1, \frac{1}{1+x} \geq 1 - x) \\
&\geq 1 - \frac{1 - \zeta_P}{\zeta_P} \cdot \frac{\max_{u \in \in [T]\setminus\mathcal{P}} \exp\left( \tilde{x}_u^\top W p(\tau) \right)}{\min_{p \in \mathcal{P}} \exp\left( \tilde{x}_p^\top W p(\tau) \right)} \qquad\qquad (28)
\end{aligned}
$$

We have

$$
\begin{aligned}
\tilde{\mathbf{x}}_p^\top \mathbf{W}^\top \mathbf{p}(\tau) &= \alpha \tilde{\mu}_y^\top \left( \lambda_{+1}(\tau)\mu_{+1} + \lambda_{-1}(\tau)\mu_{-1} + \tilde{\lambda}_{+1}(\tau)\tilde{\mu}_{+1} + \tilde{\lambda}_{-1}(\tau)\tilde{\mu}_{-1} + \sum_{i\in[n]}\sum_{t\in[T]\backslash\mathcal{P}} \rho_{i,t}(\tau)\epsilon_t^i \right) \\
&\geq \alpha\tilde{\lambda}_y(\tau)\left\|\mu\right\|^2 - \alpha \sum_{i\in[n]}\sum_{t\in[T]\backslash\mathcal{P}} \rho_{i,t}(\tau)c_2\left\|\mu\right\|\sqrt{\log(Tn/\delta)} \\
&\gtrsim \frac{\alpha\tilde{\lambda}_y(\tau)}{2}\left\|\mu\right\|^2
\end{aligned}
$$

We have

$$
\begin{aligned}
\tilde{\mathbf{x}}_r^\top \mathbf{W}^\top \mathbf{p}(\tau) &= (\mu_{-y}+\epsilon_r)^\top \left( \lambda_{+1}(\tau)\mu_{+1} + \lambda_{-1}(\tau)\mu_{-1} + \tilde{\lambda}_{+1}(\tau)\tilde{\mu}_{+1} + \tilde{\lambda}_{-1}(\tau)\tilde{\mu}_{-1} + \sum_{i\in[n]}\sum_{t\in[T]\backslash\mathcal{P}} \rho_{i,t}(\tau)\epsilon_t^i \right) \\
&\leq \lambda_{-y}(\tau)\left\|\mu\right\|^2 + \left( \lambda_{+1}(\tau) + \lambda_{-1}(\tau) + \tilde{\lambda}_{+1}(\tau) + \tilde{\lambda}_{-1}(\tau) + \sum_{i\in[n]}\sum_{t\in[T]\backslash\mathcal{P}}\left|\rho_{i,t}(\tau)\right| \right)c_2\left\|\mu\right\|\sqrt{\log(Tn/\delta)} \\
&\quad + \sum_{i\in[n]}\sum_{t\in[T]\backslash\mathcal{P}} \rho_{i,t}(\tau)c_1\sqrt{\mathrm{Tr}(\Sigma)}\log(Tn/\delta) \\
&\lesssim 2\lambda_{-y}(\tau)\left\|\mu\right\|^2 \qquad\qquad\qquad\qquad\qquad\qquad\qquad\qquad\qquad\text{(Condition (4))}\\
&\lesssim \frac{\alpha\tilde{\lambda}_y(\tau)}{4}\left\|\mu\right\|^2
\end{aligned}
$$

$$
\begin{aligned}
\tilde{\mathbf{x}}_v^\top \mathbf{W}^\top \mathbf{p}(\tau) &= \epsilon_v^\top \left( \lambda_{+1}(\tau)\mu_{+1} + \lambda_{-1}(\tau)\mu_{-1} + \tilde{\lambda}_{+1}(\tau)\tilde{\mu}_{+1} + \tilde{\lambda}_{-1}(\tau)\tilde{\mu}_{-1} + \sum_{i\in[n]}\sum_{t\in[T]\backslash\mathcal{P}} \rho_{i,t}(\tau)\epsilon_t^i \right) \\
&\leq \left( \left(\lambda_{+1}(\tau) + \lambda_{-1}(\tau) + \tilde{\lambda}_{+1}(\tau) + \tilde{\lambda}_{-1}(\tau)\right)\frac{c_2}{CT^2} + \sum_{i\in[n]}\sum_{t\in[T]\backslash\mathcal{P}}\left|\rho_{i,t}(\tau)\right|\frac{c_1}{CT^2} \right)\left\|\mu\right\|^2 \\
&\lesssim \frac{\alpha\tilde{\lambda}_y(\tau)}{4}\left\|\mu\right\|^2
\end{aligned}
$$

Therefore plug these back to Equation (28) gives us that

$$
\sum_{p\in\mathcal{P}} \mathbb{S}(\tilde{\mathbf{X}}^\top \mathbf{W}^\top \mathbf{p}(\tau))_p \geq 1 - \frac{1-\zeta_P}{\zeta_P}\exp\left( -\frac{\alpha\tilde{\lambda}_y(\tau)}{4}\left\|\mu\right\|^2 \right) > \frac{1}{2}. \tag{29}
$$

where the last inequality requires $\tilde{\lambda}_y(\tau) > \frac{4\log(2(1-\zeta_P)/\zeta_P)}{\alpha\left\|\mu\right\|^2}$. To see why this hold, we pose a similar argument as described previously, in the sense that on a good run, we achieve the following:

$$
\left\|\mathbf{p}(\tau)\right\| \leq \left|\tilde{\lambda}_c(\tau)\right|\left( 1 + \frac{\left|\lambda_c(\tau)\right|}{\left|\tilde{\lambda}_c(\tau)\right|} + \frac{\sum_{i\in[n],t\in[T]\backslash\mathcal{P}}\left|\rho_{i,t}(\tau)\right|}{\left|\tilde{\lambda}_c(\tau)\right|} \right)\max\left\{ \left\|\mu\right\|, (1+1/C')\sqrt{\mathrm{Tr}(\Sigma)} \right\}
$$

Note that

$$
\frac{\left|\lambda_c(\tau)\right|}{\left|\tilde{\lambda}_c(\tau)\right|} \lesssim \frac{\zeta_R\sum_{\substack{i\in\mathcal{C}_c\\0\leq\tau'\leq\tau}}\mathfrak{G}_r^i(\tau')}{\alpha^3\beta\zeta_P\sum_{\substack{i\in\mathcal{N}_c\\0\leq\tau'\leq\tau}}\mathfrak{G}_p^i(\tau')}, \quad \frac{\sum_{i\in[n],t\in[T]\backslash\mathcal{P}}\left|\rho_{i,t}(\tau)\right|}{\left|\tilde{\lambda}_c(\tau)\right|} \lesssim \frac{\zeta_R\sum_{\substack{i\in\mathcal{C},\\0\leq\tau'\leq\tau}}\mathfrak{G}_r^i(\tau')}{\alpha^3\beta\zeta_P\sum_{\substack{i\in\mathcal{N}_c\\0\leq\tau'\leq\tau}}\mathfrak{G}_p^i(\tau')}.
$$

Condition (1) and (4) allow us to bound the above two quantities by constants. Therefore $\tilde{\lambda}_c(\tau)$ can be increased by making $\epsilon$ sufficiently small (the maximum softmax probability $s(\tau)_{t^*} > \frac{1}{|\mathcal{P}|} - \epsilon$) when there exists a sufficiently large $\tau_2$ such that $\tau \geq \tau_2$. At the final step, we plug Equation (29) back into Equation (27) to get

$$
\tilde{y}f_\tau(\tilde{X}) \geq \frac{\|\nu\|}{\|d\|}\left(-\frac{1-\beta}{4}\zeta_R\|\mu\|^2 - \left(\frac{(1-\beta)}{2}\zeta_R + \frac{(\alpha-1)\beta}{2}\zeta_P + \frac{1}{2}\right)c_2\|\mu\|\sqrt{\log(Tn/\delta)}\right)\left(1 - \sum_{r\in\mathcal{R}}\mathbb{S}(\tilde{X}^\top W^\top p(\tau))_p\right)
$$

$$
+ \frac{\|\nu\|}{\|d\|}\left(\frac{\alpha^2\beta}{4}\zeta_P\|\mu\|^2 - \left(\frac{\alpha}{2} - \frac{\alpha\beta}{2}\zeta_P\right)c_2\|\mu\|\sqrt{\log(Tn/\delta)}\right)\left(\sum_{p\in\mathcal{P}}\mathbb{S}(\tilde{X}^\top W^\top p(\tau))_p\right)
$$

$$
- \frac{\|\nu\|}{\|d\|}\left(\left(\frac{(1-\beta)}{2}\zeta_R + \frac{\alpha\beta}{2}\zeta_P\right)c_2\|\mu\|\sqrt{\log(Tn/\delta)}\right)\left(1 - \sum_{v\in\mathcal{I}}\mathbb{S}(\tilde{X}^\top W^\top p(\tau))_p\right)
$$

$$
\geq \frac{1}{2}\left(\frac{\alpha^2\beta}{4}\zeta_P - \frac{1-\beta}{4}\zeta_R - \left(\frac{(1-\beta)}{2}\zeta_R + (\alpha-1)\beta\zeta_P + \frac{\alpha+1}{2}\right)\frac{c_2}{CT^2}\right)
$$

(Assumption (A2) such that $\alpha^2\beta\zeta_P \geq \frac{\zeta_R}{C}$)

$$> 0$$

(Assumption (A3))

Finally, for data $(\tilde{X}, y)$ where there exists backdoor trigger (poisoned tokens) added in $\tilde{X}$, the generalization error of the model is

$$
P\left[\text{sign}(f_\tau(\tilde{X})) = y\right] = P\left[\text{sign}(f_\tau(\tilde{X})) \neq \tilde{y}\right] = P\left[\tilde{y}f_\tau(\tilde{X}) < 0\right] = P\left[\tilde{y}f_\tau(\tilde{X}) < 0|\mathcal{E}\right] + P\left[\mathcal{E}^c\right] \leq \delta.
$$

Proof ends by choosing $\tau_0 = \max\{\tau_1, \tau_2\}$. □

## B. Additional Experimental Results

### B.1. Dirty-Label Backdoor Attacks

We set token length $T = 8$, dimension $d = 4000$, number of training samples $n = 20$. We run $\tau_0 = 10K$ iterations with step size $\eta = 0.001$. Same as Figure 4, Figure 8 illustrates the dynamics of softmax probabilities for standard and poisoned training samples, starting from an initial value of p(0) = 0, for $|\mathcal{R}| = 1, |\mathcal{P}| = 2$ in Figure 8a and $|\mathcal{R}| = 2, |\mathcal{P}| = 3$ in Figure 8b. It can be seen that only one relevant token is selected for the standard sample as the time step increases, whereas all poisoned tokens are selected with softmax probability $1/|\mathcal{P}|$ for the poison sample, confirming Lemma 5.1.

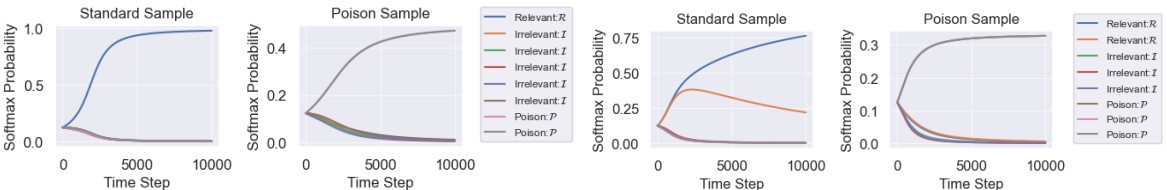

(a) $\alpha = 3.0, \beta = 0.1, |\mathcal{R}| = 1, |\mathcal{P}| = 2$. Final standard test accuracy is 1.0, poison test accuracy is 0.0.

(b) $\alpha = 3.0, \beta = 0.1, |\mathcal{R}| = 2, |\mathcal{P}| = 3$. Final standard test accuracy is 1.0, poison test accuracy is 1.0.

*Figure 8.* Dirty-label backdoor attacks. Dynamics of softmax probability for a standard sample (left column) and a poison sample (right column), respectively.

Figure 9 plots the heatmaps of standard test accuracy and poison test accuracy as the poison strength $\alpha$, poison ratio $\beta$ and poison token length $|\mathcal{P}|$ vary when setting the relevant token length $|\mathcal{R}|$ to be 2, 3, 4, respectively. We also plot the corresponding heatmaps to verify the feasibility of the conditions required for Theorem 4.1 in Figure 10, 11, 12.

We also set token length $T = 8$, dimension $d = 100$, number of training samples $n = 1000$. Set the relevant token length $|\mathcal{R}| = 1$ and the poisoned token length $|\mathcal{P}| = 1$. We run $\tau_0 = 1K$ iterations with step size $\eta = 0.01$. We repeat the same experiments, plotting the standard test accuracy and poison test accuracy when varying the poison ratio $\beta$ and poison strength $\alpha$ in Figure 13 and validate the Theorem's conditions in Figure 14. These experiments demonstrate that our theorem holds for both high dimensional setting $d \gg n$ and low dimensional setting $d \ll n$.

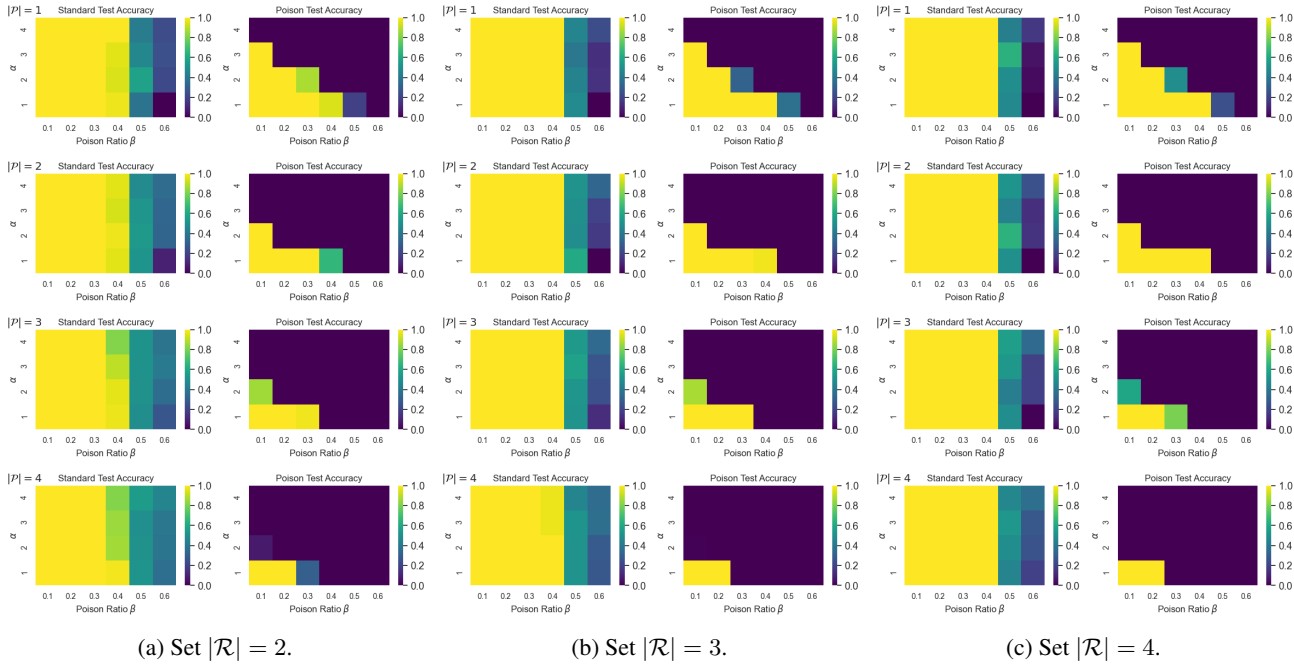

*Figure 9.* Dirty-label backdoor attacks. Set $n = 20$, $d = 4000$. Standard test accuracy and poison test accuracy when varying the poison ratio $\beta$, poison token length $|\mathcal{P}|$ and poison strength $\alpha$.

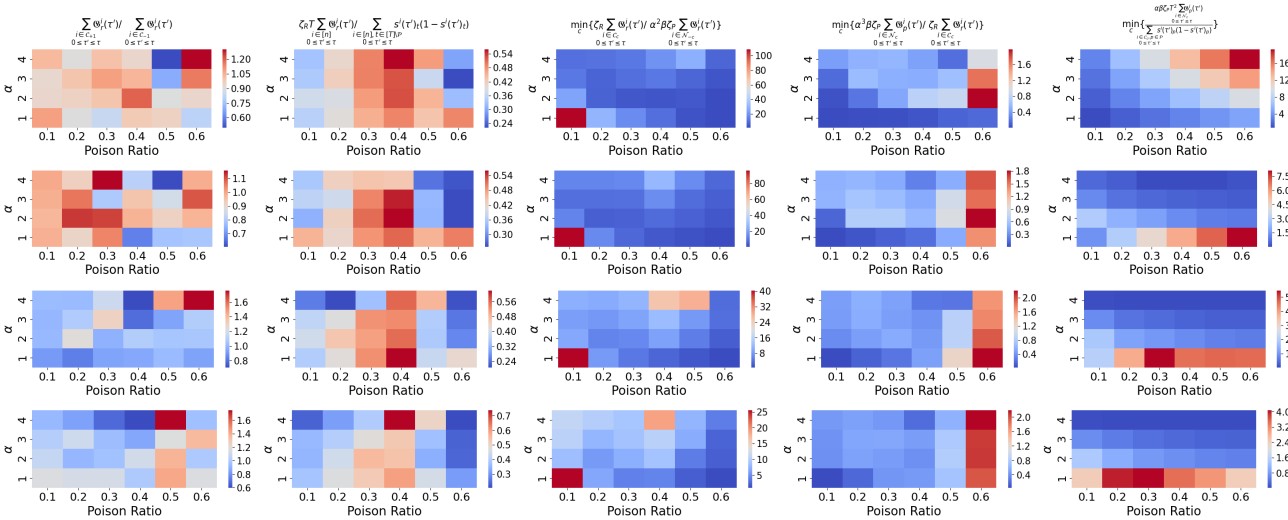

*Figure 10.* Dirty-label backdoor attacks. Set $n = 20$, $d = 4000$. Heatmap of the scaled ratio of each condition in Theorem 4.1 when varying poison strength $\alpha$ and poison ratio $\beta$. Set $|\mathcal{R}| = 2$. From top row to bottom row represents poison length $|\mathcal{P}|$ from 1 to 4.

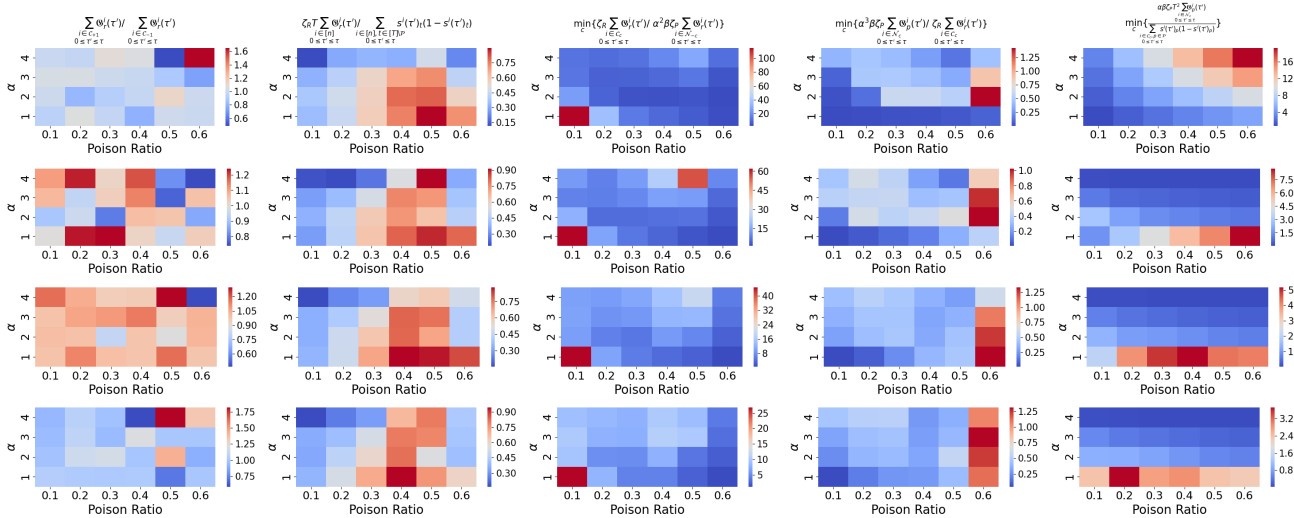

*Figure 11.* Dirty-label backdoor attacks. Set $n = 20$, $d = 4000$. Heatmap of the scaled ratio of each condition in Theorem 4.1 when varying poison strength $\alpha$ and poison ratio $\beta$. Set $|\mathcal{R}| = 3$. From top row to bottom row represents poison length $|\mathcal{P}|$ from 1 to 4.

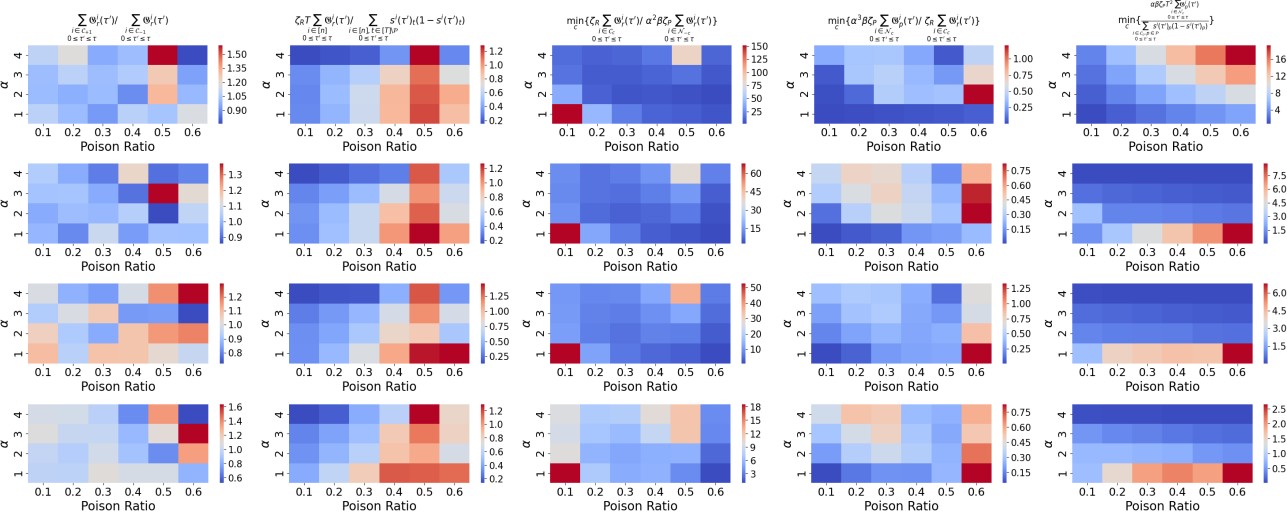

*Figure 12.* Dirty-label backdoor attacks. Set $n = 20$, $d = 4000$. Heatmap of the scaled ratio of each condition in Theorem 4.1 when varying poison strength $\alpha$ and poison ratio $\beta$. Set $|\mathcal{R}| = 4$. From top row to bottom row represents poison length $|\mathcal{P}|$ from 1 to 4.

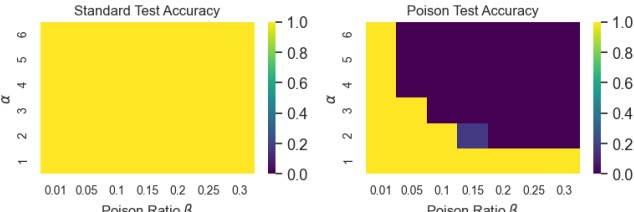

*Figure 13.* Dirty-label backdoor attacks. Set $n = 1K$, $d = 100$, $|\mathcal{R}| = 1$, $|\mathcal{P}| = 1$. Standard test accuracy and poison test accuracy when varying the poison ratio $\beta$ and poison strength $\alpha$.

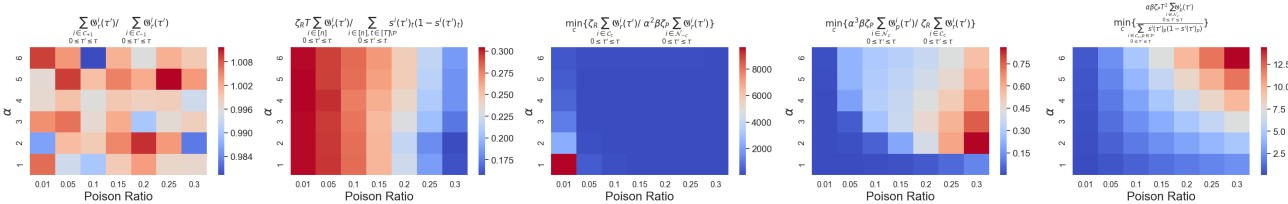

*Figure 14.* Dirty-label backdoor attacks. Set $n = 1K$, $d = 100$, $|\mathcal{R}| = 1$, $|\mathcal{P}| = 1$. Heatmap of the scaled ratio of each condition in Theorem 4.1 when varying poison strength $\alpha$ and poison ratio $\beta$.

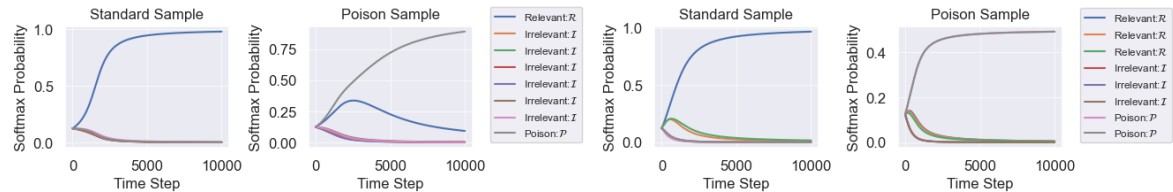

(a) $\alpha = 5.0, \beta = 0.1, |\mathcal{R}| = 1, |\mathcal{P}| = 1$. Final standard test accuracy is 1.0, poison test accuracy is 0.0.

(b) $\alpha = 3.0, \beta = 0.1, |\mathcal{R}| = 3, |\mathcal{P}| = 2$. Final standard test accuracy is 1.0, poison test accuracy is 1.0.

*Figure 15.* Clean-label backdoor attacks. Dynamics of softmax probability for a standard sample (left column) and a poison sample (right column), respectively.

## B.2. Clean-Label Backdoor Attacks

We consider the same synthetic data generation process as described in Section 3 except here we consider clean-label backdoor attacks. We set token length $T = 8$, dimension $d = 4000$, number of training samples $n = 20$. We start with plotting the dynamics of softmax probability for standard and poison training sample in Figure 15 and observe the same phenomena as in dirty-label backdoor attacks. Figure 16 plots the the heatmaps of standard test accuracy and poison test accuracy as the poison strength $\alpha$, poison ratio $\beta$ and poison token length $|\mathcal{P}|$ vary when setting the relevant token length $|\mathcal{R}|$ to be 1, 2, 3, respectively. We also validate the Theorem's conditions in Figure 17, 18 and 19. These results demonstrate that our theorem also holds for clean-label backdoor attacks.

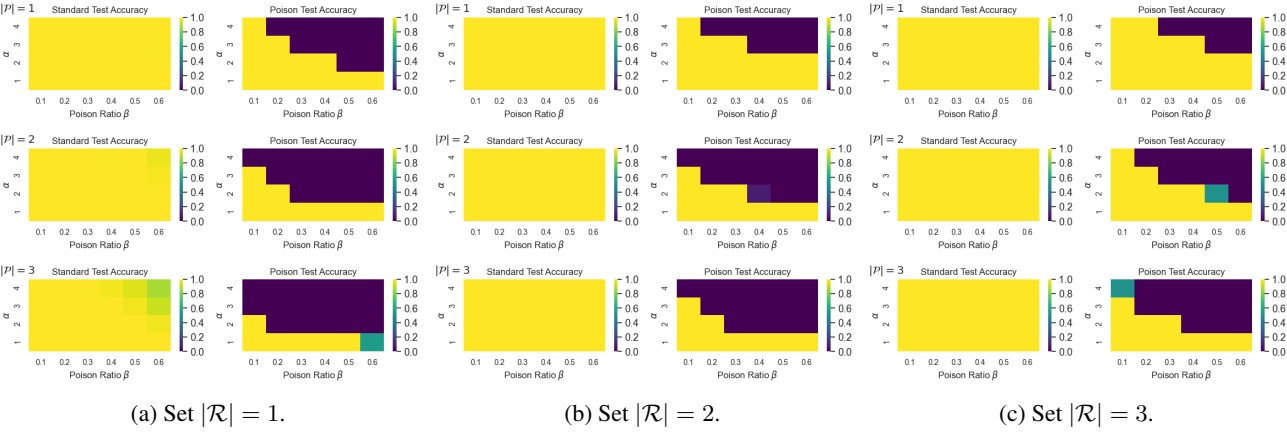

(a) Set $|\mathcal{R}| = 1$.   (b) Set $|\mathcal{R}| = 2$.   (c) Set $|\mathcal{R}| = 3$.

*Figure 16.* Clean-label backdoor attacks. Set $n = 20$, $d = 4000$. Standard test accuracy and poison test accuracy when varying the poison ratio $\beta$, poison token length $|\mathcal{P}|$ and poison strength $\alpha$.

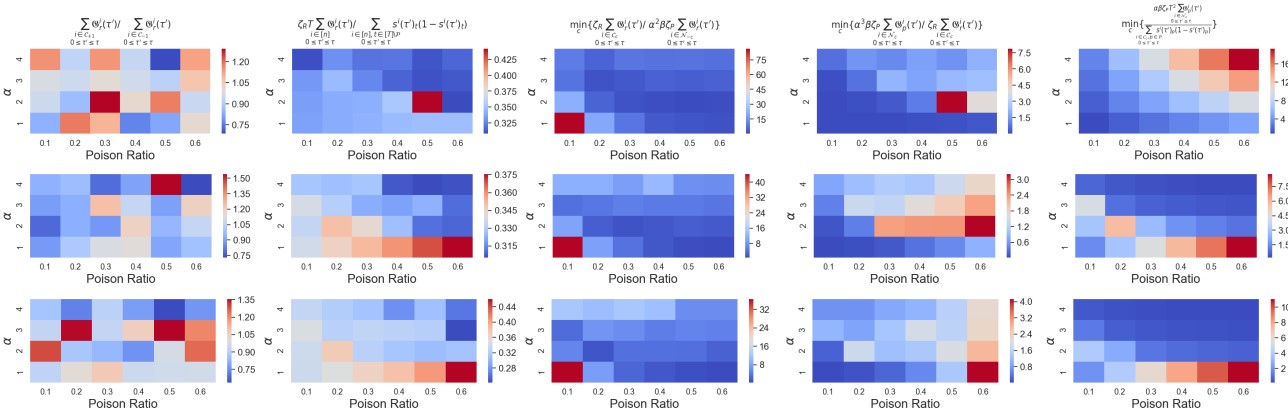

*Figure 17.* Clean-label backdoor attacks. Set $n = 20$, $d = 4000$. Heatmap of the scaled ratio of each condition in Theorem 4.1 when varying poison strength $\alpha$ and poison ratio $\beta$. Set $|\mathcal{R}| = 1$. From top row to bottom row represents poison length $|\mathcal{P}|$ from 1 to 3.

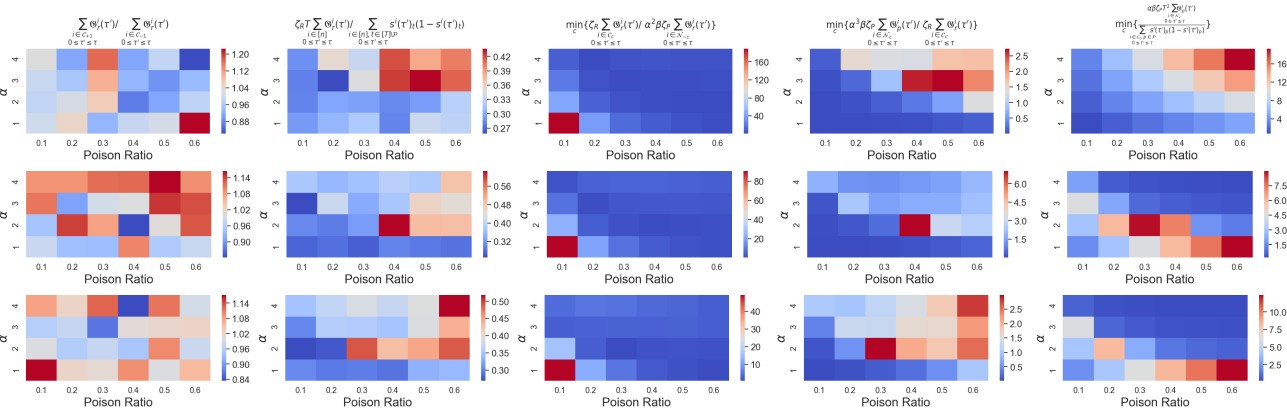

*Figure 18.* Clean-label backdoor attacks. Set $n = 20$, $d = 4000$. Heatmap of the scaled ratio of each condition in Theorem 4.1 when varying poison strength $\alpha$ and poison ratio $\beta$. Set $|\mathcal{R}| = 2$. From top row to bottom row represents poison length $|\mathcal{P}|$ from 1 to 3.

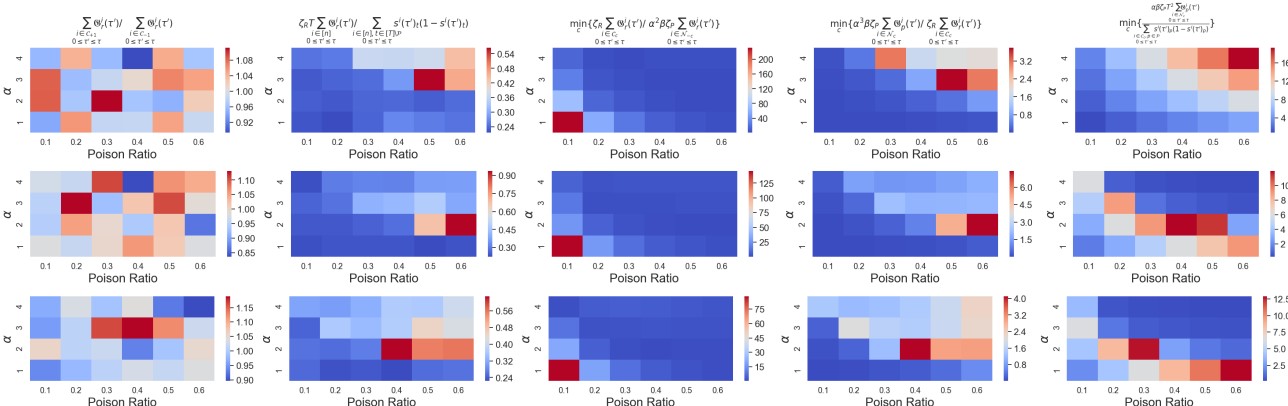

*Figure 19.* Clean-label backdoor attacks. Set $n = 20$, $d = 4000$. Heatmap of the scaled ratio of each condition in Theorem 4.1 when varying poison strength $\alpha$ and poison ratio $\beta$. Set $|\mathcal{R}| = 3$. From top row to bottom row represents poison length $|\mathcal{P}|$ from 1 to 3.

