# OpenReview forum: "Backdoor Attacks in Token Selection of Attention Mechanism"
_ICML.cc/2025/Conference — ICML 2025 poster_

### Official Review · Reviewer_fWct · 2025-03-09

**Overall Recommendation:** 3

**Summary:**

Motivated by the need for theoretical foundations underpinning backdoor attacks on self-attention transformers/LLMs (good), this paper: (1) investigates LLM backdoor attacks targeting the token selection mechanism of attention, (2) proves that "single-head attention transformers can interpolate poisoned training data through gradient descent", and (3) identifies the theoretical conditions enabling such attacks. These conditions are supported empirically with “simple experiments on synthetic datasets”.

**Claims And Evidence:**

The authors claim that single head self-attention transformers trained using gradient descent can interpolate poisoned training data and maintain good generalisation on clean data. This is evidenced using the mechanics of gradient descent and the probabilities of selecting relevant / poisoned tokens after training on standard / poisoned signal vectors. The proof seems sound. The extent of evidence / the proof is limited in scope by the focus on gradient descent which is more susceptible to overfitting than e.g., Adam or when regularisation is applied. Nevertheless, this is interesting and novel as a first step towards building a theoretical foundation – as stated by the authors. Empirical results back up the claim and proofs.

**Essential References Not Discussed:**

N/A

**Experimental Designs Or Analyses:**

A very small synthetic dataset is composed (n=20) and used to validate the earlier theoretical claims. 10 and 40% of the dataset is poisoned for different experiments. It does not seem that multiple training runs occurred but likely unnecessary because (line ~196) the weights are initialised to 0. Though limiting the ability to extrapolate, the experimental design allows for total control over the moving parts during optimisation and for the theory to be backed up empirically.

**Methods And Evaluation Criteria:**

Yes they seem to although the dataset size is very small (n=20) and the number of poisoned samples is a generously large proportion (10% and 40% is used). This seems OK for the purposes of validating the theoretical results but limits ability to extrapolate to real-world phenomena.

**Other Comments Or Suggestions:**

~ 22: “The behavior of backdoor attack” -> attacks
~24: The vulnerability of large language models (LLMs) -> you have already defined LLM acronym
~98: “e.g.” -> “e.g.,” for consistency with the rest of the paper.
~137: “The rest tokens remains unchanged” -> The rest of the tokens remain unchanged
~139: “1 control the strength of the poisoned signal.” -> controls the strength…
~152: “are generated i.i.d.” -> is generated i.i.d.
~313-316: “To interpolate all training data, Lemma 5.1 guarantees that the attention mechanism select a relevant token for clean training data, while prioritizes the poisoned tokens for poisoned training data.” -> “To interpolate all training data, Lemma 5.1 guarantees that the attention mechanism selects a relevant token for clean training data, yet prioritizes the poisoned tokens for poisoned training data.”

**Other Strengths And Weaknesses:**

The contributions seem novel and I appreciate the advancement of theoretical foundations underpinning attacks on transformer models. I think the paper would benefit from a more detailed framing of the result in terms of extrapolating to real-world attacks or defences.

**Questions For Authors:**

N/A

**Relation To Broader Scientific Literature:**

The authors position their contributions w.r.t the literature in Sections 1 and 2. Their proof and results are aligned and provide some theoretical foundations to explain prior work (e.g., Dai et al. 2019, Wan et al. 2023) ~ that backdoor attacks are feasible on language models. This work incorporates and extends work by Tarzanagh et al 2023a;b which proves convergence in the direction of a max-margin solution separating locally-optimal tokens from non-optimal tokens in the attention mechanism of transofmer models. The novelty w.r.t prior work is proving how gradient descent interpolates backdoors in the attention mechanism of a single-head self-attention transformer model.

**Theoretical Claims:**

I read the proof sketches in the main paper which seem sound (I did not attempt to parse the full proofs from the Appendix).

---

> ### Author Rebuttal · Authors · 2025-04-01
>
> We appreciate the reviewer's interest and recognition of our contributions and the novelty of our work. We will correct the typo in the final version.
>
> Regarding the connection with practical settings, we have some conjectures about possible defense mechanisms. Suppose that the learner has knowledge of the relevant tokens. In that case, a simple sanity check could be performed: after training the transformer by optimizing only the tunable token $\mathrm{p}$, one can examine whether $\mathrm{p}$ exhibits a strong correlation with signals that are not relevant tokens. If such a correlation exists, it may indicate that the transformer has been compromised by a backdoor attack.
>
> Since optimizing $\mathrm{p}$ is equivalent to optimizing $\mathrm{W}$, our results suggest that backdoor triggers are injected into the key and query matrices. Therefore, another potential defense strategy in practice would be to apply dropout layers for the attention model. By randomly masking out poisoned neurons in the key or query matrices, dropout could introduce inconsistencies in the model’s output if the model has been poisoned. One can also adopt the idea from [1] to detect poisoned data within the training sample by identifying cases where a small proportion of the extracted features differ significantly from the rest. These are immature ideas and require thorough investigation, particularly in the context of practical transformer architectures, which is beyond the scope of this paper.
>
> [1] Tran, Brandon, Jerry Li, and Aleksander Madry. "Spectral signatures in backdoor attacks." NeurIPS 2018.

---

### Official Review · Reviewer_Mvuk · 2025-03-10

**Overall Recommendation:** 3

**Summary:**

This paper discusses the vulnerability of the attention module to backdoor attacks from an interesting perspective, and this work provides theoretical analysis and simulation verification. This paper proves that a layer of attention module does remember poisoned samples after some assumptions are met.

**Claims And Evidence:**

The claims made in the paper are reasonable and verifiable.

**Essential References Not Discussed:**

N/A

**Experimental Designs Or Analyses:**

All the results of the experimental demonstration part of the paper are checked.

**Methods And Evaluation Criteria:**

The evaluation method used (simulation experiment of synthetic data) is reasonable, but has some limitations.

**Other Comments Or Suggestions:**

Some formulas in the paper have punctuation, some formulas lack punctuation, need to unify the use of punctuation

**Other Strengths And Weaknesses:**

This paper makes a theoretical analysis of the fragility of attention. My main concerns are as follows:

1. Is the time step tao_0 in Theorem 4.1 bounded? This time step needs to be large enough to make sure that the theorem holds, and what variables does this time step depend on. It needs to be analyzed that the time step tao is greater than tao_0 is actually a condition that can be met.

2. Lack of verification of theory correctness on real world data sets. For example, experimental demonstration can be performed on IMDB and sentiment140 datasets

3. In the introduction of poisoning data generation, the author says that the union of P and R needs to be an empty set. What is the reason for this condition?

**Questions For Authors:**

N/A

**Relation To Broader Scientific Literature:**

This paper presents their work in relation to the work discussing the security of LLM

**Theoretical Claims:**

The proof of theorem 4.1 provided in this paper is reasonable

---

> ### Author Rebuttal · Authors · 2025-04-01
>
> We appreciate the reviewer's interest and recognition of our contributions and the novelty of our work.
>
> W1: The lower bound on the number of iterations $\tau_0$ depends on the proportion of relevant tokens $\zeta_R$, the proportion of irrelevant tokens $\zeta_P$, the number of tokens $T$, and the strength of poisoned signal $\alpha$.
> $\tau\geq \tau_0$ is required to guarantee that, for any give $\epsilon>0$, the softmax probability of the relevant token is at least $1-\epsilon$ for the standard training sample, and the softmax probability of the poisoned token is at least $\frac{1}{|\mathcal{P}|}-\epsilon$ for the poisoned training sample. Such condition can be met due to Lemma 5.1.
> The proof of generalization guarantee requires $\epsilon\lesssim \min(\frac{T}{(\frac{1}{\zeta_R}-1)^4}, \frac{1}{(\frac{1}{\zeta_P}-1)^{\frac{4}{\alpha}-1}})$. Smaller $\epsilon$ leads to larger $\tau_0$.
>
> W3: We assume that the union of $\mathcal{P}$ and $\mathcal{R}$ is an empty set to guarantee that adding poisoned tokens does not alter the semantic meaning of the original input. Intuitively, if a poison pattern modifies an image in a way that changes the original object, or if a modified word changes the semantic meaning of the sentence, the classifier should not be expected to predict the original label. We will include a remark on this in the final version.
>
> We will unify the use of punctuation of formulas as suggested by the reviewer.

---

### Official Review · Reviewer_gcbi · 2025-03-11

**Overall Recommendation:** 3

**Summary:**

This paper presents a theoretical analysis of backdoor attacks targeting the token selection process in single-head self-attention transformers. The authors demonstrate that gradient descent can interpolate poisoned training data and establish conditions under which backdoor triggers dominate model predictions while preserving generalization on clean data. Empirical experiments on synthetic data validate the theoretical findings.

**Claims And Evidence:**

Yes

**Essential References Not Discussed:**

Theoretical work on multi-head attention is omitted but relevant for extensions.

**Experimental Designs Or Analyses:**

While useful for controlled analysis, real-world relevance is unclear. For example, real triggers (e.g., "James Bond") may exhibit complex interactions with context.

**Methods And Evaluation Criteria:**

- The experiments use synthetic data only and lack real-world benchmarks.
- Fixed linear head $\nu$ simplifies analysis but limits practical relevance.

**Other Comments Or Suggestions:**

None

**Other Strengths And Weaknesses:**

**Strengths**:
• Theoretically grounded conditions for attack success.
• Clear exposition of attention manipulation dynamics.

**Weaknesses**:
• Narrow scope (single-head, synthetic data).
• Assumptions may not generalize to real-world models.

**Questions For Authors:**

1. How do your theoretical conditions translate to real-world triggers (e.g., phrases or syntax patterns) that may correlate with natural tokens?
2. Could joint optimization of $\nu$ and $p$ weaken or strengthen backdoor success?
3. Have you tested the approach on transformers pre-trained on large corpora?
4. Are the conclusions revealed in the paper instructive for defense? Discussing this will help increase the value of the work.

**Relation To Broader Scientific Literature:**

This is the first theoretical study of backdoors in attention mechanisms (prior work focused on empirical attack designs).

**Theoretical Claims:**

• **Orthogonality Assumption**: Signals $\mu_{\pm 1}$ and $\tilde{\mu}_{\pm 1}$ are orthogonal (Assumption 1). In practice, triggers (e.g., rare words) may correlate with natural tokens, weakening the theory’s applicability.
• **Fixed Linear Head**: Training $\nu$ and $p$ jointly could alter dynamics; this is not addressed.

---

> ### Author Rebuttal · Authors · 2025-04-01
>
> We appreciate the reviewer's interest and recognition of our contributions and the novelty of our work.
>
> Q1: Regarding the orthogonality assumption, such assumption can be relaxed to the setting where the relevant signals $\mu_{\pm 1}$ and the poisoned signals $\tilde\mu_{\pm 1}$ are correlated, and our proof still holds with minor modifications. We discuss this on page 6, left column, lines 295–303. We can add more clarification in our final version.
>
> Q2: We conduct several experiments using the same synthetic dataset as described in the paper. We choose $|\mathcal{R}|=|\mathcal{P}|=1$ and vary $\beta$ across $0.1,0.2,0.3,0.4$ and $\alpha$ across $1,2,3,4$. We compare the poison accuracy between jointly optimizing $\nu$ and $\mathrm{p}$ versus optimizing only $\mathrm{p}$ under the same $\alpha$ and $\beta$. Our results show that while the final poison accuracy is similar in both cases after sufficient training iterations, joint optimization leads to a faster convergence rate. We hypothesize that jointly optimizing $\nu$ and $\mathrm{p}$ may strengthen the backdoor attack in more practical scenarios, such as when training on a more complex dataset or using a more sophisticated attention architecture. Understanding the effects of joint optimization remains an interesting research direction, which we highlight as a future avenue in our paper (page 6, right column, lines 322–324).
>
> Q3: We didn't run experiments on transformers pre-trained on large corpora.
>
> Q4: This is an excellent question. We have some conjectures about possible defense mechanisms. Suppose that the learner has knowledge of the relevant tokens. In that case, a simple sanity check could be performed: after training the transformer by optimizing only the tunable token $\mathrm{p}$, one can examine whether $\mathrm{p}$ exhibits a strong correlation with signals that are not relevant tokens. If such a correlation exists, it may indicate that the transformer has been compromised by a backdoor attack.
>
> Since optimizing $\mathrm{p}$ is equivalent to optimizing $\mathrm{W}$, our results suggest that backdoor triggers are injected into the key and query matrices. Therefore, another potential defense strategy in practice would be to apply dropout layers for the attention model. By randomly masking out poisoned neurons in the key or query matrices, dropout could introduce inconsistencies in the model’s output if the model has been poisoned. One can also adopt the idea from [1] to detect poisoned data within the training sample by identifying cases where a small proportion of the extracted features differ significantly from the rest. These are immature ideas and require thorough investigation, particularly in the context of practical transformer architectures, which is beyond the scope of this paper.
>
> [1] Tran, Brandon, Jerry Li, and Aleksander Madry. "Spectral signatures in backdoor attacks." NeurIPS 2018.
>
> Regarding missing references: we have discussed some of the theoretical work on multi-head attention in Section 2, page 2, right column, lines 72-78. We will include more references in our final version.

---

### Official Review · Reviewer_L1SQ · 2025-03-13

**Overall Recommendation:** 3

**Summary:**

This paper uses extensive mathematical proofs to reveal how backdoor triggers affect model optimization. If the signal from the backdoor trigger is strong enough but not overly dominant, an attacker can successfully manipulate the model predictions.

**Claims And Evidence:**

Yes.

**Essential References Not Discussed:**

none

**Experimental Designs Or Analyses:**

Yes, the experiment is simple and small just only to support the theoretical results...

**Methods And Evaluation Criteria:**

Yes.

**Other Comments Or Suggestions:**

N/A

**Other Strengths And Weaknesses:**

Strengths:

+Extensive mathematical proofs.

+Revealing how backdoor triggers affect model optimization.

+Reveals and defines the necessary conditions for a successful backdoor attack in a single-head self-attention transformer.

Weaknesses:

-The experiment is too simple.
Only test a single layer self-attention transformer.

-Whether vector L2 norm is truly representative of trigger signal strength in deep learning needs to be further explored.

-Too many assumptions, whether it applies in real large-scale transformer or large-scale datasets is a question.

**Questions For Authors:**

I do not have further questions to authors.

**Relation To Broader Scientific Literature:**

The process of backdoor attack has been investigated at both the mathematical analysis and theoretical levels and have contributed to the theory and interpretability of Transformer-based models.

**Theoretical Claims:**

Yes, but I'm a bit skeptical that the A1-5 assumptions hold up with practical and large data.

---

> ### Author Rebuttal · Authors · 2025-04-01
>
> We appreciate the reviewer's interest and recognition of our contributions and the novelty of our work. We acknowledge that our current results rely on restrictive assumptions and that our experiments serve primarily as a proof-of-concept for our theoretical findings. We have discussed these limitations in the paper. As this is the first work to address how backdoor triggers influence the optimization of attention-based models, our goal is to provide valuable insights into this problem. We agree that refining and relaxing these assumptions is an important direction for future research.

---

### Decision · Program_Chairs · 2025-05-01

**Decision:**

Accept (poster)

**Comment:**

The authors explore a type of backdoor attacks that exploit the token selection process within attention mechanisms. The authors provide a theoretical analysis demonstrating that single-head self-attention transformers can interpolate poisoned training data through standard gradient descent. They further show that when the poisoned data contains sufficiently strong, yet not overly dominant, backdoor triggers, adversaries can reliably influence model predictions.

The study offers valuable insight into the dynamics of how attention-based token selection can be manipulated to compromise model behavior. The authors derive theoretically grounded conditions under which such attacks succeed and support their analysis with empirical validation using synthetic datasets.